# Identification of multiple co-located hydrometeor types in Doppler spectra from scanning polarimetric cloud radar observations

Majid Hajipour [1], Patric Seifert [1], Hannes Griesche [1], Kevin Ohneiser [1], and Martin Radenz [1]

[1]Leibniz Institute for Tropospheric Research, Leipzig, Germany

**Correspondence:** Majid Hajipour (hajipour@tropos.de)

**Abstract.** To date, there remains a noticeable gap in reliable techniques for retrieving the shape and orientation of ice particles from observational data. This paper introduces a method using ground-based scanning polarimetric Doppler cloud radar to retrieve the shape and orientation of multiple hydrometeor types within deep mixed-phase clouds. Building on the strong performance of an existing method, which is effective in retrieving the shape and orientation of pristine ice particles in stratiform clouds, we extended this technique by analyzing the entire Doppler spectrum. The previously developed main-peak approach focuses only on the part of the Doppler spectrum with the highest signal-to-noise ratio to retrieve the shape and orientation of the dominant hydrometeor types within stratiform clouds. With the extended technique, referred to as the spectrally resolved approach, the section of the Doppler spectrum containing valid data points exceeding the noise level is analyzed by dividing it into five equally spaced parts. This allows to retrieve up to five distinct velocity-segregated hydrometeor types. The technique utilizes Range-Height Indicator (RHI) scans (ranging from $30\,°$ to $90\,°$ elevation) of the Doppler spectra of differential reflectivity (ZDR) and correlation coefficient (RHV) from a polarimetric Ka-band cloud radar. The potential of the improved approach is presented by means of two case studies. The first case demonstrates the effectiveness of the spectrally resolved approach and in the second case secondary ice production is investigated. These findings contribute to a profound understanding of hydrometeor characteristics, shedding light on dynamic cloud processes, especially in the context of precipitation and ice particle formation.

## 1 Introduction

Integral to Earth's climate system, clouds play a crucial role in the global water cycle. Mixed-phase clouds, containing both liquid droplets and ice crystals (Korolev et al., 2017), are a major source of precipitation in mid-latitude regions (Mülmenstädt et al., 2015). These clouds can form within the temperature range from $-38\,°C$ and $0\,°C$. Ice particles can form only at temperatures below $0\,°C$, while liquid droplets can exist in the atmosphere down to about $-38\,°C$. Ice particles can form via several distinct ways, such as homogeneous ice formation (Austin and Jeffery, 1997), heterogeneous ice formation (Hoose and Möhler, 2012), or secondary ice production (SIP) (Korolev and Leisner, 2020). Consequence is, that usually, multiple hydrometeor types do co-exist in a cloud volume as long as the cloud system extends over a diverse range of temperature, humidity and dynamics.

In general, mixed-phase clouds can be categorized into deep or stratiform types based on their vertical extent. Deep clouds exhibit a distinct vertical structure, often resulting in complex aggregated or rimed ice crystals. In contrast, stratiform mixed-phase clouds are characterized by a shallow layer, dominated by supercooled liquid droplets at the cloud top that contains predominantly pristine crystals which may precipitate further downward (Field et al., 2005; Zeng et al., 2023; Qi et al., 2019). Deep mixed-phase clouds display diverse microphysical processes that are affected by ambient temperature, pressure, and humidity as well as their dynamics-driven variability (Pruppacher et al., 1998). The microphysical processes within mixed-phase clouds include coalescence, aggregation, riming, sublimation, crystal breakup, and melting, which influence the size, concentration, and chemical composition of ice crystals (Hallet and Mossop, 1974). Studying the complex shapes of ice crystals in various temperature and saturation environments is crucial for advancing our understanding of the microphysical processes within clouds (Matsuo and Fukuta, 1987; Guichard and Couvreux, 2017).

Ground-based radar measurements, particularly those utilizing polarimetric cloud radars, have been crucial in examining the microphysical properties of ice crystals. Polarimetric cloud radars, especially at higher frequencies such as Ka-band or W-band, provide increased sensitivity to ice crystals, enhancing the remote sensing of cloud microphysical properties (Kollias et al., 2007; Küchler et al., 2017; Görsdorf et al., 2015). The differential depolarization and backscattering characteristics of incident electromagnetic waves caused by hydrometeors offers valuable insights into their phase, shape, aspect ratio, orientation, and bulk density. Cloud Doppler radars, provide backscattered signal as function of Doppler velocity referred to as the Doppler spectrum. As the terminal fall speed of ice particles depends on their size, mass, and shape (Mitchell, 1996; Bühl et al., 2019), the Doppler spectrum can be considered as a fingerprint of the cloud microphysical structure. Early work by Matrosov (1991) demonstrated the possibility to distinguish prolate-shaped scatterers from oblate ones based on the depolarization ratio's dependency on the elevation angle of the radar antenna. Melnikov and Straka (2013) further showed that assuming a spheroid shape model, a combined analysis of polarimetric variables could provide information about ice particle habits. Matrosov (2020) expanded these efforts by evaluating the aspect ratio of ice hydrometeors in Arctic clouds using polarimetric radar-based retrievals, highlighting the potential to obtain information about particle growth and microphysics. Myagkov et al. (2016a) introduced an approach, hereafter denoted as main-peak approach, utilizing only the peak signal of spectral signal-to-noise ratio (SNR) from the cloud radar Doppler spectrum at a range of elevation angles at a given height level to retrieve the shape and orientation of the dominant ice particle type through the analysis of backscatter intensity. This methodology involves comparing elevation (range-height indicator, RHI) scans of observed differential reflectivity (ZDR) and correlation coefficient (RHV) of particles with simulated angular dependencies of ZDR and RHV. The resulting output consists of a pair of values: the polarizability ratio (density-weighted axis ratio) and the degree of orientation. The polarizability ratio indicates the shape, while the degree of orientation indicates the orientation. These values represent the best match between the measured and modeled angular dependencies of the polarimetric variables. The main-peak approach was demonstrated by Myagkov et al. (2016b), to yield valuable information about the shape and orientation of primarily formed ice particles which formed in thin stratiform mixed-phase clouds. Since the polarizability ratio and degree of orientation are analytically defined, they can in principle be used in model-observations comparison studies. This was, e.g., recently done for the polarizability ratio by Welss et al. (2024). Since clouds typically contain a mixture of various hydrometeor types, even at small spatial scales, the main-peak

approach is however insufficient for retrieving multiple hydrometeor types. Therefore, this approach needs to be extended to account for the diverse compositions of clouds.

The measured polarimetric variables of Doppler cloud radars exhibit sensitivity to shapes of hydrometeors, allowing the identification of the ice particle types (Shupe et al., 2004; Luke et al., 2010; Verlinde et al., 2013; Kalesse et al., 2016; Kollias et al., 2016, 2007; Fukao and Hamazu, 2014; Radenz et al., 2019; Mak and Unal, 2025). Pristine ice particles predominantly are observed in the low Doppler velocity range (down to $0\,\mathrm{m\,s^{-1}}$), while heavily rimed or large aggregated ice particles tend to be situated in the high Doppler velocity range, with aggregates saturating at around $1\,\mathrm{m\,s^{-1}}$ and graupel exceeding even $2\,\mathrm{m\,s^{-1}}$. This paper presents an extension of the main-peak approach, which involves analyzing five parts of the Doppler spectra observed over a range of elevation angles from 30 to $90\,^\circ$ to yield vertical profiles of polarizability ratio and degree of orientation as a function of Doppler velocity. To do so, cloud radar data from the ACCEPT (Analysis of the Composition of Clouds with Extended Polarization Techniques) campaign conducted in 2014 at Cabauw, Netherlands, will be analyzed and the potential of the proposed method will be highlighted. The whole cloud radar Doppler spectrum is utilized, with the goal of achieving a comprehensive retrieval of all hydrometeor types present in the clouds. In section 2, the instrumentation employed in this study is introduced. In section 3, we present a detailed exposition of the extended approach, providing insights into the enhancements made to the existing methodology. The practical application of the spectrally resolved approach is explored in section 4. Additionally, the capability of this approach to identify secondary ice production is investigated in a second case study, presenting its versatility in addressing different aspects of hydrometeor characterization within clouds.

## 2   Instrumentation

The ACCEPT measurement campaign was led by the Leibniz Institute for Tropospheric Research (TROPOS) in Leipzig, Germany, with partners from the Technical University of Delft and METEK GmbH in Elmshorn, Germany. One of its goals was to explore the polarimetric capabilities of the MIRA-35 cloud radar. A second objective was the study of ice particle growth processes in mixed-phase clouds using spectral polarimetric S- band radar measurements (Pfitzenmaier et al., 2017, 2018). The campaign took place at the Cabauw Experimental Site for Atmospheric Research (CESAR) in the Netherlands from October 7 to November 18, 2014. Table 1 provides a comprehensive list of the instruments utilized specifically for this study. Two Doppler cloud radars of type Mira-35 (Görsdorf et al., 2015), were installed and operated during the campaign to cover different objectives. One MIRA-35 (MIRA-35 MBR4) configured in hybrid polarimetric mode (simultaneously transmitting and receiving signals across both horizontal and vertical channels.), was integrated into the LACROS (Leipzig Aerosol and Cloud Remote Observations System; Bühl et al., 2013) container at CESAR. It performed 15-minute scanning cycles, including two RHI scans per cycle, covering elevation angles from $30\,^\circ$ to $150\,^\circ$ at two perpendicular azimuth angles: $0\,^\circ$ (north-south) and $90\,^\circ$ (east-west), with a scanning speed of $0.5\,^\circ$ per second. Additionally, each cycle included one PPI (Plan Position Indicator) scan, sweeping azimuth angles from $0\,^\circ$ to $360\,^\circ$ at two elevation angles ($75\,^\circ$ and $105\,^\circ$), at a scanning speed of $0.3\,^\circ$ per second. This system was in detail described by Myagkov et al. (2016a). The radar's accuracy was ensured through antenna coupling corrections and occasional calibration using light rain events (Myagkov et al., 2015, 2016a). Another MIRA-35

radar (MIRA-35 NMRA), with a linear depolarization configuration (LDR), operated simultaneously in continuous vertical-stare mode to observe cloud systems. To prevent interference, the two radars operated slightly at different frequencies (see

Table 1) and were positioned 30 meters apart. A PollyXT multiwavelength Raman lidar (Engelmann et al., 2016) was used to detect supercooled liquid particles in mixed-phase clouds near the Cabauw meteorological tower. The lidar was strategically tilted by $5\,^\circ$ off-zenith to avoid specular reflection at the planar planes of horizontally oriented falling ice crystals (Westbrook et al., 2010). Cloud-top temperatures were determined from the thermodynamic profiles of the atmosphere derived by various methods, including radiosondes, microwave radiometer, and the GDAS1 model dataset, with priority on the first available

dataset in the given order (Myagkov et al., 2016b).

    The analysis was focused on mid-level mixed-phase clouds by meeting specific criteria, including the operation of the MIRA-35 MBR4, availability of calibrated polarimetric variables, and cloud-top temperatures ranging from approximately $-20\,^\circ$ to $0\,^\circ$C. The goal was to study ice crystal properties in non-precipitating cases, considering the shape retrieval algorithm's assumptions.

## 2.1   Mira-35 radar in hybrid mode

METEK GmbH, based in Elmshorn, Germany, manufactures the widely-used MIRA-35 cloud radar operating at 35 GHz. It is extensively employed across European measurement sites, particularly within the Cloudnet (Illingworth et al., 2007; Tukiainen et al., 2020) framework of the Aerosols, Clouds, and Trace Gases Research Infrastructure (ACTRIS). Renowned for its high sensitivity and reliability, MIRA-35's popularity is validated by various studies (Martucci and O'Dowd, 2011; Di Girolamo

et al., 2012; Bühl et al., 2013; Griesche et al., 2020; Radenz et al., 2019). Görsdorf et al. (2015) offer comprehensive insights into the technical aspects of MIRA-35, accuracy considerations, and operational statistics based on over a decade of continuous measurements.

    Unlike traditional MIRA-35 cloud radars operating in LDR-mode, which are commonly used for clutter filtering and detecting the melting layer, this paper employs the hybrid mode to study the ice crystal shape and orientation in detail. The hybrid

mode, commonly used in weather radars (Ryzhkov et al., 2005), simultaneously transmits horizontal and vertical components of the signal, eliminating the need for expensive high-pulse-power polarization switching. Occasionally, this mode is also referred to as Simultaneous Transmission, Simultaneous Reception (STSR) mode. As demonstrated, the hybrid mode provides polarimetric parameters for a quantitative estimate of particle shape and orientation characteristics. In this paper the differential reflectivity (ZDR) and correlation coefficient (RHV), key parameters of polarimetric radar, are explored.

The observation by a polarimetric cloud radar is based on the capabilities of this radar type to detect the returned signals in two detection channels as function of Doppler velocity $\omega$ which are representative of the horizontally ($\dot{E}_\mathrm{h}(\omega)$) and vertically ($\dot{E}_\mathrm{v}(\omega)$) polarized planes of the received waves. Based on these two observables, polarimetric properties of the scatterers can be determined. Those are introduced in the following. It should be noted that detailed explanation of the calibration and calculation of polarimetric variables can be found in Myagkov et al. (2016a).

**Table 1.** Characteristics of the instruments employed during the ACCEPT campaign.

| Instruments | Main specifications | Measurements | References |
|---|---|---|---|
| MIRA-35 NMRA | Frequency: 35.5 GHz<br>configuration: LDR<br>pointing: zenith<br>temporal resolution: 1 s<br>Range resolution: 30 m<br>Doppler velocity resolution: 0.08 m s$^{-1}$ | Radar reflectivity factor<br>LDR<br>mean Doppler velocity<br>Doppler width<br>complete Doppler spectrum | Görsdorf et al. (2015) |
| MIRA-35 MBR4 | Frequency: 35.17 GHz<br>configuration: hybrid<br>pointing: scanning<br>temporal resolution: 1 s<br>range resolution: 30 m<br>Doppler velocity resolution: 0.08 m s$^{-1}$ | Radar reflectivity factor<br>mean Doppler velocity<br>Doppler width<br>complete Doppler spectrum<br>differential reflectivity<br>correlation coefficient<br>differential phase shift | Myagkov et al. (2016a) |
| Multiwavelength Raman lidar PollyXT | Wavelengths:<br>355 nm, 532 nm, 1064 nm<br>pointing angle: 5 ° off-zenith<br>temporal resolution: 30 s<br>range resolution: 7.5 m | Backscatter coefficient,<br>volume depolarization ratio | Althausen et al. (2009) |
| Microwave radiometer HATPRO | Bands: 22–31 GHz,<br>51–58 GHz<br>temporal resolution: 1 s | Brightness temperatures<br>temperature profile<br>liquid water path<br>integrated water vapor | Rose et al. (2005) |
| Radiosonde Vaisala RS92 | Temporal resolution is fixed,<br>range resolution can be<br>estimated by assuming an<br>ascend speed of 5 m s$^{-1}$ | Temperature<br>air pressure<br>relative humidity<br>horizontal wind vector | Suortti et al. (2008) |

ZDR quantifies the ratio of reflectivity measurements in horizontal ($Z_{\mathrm{hh}}$, Eq. 1) and vertical ($Z_{\mathrm{vv}}$, Eq. 2) polarizations, (Eq. 3). $\langle \rangle$ denotes averaging over a series of pulses.

$$Z_{\mathrm{hh}}(\omega) = C_1 \left\langle \dot{E}_{\mathrm{h}}(\omega) \dot{E}_{\mathrm{h}}(\omega)^* \right\rangle \tag{1}$$

$$Z_{\mathrm{vv}}(\omega) = C_2 \left\langle \dot{E}_{\mathrm{v}}(\omega) \dot{E}_{\mathrm{v}}(\omega)^* \right\rangle \tag{2}$$

$$ZDR(\omega) = \frac{Z_{\mathrm{hh}}(\omega)}{Z_{\mathrm{vv}}(\omega)}. \tag{3}$$

In Eq. 1 and 2, the dot above $\dot{E}_{\mathrm{h}}(\omega)$ and $\dot{E}_{\mathrm{v}}(\omega)$ indicates that these are complex-valued quantities. Also, the constants $C_1$ and $C_2$ depend on the radar system parameters such as transmitted power, radar geometry, wavelength of the radar signal,

and system gains. ZDR is expressed in decibels (dB). At zenith-pointing direction, ZDR is zero. At slant-pointing direction, a positive ZDR value may indicate ice particles that are horizontally aligned, whereas a negative value might suggest particles aligned vertically (Seliga and Bringi, 1976).

The correlation coefficient (RHV) is a crucial polarimetric parameter that quantifies the similarity between horizontally ($Z_{hh}$) and vertically ($Z_{vv}$) polarized backscattered signals. It provides insight into the diversity of particle shapes and orientations within a radar resolution volume and is typically expressed as a value between 0 and 1 (Defined by Eq. 4).

$$RHV(\omega) = \frac{\left\langle \dot{E}_{h}(\omega)\dot{E}_{v}(\omega)^{*} \right\rangle}{\sqrt{\left\langle \dot{E}_{h}(\omega)E_{h}(\omega)^{*} \right\rangle \left\langle \dot{E}_{v}(\omega)\dot{E}_{v}(\omega)^{*} \right\rangle}} \tag{4}$$

    A correlation coefficient of 1 indicates perfect correlation between horizontal and vertical polarizations signals, typically
implying uniform particle type, shape, and orientation within the radar resolution volume. This suggests consistent scattering behavior, often associated with isotropic scatterers such as spherical particles. Conversely, low values indicate a low correlation, associated with diverse or irregular scattering particles (Caylor et al., 1990). In polarimetric radar applications, this parameter is applied in the differentiation between precipitation types and in the identification of specific scattering mechanisms. For example, raindrops, with a slightly spheroidal shape and uniform orientation, often exhibit higher correlation coefficients,
while snow or hail particles, with irregular shapes and orientations, may result in lower correlation coefficients.

    Linear Depolarization Ratio (LDR) is a parameter frequently measured by cloud radars which emit linearly polarized radiation only in one channel, but receive the co- and cross-polarized components of the backscattered signal. This kind of radar is usually referred to as LDR-mode cloud radar. The complex amplitudes of the received pulses in both the co-channel and cross-channel can be expressed as follows:

$\dot{E}_{c}(\omega) = E_{c}(\omega)\exp(i\phi_{c}(\omega))$                 (5)

$\dot{E}_{x}(\omega) = E_{x}(\omega)\exp(i\phi_{x}(\omega))$                 (6)

where $E_{c}$, $\phi_{c}$ and $E_{x}$, $\phi_{x}$ are amplitudes and phases of the received pulses in the co- and cross-channels, respectively. The LDR can be expressed as follows:

$$LDR(\omega) = \frac{\left\langle E_{x}^{2}(\omega) \right\rangle}{\left\langle E_{c}^{2}(\omega) \right\rangle} \tag{7}$$

LDR helps distinguish scatterer types within a radar pulse by measuring randomness or isotropy in particle orientation. Low LDR values indicate more ordered or isotropic scattering, often suggesting isometric particles well-aligned horizontally (Trömel et al., 2021).

## 3 Methodology

This section introduces the spectrally resolved shape and orientation retrieval. As the basis for this retrieval is the main-peak approach that was previously developed by Myagkov et al. (2016a), we will begin by outlining the principles of the established retrieval, followed by a detailed illustration of the spectrally resolved approach.

### 3.1 Main-peak approach

Melnikov and Straka (2013) showed that the shape of ice particles can be determined by analyzing how polarimetric radar variables change with elevation angle. This idea is used in methods like the main-peak approach by Myagkov et al. (2016a) and the vertical distribution of particle shape (VDPS) method developed later by Teisseire et al. (2024).

In the main-peak approach, the core idea is to study how two key radar measurements—differential reflectivity (ZDR) and correlation coefficient (RHV)—change with elevation angle at different heights in the atmosphere. These measurements are then compared with simulated values based on a spheroid model that assumes the particles are shaped like spheroids (3D ellipsoids) and that Rayleigh scattering applies. The goal is to find the best match between the real radar data and the model predictions. The introduction of the retrieval method, error analysis approach, as well as its evaluation are in detail presented by Myagkov et al. (2016a). For the sake of space we kindly point the reader to their publication for obtaining deep insights into the technique. In here, we only provide a general introduction of the retrieval.

In the simulation part of the method, ZDR and RHV values are calculated for many combinations of particle shapes, densities (related to their refractive index), and orientations, across a wide range of elevation angles (from $30°$ to $150°$). For doing so, Myagkov et al. (2016a) utilized spheroidal (Rayleigh-) scattering theory which enables to establish a direct relationship between the observables elevation angle, ZDR, and RHV and the particle's properties of density-weighted axis ratio and the distribution of the canting angles. The density-weighted axis ratio is denoted polarizability ratio $\xi_e$ which can be represented as the ratio of polarizability elements $p_1$ and $p_2$ (Eq. 8):

$$\xi_e = \frac{\langle p_2 \rangle}{\langle p_1 \rangle} \tag{8}$$

which polarizability elements $p_1$ and $p_2$ are defined as Eq. 9:

$$p_{1,2} = V \epsilon_0 (\epsilon_r - 1) A_{1,2}(\xi_g) \tag{9}$$

where $V$ is the volume of the spheroid, $\epsilon_0$ is the vacuum permittivity, $\epsilon_r$ is the relative permittivity, and $A_{1,2}(\xi_g)$ are function of the axis ratio $\xi_g$. The polarizability ratio ranges from $0.3$ to $2.3$. Within this range, values of $\xi = 0.3$ represent strongly oblate particles, $\xi = 2.3$ indicate strongly prolate particles, and $\xi = 1$ signifies centrally positioned spherical particles. As the polarizability ratio is also a function of the particle refractive index, i.e., density, its absolute value approaches unity for values of very low density. This aspect has to be considered in the interpretation of $\xi$.

The general orientation distribution of the hydrometeor population is described by the degree of orientation $\kappa$ Hendry et al. (1976) and Eq. 10.

$$\kappa = 1 - 2T_1 \tag{10}$$

which $T_1$ is defined as Eq. 11:

$$T_1 = \int_{-\frac{\pi}{2}}^{\frac{\pi}{2}} \sin^2(\theta + \theta_0) W(\theta)\, d\theta \tag{11}$$

where $\theta$ and $\theta_0$ is considered as orientation angle and horizontal angle of particle (which is $\theta_0 = 0$ for oblate spheroids and $\theta_0 = \frac{\pi}{2}$ for prolate spheroids) respectively, and $W$ is the probability density of orientation angle. The values of $\kappa$ range from $-1$ to 1, where $\kappa = -1$ denotes a horizontal orientation for prolate particles and a vertical orientation for oblate particles. Conversely, a value of $\kappa = +1$ signifies a vertical orientation for prolate particles and a horizontal orientation for oblate particles. A value of 0 indicates randomly oriented particles.

The complex scattering amplitudes are derived using the polarizability ratio and degree of orientation, which vary with elevation angle, to calculate ZDR and RHV. For further details, we refer readers to Section 3.1 of Myagkov et al. (2016a). It should be noted that Myagkov et al. (2016a) demonstrated that the retrieval also works with observations of slanted linear depolarization ratio (SLDR) and co-cross correlation coefficient (RHX) as they are provided by LDR-mode cloud radars, as it was demonstrated by (Teisseire et al., 2024).

On the observational side, the analysis focuses only on the main peaks in the radar's Doppler spectra, which represent the strongest signals detected at different angles. To make sure the measurements are reliable, the method uses the signal-to-noise ratio (SNR)—a measure of how strong the signal is compared to background noise. SNR is usually expressed as a ratio or in decibels (dB).

Finally, the method finds the best agreement between the measured and simulated ZDR and RHV values using error functions which compare the deviation of simulated from observed ZDR and RHV for all elevation angles of the deployed RHI scans. From this match, it can determine the most likely shape and orientation of the ice particles. Myagkov et al. (2016a) employed the polarizability ratio $\xi$ as a density-weighted axis ratio.

As the retrieval method deploys RHI scans from 30° to 150° elevation angle, homogeneity of the probed cloud layers has to be given. While actual homogeneity is hard to be evaluated, two measures are applied to ensure that homogeneity of the hydrometeor distribution over the horizontal range covered by the RHI scan is given. First, a minimum amount of 50% of all data points of an RHI scan have to contain valid values of ZDR and RHV. Second, the monotonic behaviour of the elevation-dependency of ZDR and RHV was evaluated. As can be seen in the theoretical model of the relationship between ZDR, RHV, polarizability ratio and degree of orientation (see Fig. 1 by Myagkov et al. (2016a)), the elevation dependency is always required to be monotonic. When a non-monotonic behavior of the elevation dependency of ZDR and RHV is detected, then the

presence of inhomogeneities is likely and analysis of the case should be skipped. Third, also fluctuations in the horizontal wind field might indicate an inhomogeneity of the cloud layers. We thus corrected the Doppler spectra at each elevation angle of an RHI scan for influences by the horizontal wind field, as is explained further in Section 3.3. If the resulting Doppler spectogram

as a function of elevation angle shows too high variability, care should be taken in the application of the shape and orientation retrieval.

## 3.2  Spectrally resolved approach

Ice particles adopt varying shapes, such as oblate, prolate or irregular, at the top of the cloud before descending based on their size. Along their descent, these particles interact by colliding either with other ice particles or with supercooled liquid

droplets, or they grow by water vapour diffusion. The growth processes induce alterations in their shape, size, fall speed, and trajectory. Radar signals received from these particles can be used to discern their reflectivity and fall speed, when the radar is pointed vertically (e.g., to $90°$ elevation). Consequently, the Doppler spectra observed with a vertically pointing cloud radar offer insights into the variability of sizes and shapes of the ice particles (Bühl et al., 2019; Vogl et al., 2024). In the low-velocity range around $0\,\mathrm{m\,s^{-1}}$ typically cloud droplets or small or low-density ice crystals, such as primarily ice crystals or secondarily

formed ice occur. At intermediate velocities from $\approx -0.5$ to $-1.5\,\mathrm{m\,s^{-1}}$ usually aggregates of ice crystals or drizzle droplets exist. The high-velocity range from $-2\,\mathrm{m\,s^{-1}}$ and faster, typically corresponds to graupel or small hail. Rain droplets and large hail can take much faster fall velocities of $5$ to $10\,\mathrm{m\,s^{-1}}$. The width of the Doppler spectra is thereby characterized by size- and shape-dependent fall velocities of the particles, which are super-imposed by influences of turbulence and (predominantly in case of off-zenith antenna pointing angle) horizontal wind variability that cause additional broadening of the spectrum (Radenz

et al., 2019). Nevertheless, as the above-mentioned references point out, zenith-pointing cloud radar observations of Doppler spectra only allow for the classification of different co-located particle shapes when certain assumptions are taken which are usually based on the observations of particle fall velocity or the temperature regime of the particle formation.

This study thus extends the main-peak shape and orientation retrieval through the spectrally resolved approach, which assumes that different hydrometeor types in a cloud volume are separated by their distinct fall speeds, as explained in the

240 previous paragraph and as illustrated in Fig. 1. In Figure 2 the block diagram of the spectrally resolved approach is presented. As for the main-peak approach, the spectrally-resolved shape and orientation retrieval is based on observations of the Doppler spectrum over a range of elevation angles. In contrast to the main-peak approach, the Doppler spectra observed at all elevation angles are divided into five equal parts, as illustrated in Fig. 1. To achieve this, for each Doppler spectrum, the starting and ending points are identified, then the spectrum's width is divided evenly into five parts. The average values of ZDR and

245 RHV are then calculated for each part. These determined ZDR and RHV values are subsequently used for the independent retrieval of hydrometeor shape and orientation. While the conventional main-peak approach provides only one pair of ZDR and RHV, this approach yields five pairs for each Doppler spectrum over the range of elevation angles. Thereby, in principle any amount of spectral parts can be selected in the algorithm. The amount of five spectral parts was empirically chosen based on findings from prior studies (e.g.,  Shupe et al., 2004; Kollias et al., 2007; Kalesse et al., 2016; Vogl et al., 2024) which

showed that Doppler spectra observed in mixed-phase clouds commonly exhibit up to five distinguishable spectral peaks,

which are typically associated to different particle populations (e.g., drizzle, different habits of small primary and secondary cloud ice, aggregates, or rimed particles). On the other hand, the shape and orientation retrieval approach requires a sufficient amount of data points and homogeneity at all elevation angles of the analyzed RHI scans, as is discussed in Section 3.1. A number of 5 Doppler spectra parts thus represents a practical balance between spectral complexity and interpretability. Note, that the different parts are not necessarily associated to individual peaks in the Doppler spectra, as they are e.g. derived by the peakTree method of Radenz et al. (2019). As will be outlined below, the observation of a cloud layer at different elevation angles makes it virtually impossible to track individual peaks in the Doppler spectra over the range of elevation angles. Instead, we assume that the fall attitude of the individual hydrometeor types contained in the cloud volume is similar at all elevation angles. E.g., the fastest-falling Doppler spectrum part at all elevation angles is associated to one hydrometeor type, and so on. Generally speaking, instead of spectral peaks the spectrally resolved approach aims on identification of spectral regimes of distinct hydrometeor properties.

Subsequent to the separation of the Doppler spectra at all elevation angles, the spectrally resolved approach operates akin to the main-peak approach, but for each spectral part separately. It compares the observed values of ZDR and RHV with their modeled counterparts using minimum mean square error function for each data point. By identifying the best agreement, the approach retrieves the polarizability ratio and degree of orientation. Thus, five sets of values for polarizability ratio and degree of orientation are obtained for each height level, signifying to distinguish up to five distinct hydrometeor shapes and orientations.

The uncertainties of the spectrally resolved retrieval are treated similar as introduced by Myagkov et al. (2016a) and Myagkov et al. (2016b) for the main-peak approach. During averaging of all ZDR and RHV values that fit to each range gate and elevation angle interval, respectively, also the standard deviations of both parameters are determined. It should be noted that the number of values contributing to the average depends on the elevation angle resolution of the RHI scan; in this study, typically 121 elevation angles are used. The retrieval is than also applied to pairs of ZDR and RHV, considering two times their standard deviation, which yields a measure of uncertainty.

In order to utilize the spectrally resolved approach, the Doppler spectra observed at the different elevation angles need to be harmonized as good as possible to derive the aforementioned estimate of the vertical-stare radial velocity for all angles. With increasing off-zenith angle, signatures of the falling particles get increasingly masked by contributions of the horizontal wind component. For the spectrally-resolved retrieval, it is thus aspired to obtain a best-estimate of the vertical component of the radial velocity at each elevation angle of the analysed RHI scans.

### 3.3 The influence of air motion on the Doppler spectra observed by a scanning cloud radar

The new spectrally resolved approach is based on the assumption of the ideal condition that the variability of the Doppler spectrum is caused by the differential fall motions of particles of distinct shape and orientation characteristics. Further, the signatures of the different shapes are required to be present homogeneously in the Doppler spectra from all elevation angles probed by the cloud radar in the RHI scans of the shape retrieval, i.e., usually the range from 0 to $60\,^\circ$ off-zenith angle ($30\text{–}90\,^\circ$ or $90\text{–}150\,^\circ$ elevation angle). In reality, atmospheric air motions occur, which introduce certain biases to the particle-

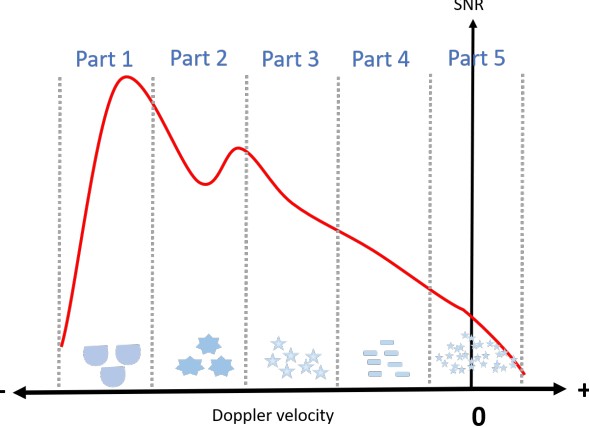

**Figure 1.** Illustration of an idealized Doppler spectrum containing 5 different hydrometeor types. The division into five parts enables the potential identification of up to five different hydrometeor types; however, the actual number of retrievable distinct types depends on the spectral and microphysical characteristics of the observed cloud. Positive Doppler velocities indicate the impact of turbulent motion on the Doppler spectrum.

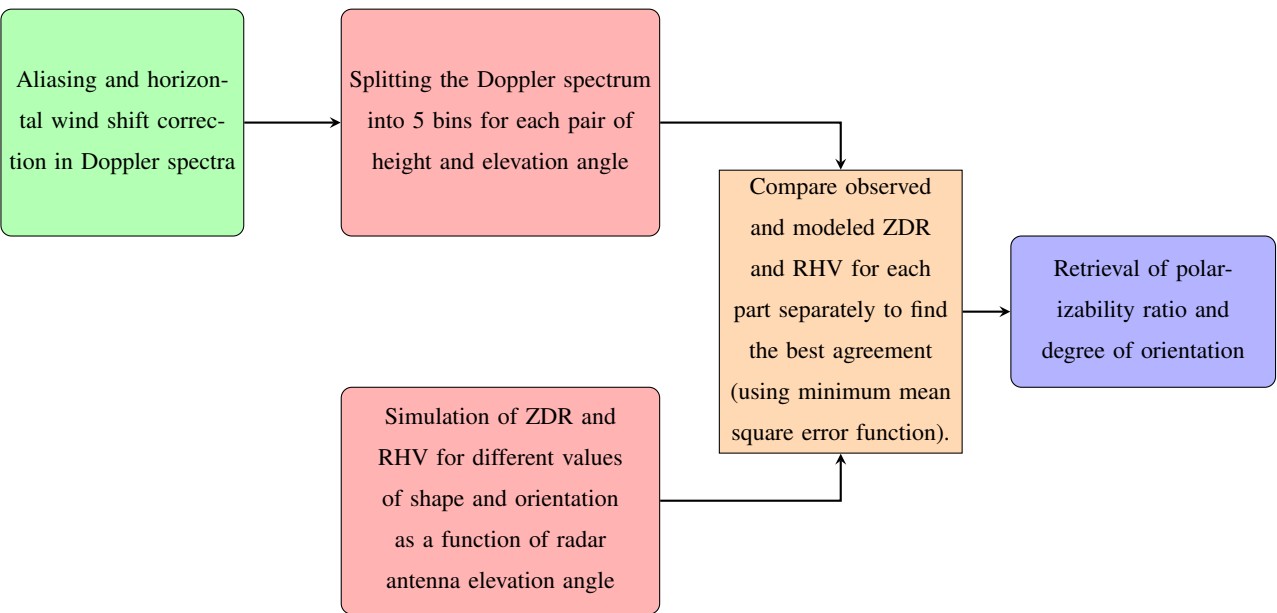

**Figure 2.** Block Diagram illustrating the Spectrally resolved approach.

driven Doppler spectra at the different elevation angles. As depicted in Fig. 1 for a Doppler spectrum observed in zenith-pointing direction, turbulent vertical motions are superimposed onto the gravitational downward motion of particles and result in Doppler spectra which are broadened toward both positive and negative velocities. Horizontal winds also impact the Doppler

spectra, especially for lower elevation angles, by introducing a horizontal velocity component into the downward-pointing component of the falling particles. Because the observation volume of a cloud radar is actually a conic section, a velocity wind 290 shear within the radar observation volume introduces an artificial broadening of the Doppler spectrum, as well.

### 3.3.1 Retrieval of horizontal wind

In order to correct horizontal wind effects on the Doppler spectrum at different elevation angles, a best guess of the horizontal wind vector is needed for each range gate to which the shape retrieval is about to be applied. A common approach to retrieve the profile of the horizontal wind with a scanning radar, is to employ a rotating beam that revolves around the vertical axis 295 and focuses on a specific elevation angle to scan a circular area centered at the origin, which is commonly known as PPI (Plan Position Indicator) scan. Figure 3a visually represents the PPI scan geometry, showcasing the elevation angle $\psi$, azimuth angle $\beta$, wind speed vector $V_h$, wind direction angle $\alpha$, and radial wind velocity $V_w$. This illustration provides a comprehensive depiction of the various components involved in the PPI scan configuration. As the radar emits signals, it captures returns from particles scattered across the entire scanned circle. In this diagram, the wind direction is assumed to be westward. When the 300 radar beam targets point A, the wind direction opposes the radar line of sight, resulting in the highest negative radial wind velocity. Conversely, when the radar beam aligns with point C, the wind moves parallel to the radar line of sight, yielding the highest positive radial wind velocity. At points B and D, where the radar line of sight is perpendicular to the wind direction, the radial wind velocity registers at zero. Hence, the radial wind velocity acquired follows a sine pattern relative to the azimuth angle, termed the Velocity-Azimuth Display (Fig. 3,b). The phase shift of the sine curve serves as an indicator for wind direc- 305 tion, while the sine's curve amplitude yields the wind velocity Vh multiplied by the cosine of the elevation angle. Additionally, the entire curve's displacement from the zero velocity relates to the precipitation fall speed. Eq. 12 represents the radial wind velocity $V_W$ as function of the azimuth angle $\beta$.

$$V_W = -V_h \cos(\beta - \alpha) \cos(\psi) \tag{12}$$

For the retrieval of the horizontal wind components we specifically followed the approach that was introduced by (Baars 310 et al., 2023). They use the traditional analytical technique which searches for the best fit between the observed azimuth dependency of radial velocity and a sine function in order to derive horizontal wind speed and direction.

### 3.3.2 Aliasing problem and effects of horizontal winds on the determination of the vertical velocity component

Radars can ascertain particle speeds up to a limit known as the unambiguous (Nyquist) speed. Aliasing occurs when Doppler velocities surpass the Nyquist velocity. Especially at low elevation angles the contribution of the horizontal wind component 315 can lead to Doppler velocities that surpass the Nyquist range. Additionally, the increasing contribution of the horizontal wind with decreasing elevation angle needs to be considered when the vertical component of the Doppler velocity, e.g., the best guess of the particle vertical velocity, is to be determined for all angles of an RHI scan. To address both the aliasing problem and the contamination of the vertical velocity component by the horizontal wind, the first step involves resolving any existing

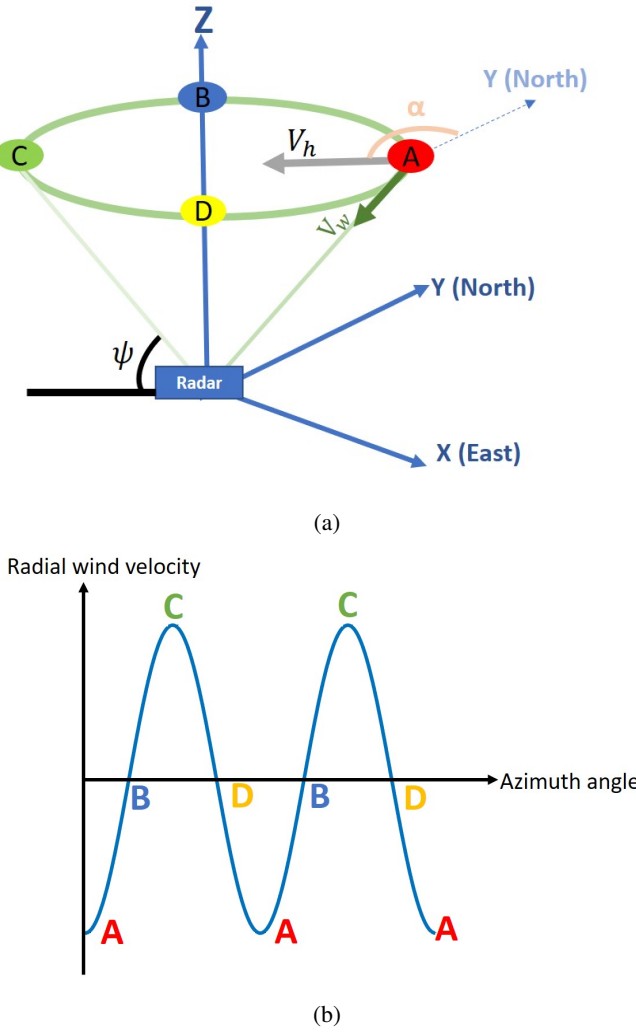

(a)

(b)

**Figure 3.** Representation of the horizontal wind vector by means of a PPI scan, showcasing the elevation angle $\psi$, wind speed vector $V_{\mathrm{h}}$, wind direction angle $\alpha$, and radial wind velocity $V_{\mathrm{W}}$. (a) Illustration of the location of the 4 data points A, B, C, D which are orthogonal and perpendicular to the horizontal wind vector, respectively. (b) Angular relationship between horizontal wind vector and azimuth angle.

aliasing issues within the measured velocity. Subsequently, the correction for the horizontal wind effect in the determination of the vertical velocity component is carried out. These two corrections are detailed in the following subsections, each addressing their respective aspect.

1. Aliasing problem correction

Considering the characteristics of the Mira-35 cloud radar (pulse repetition frequency $f_{\mathrm{n}} = 5000\,\mathrm{s}^{-1}$, $\lambda = 8\,\mathrm{mm}$), the unambiguous speed for this radar $v_{\mathrm{max}}$ is $10.3\ \mathrm{ms}^{-1}$.

Folded velocities occur when particle's radial velocities $V_R$ surpass the Nyquist velocity $v_{max}$. The connection between the measured particle's radial velocity $V_R$, Doppler velocity, and Nyquist velocity is defined by Equation 13:

$$V_D = V_R \pm 2nv_{max} \tag{13}$$

    Where $n$ represents the Nyquist interval number as an integer. Correcting such folded velocities, often termed as unfolding or dealiasing, necessitates determining an appropriate value of $n$ to retrieve the correct velocity. Various correction tech-
niques, ranging from basic to sophisticated algorithms, have been proposed by researchers (Ray and Ziegler, 1977; Jing and Wiener, 1993). When particle's radial velocity surpasses the $v_{max}$, the radar mistakenly deducts $2nv_{max}$ from the true values. Conversely, when the Doppler velocity falls below $-v_{max}$, the radar processing erroneously adds $2nv_{max}$ to the true values, resulting in aliasing or folding issues. To address folding issues in the Doppler spectrum, the critical elevation angle, where the velocity matches the unambiguous velocity, for each specific height is determined. For elevation angles beyond this critical
angle, Doppler spectra need to be shifted by $\pm 2nv_{max}$. In this study, since the wind speed was not very strong, the aliasing issue was consistently resolved by setting n = 1. In this case, folding occurs at low elevation angles (between 30-60 $^\circ$ or 120-150 $^\circ$). However, in the presence of stronger winds, folding begins at elevation angles closer to the zenith (between 60-90 $^\circ$ or 90-120 $^\circ$), requiring the consideration of n > 1. The correction involves a time-profile-by-time-profile examination (Johnson et al., 2020). It should be noted that if there is no aliasing problem, then $n = 0$, and consequently $V_D = V_R$.
2. Horizontal-wind correction

    The velocities recorded in Doppler spectra often include contamination from radial wind velocity. To obtain a best guess of the downward-pointing velocity of hydrometeors, it is crucial to eliminate the radial wind speed component from the measured Doppler velocities. This process, known as horizontal wind correction, helps in retrieving a best guess of the true values of downward-pointing component of the velocity vector at all elevation angles and thus the falling attitude of the observed cloud
hydrometeors. By subtracting the radial wind speed (Equation 12) from the aliasing-corrected radial particle's speed, one can derive corrected values for the particles' radial speed $V_{R,cor}$ (Eq. 14).

$$V_{R,cor} = V_D - V_W \tag{14}$$

## 4   Application and results

    The strength of the main-peak approach to derive the shape and orientation of hydrometeors formed in supercooled stratiform
clouds was effectively proven by Myagkov et al. (2016a, b) and produced valuable results due to the homogeneity of this cloud type. In contrast, deep clouds comprise a variety of hydrometeor types, limiting the main-peak approach to identify only the dominating one, while others remain undetected. In this section, an application of the spectrally resolved approach will be presented by means of different case studies. The novelty of the presented method will be highlighted by comparing the results to the results obtained from the main-peak approach.

## 4.1 First case study 07 Nov 2014, 09:15 – 09:30: retrieval of various hydrometeor types

A complex cloud system that was observed on November 7, 2014, in Cabauw, the Netherlands, is introduced in Fig. 4. Figure 4, is divided into six panels. Panels (a) through (d) present key radar moments measured by the vertically pointing Mira-35 NMRA cloud radar, including radar reflectivity (a), linear depolarization ratio (LDR) (b), Doppler velocity (c), and spectral width (d). These parameters provide insight into the microphysical and dynamical properties of the observed cloud structures. Reflectivity and LDR offer information on particle size and shape, while Doppler velocity and spectral width reflect the vertical motion and turbulence within the cloud. Complementary lidar measurements from the PollyXT system are shown in panels (e) and (f), displaying the attenuated backscatter coefficient at 1064 nm and the volume depolarization ratio at 532 nm, respectively.

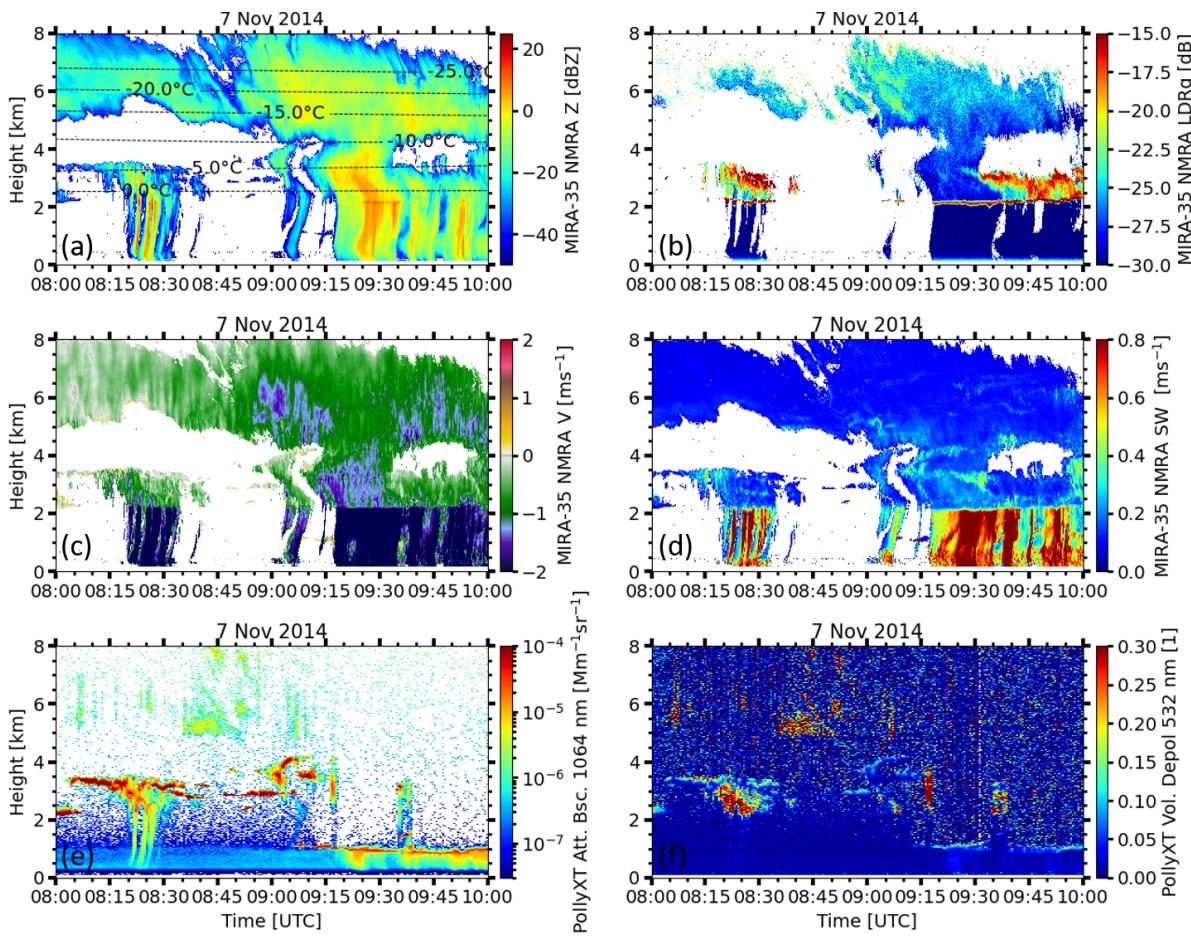

**Figure 4.** (a) Radar reflectivity, (b) LDR, (c) Doppler velocity, (d) spectral width measured by Mira-35 NMRA and (e) attenuated backscattered coefficient, and (f) volume depolarization ratio measured by PollyXT lidar as observed on November 7, 2014. The time interval from 09:15–09:30 UTC and altitude range from 0–8000 m correspond to the case study to which the main-peak shape retrieval method was applied (see Fig. 6). Similarly, the time interval from 09:45–10:00 UTC and altitude range from 0–4000 m correspond to the case study to which the spectral shape retrieval method was employed (see Fig. 7 – Fig. 9)

.

Focus of this case study is set on the time period between 09:15 and 09:30 UTC, when a deep cloud, reaching up to 8 km height was present above Cabauw, which caused slight precipitation already since 09:00 UTC. Ice particles descended, creating a melting layer at approximately 2 km altitude. The evolution of the ice phase in this deep cloud spanned from the melting layer up to its top at around 8 km altitude during the specified time period. At cloud top, the LDR was approximately −25 dB and decreased to around −30 dB close to the melting layer. This leads to the impression that hydrometeors transitioned from a non-spherical (prolate) shape into a more isometric shape. Already at heights from 6.5 km downward, vertical velocities approached values of up to $-1.5\,\mathrm{ms}^{-1}$, indicating the presence of rather large, probably rimed particles. During the time period of interest

for this case study, prominent features in vertical velocity and spectral width occur at heights between slightly below 4 km and the melting layer. Spectral width shows increased values which are indicative for increased in-cloud turbulence. Fall velocities reach values of between 1 and $1.5\,\mathrm{m\,s^{-1}}$. Given in addition the temperature range from approximately $-10$ to $0\,°\mathrm{C}$ in this height range, the co-located presence of liquid water and ice crystals might have led to the occurrence of riming (Vogl et al., 2024). This notion is also corroborated by the measurement of lidar volume depolarization ratio (VLDR). Albeit that the lidar

signal was attenuated during large portions of the observation period, low values of VLDR (and co-located elevated values of attenuated backscatter coefficient) at heights around 4 km at 09:15 suggest the presence of liquid water at this height level. Nevertheless, at times after 09:30 UTC the cloud transitioned from a deep cloud into a rather shallow cloud with a cloud top height of around 3.5 km. The layer containing the ice or mixed phase spans only approximately 1 km of total cloud depth. Spectral width decreases, indicating less variability in vertical velocity. Overall, values of vertical velocity are much lower

compared to the deep-cloud period just before. A short time window between 09:30 and 09:45 allowed the lidar beam to penetrate the atmosphere up to the top of this cloud layer. This window is sufficient to identify a layer of liquid water at the cloud top at around 3.5 km height. Temperatures of around $-5\,°\mathrm{C}$ associated to high values of radar LDR highlight that prolate(columnar) hydrometeors dominated the ice phase during this time period. We will in the following deploy the main-peak and spectrally-resolved shape and orientation retrievals to evaluate the actual morphology of the observed hydrometeors

during the time period under investigation.

    To assess the morphology of the actually dominating ice particle population in this cloud, the main-peak approach is applied first. The main SNR peak for each range and elevation angle pair is considered. These values are presented in the SNR profile shown in Fig 5 a. In this figure, SNR values exceeding 30 dB are prominent at heights between 2 and 4 km. Near the cloud top, SNR values drop significantly, indicating an insufficient number of data points for analysis. This scarcity is evident in the

ZDR and RHV visualizations in Fig 5 b and Fig 5 c. In Fig 5 b, ZDR remains relatively homogeneous around zero above 2 km height. Fig 5 c shows that RHV values remain around 1 at heights above 2 km. The ZDR and RHV profiles above height of 2 km suggest the presence of spherical particles. No evidence of prolate-shaped particles is observed in these profiles.

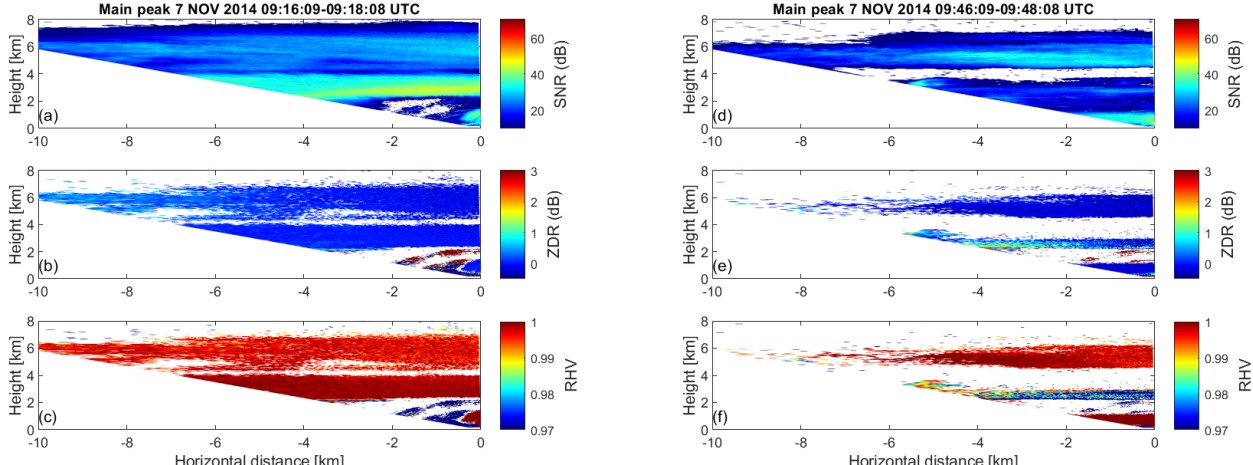

**Figure 5.** Left panel: Height-vs-distance cross-sections of: (a) maximum SNR, and (b) ZDR, and (c) RHV as derived from a maximum SNR of Doppler spectra of RHI scan of Mira-35 MBR4 performed at Cabauw, NL, on November 7, 2014, 09:16–09:18 UTC. Right panel: (d) maximum SNR, and (e) ZDR, and (f) RHV as derived from a maximum SNR of Doppler spectra of RHI scan of Mira-35 MBR4 performed at Cabauw, NL, on November 7, 2014, 09:46–09:48 UTC.

In the left panel of Fig 6, representing the deep cloud during the period from 9:15 to 09:30 UTC parameters relevant for the retrieval of the ice particle shape and orientation by means of the main-peak approach are illustrated. In Fig 6 a, the polarizability ratio at cloud top is $0.79$ and shows a slight increase with decreasing height ($0.83$ at $5\,\text{km}$ height), indicating the presence of rather isometric or slightly oblate particles. Descending to an altitude of $4\,\text{km}$, the determined polarizability ratio remains approximately $1.1$, suggesting the continued existence of a spherical shape.

As shown in Fig. 4, two distinct thin clouds at varying altitudes are observable during the 9:30 to 10:00 UTC period. Precipitation from the upper-level cloud layer with a top height of $7\,\text{km}$ stops, revealing a lower-level cloud layer with a top height of about $3.5\,\text{km}$. This layer exhibits depolarization signatures (as depicted in Fig. 4 b), indicating a shift in the shape of ice particles towards non-spherical. In the deep cloud (09:15 to 09:30 UTC) discussed above, no evidence of depolarization was observed in the moment data at the same altitude range ($2.3 - 3\,\text{km}$), suggesting the presence of isometric particles as it was also confirmed by application of the main-peak shape and orientation retrieval.

In the right panel of Fig 5, depicting the thin cloud during the period from 9:45 to 10:00 UTC, the maximum SNR in panel d, ZDR in panel e, and RHV in panel f are displayed. At heights ranging from 2.3 to $3\,\text{km}$ in Fig 5 b, ZDR at the zenith angle is zero, rising to $1.5\,\text{dB}$ at the lowest (most-tilted) elevation angle ($150\,^{\circ}$). In Fig 5 c, RHV at the zenith angle around $2.3\,\text{km}$ is $0.9$, increasing to $0.97$ as height ascends to $3\,\text{km}$. At these altitudes, RHV increases with decreasing elevation angles, a characteristic signature of prolate particles.

Figures 6 c, and d illustrate the analysis of polarizability ratio and degree of orientation as quantified by the main-peak approach for the thin cloud layer for an RHI scan performed from 09:46 and 09:48 UTC. The derived polarizability ratio and

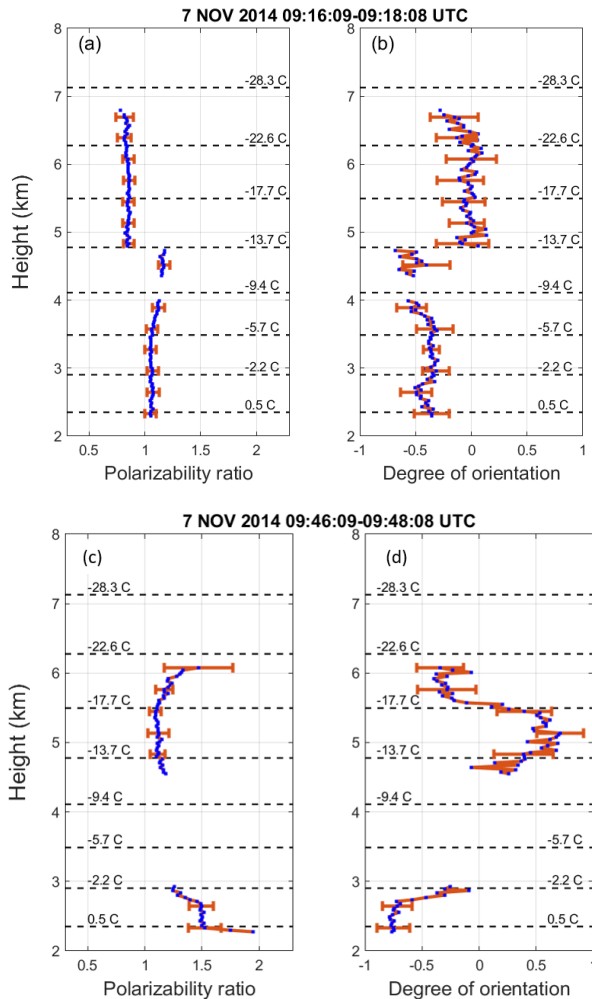

**Figure 6.** (a) Polarizability ratio and (b) degree of orientation values retrieved by the main peak approach from a Mira-35 RHI scan performed at 09:16-–09:18 UTC, and (c) Polarizability ratio and (d) degree of orientation values retrieved by the main peak approach from a Mira-35 RHI scan performed at 09:46–09:48 UTC on November 7, 2014. Orange horizontal bars denote 2 standard deviations of polarizability ratio and degree of orientation, respectively. Dashed horizontal lines display the temperature levels from the GDAS1 model output that is closest to the observational time (9 UTC on 7 November 2014).

degree of orientation for the layer between 2 and 3 km height clearly show the presence of predominantly horizontally aligned prolate ice crystals.

In summary, the main-peak approach effectively identifies the primary hydrometeor types in the deep cloud (spherical particles) and the thin cloud (prolate particles). The occurrence of prolate-shaped particles in the thin cloud shortly after the

formation of the deep cloud suggests the possible coexistence of prolate-shaped particles within the deep cloud at heights ranging from 2.3 to 3 km, although they may not be the prevailing particle morphology. Due to notable changes within the

cloud from 9:30 to 9:45 UTC, the dominant particle shape transitions from spherical to prolate, and this transition is captured by the main-peak approach during the 9:45 to 10:00 UTC period. Consequently, the main peak approach may not accurately identify this particular hydrometeor type in the deep cloud.

Figure 7 presents height-vs-distance cross-sections showing the SNR, ZDR, and RHV across the five selected parts of the Doppler spectra obtained during an RHI scan that was performed from 09:16–09:18 UTC. This visual representation offers an insightful view of these parameters across different parts of the Doppler spectra acquired within the specified time frame and over the range of elevation angles covered by the RHI scan.

The spectrally resolved approach starts with splitting the Doppler spectra at all individual height levels with a vertical
resolution of 30 m. Splitting the spectra into the 5 parts allows a separate analysis of different particle types which might be present in the probed cloud volume. In Fig. 7 a and e, SNR in parts 1 and 5 appears lower compared to other parts, indicating possible contamination of ZDR and RHV data by noise. Managing noise can be challenging in the spectrally resolved approach, when the SNR is too low. This issue is particularly pronounced in the first and final parts of the Doppler spectrum. Figure 7 a reveals low SNR levels above 4 km height, with most data points above 6 km height registering below 20 dB. At higher altitudes,

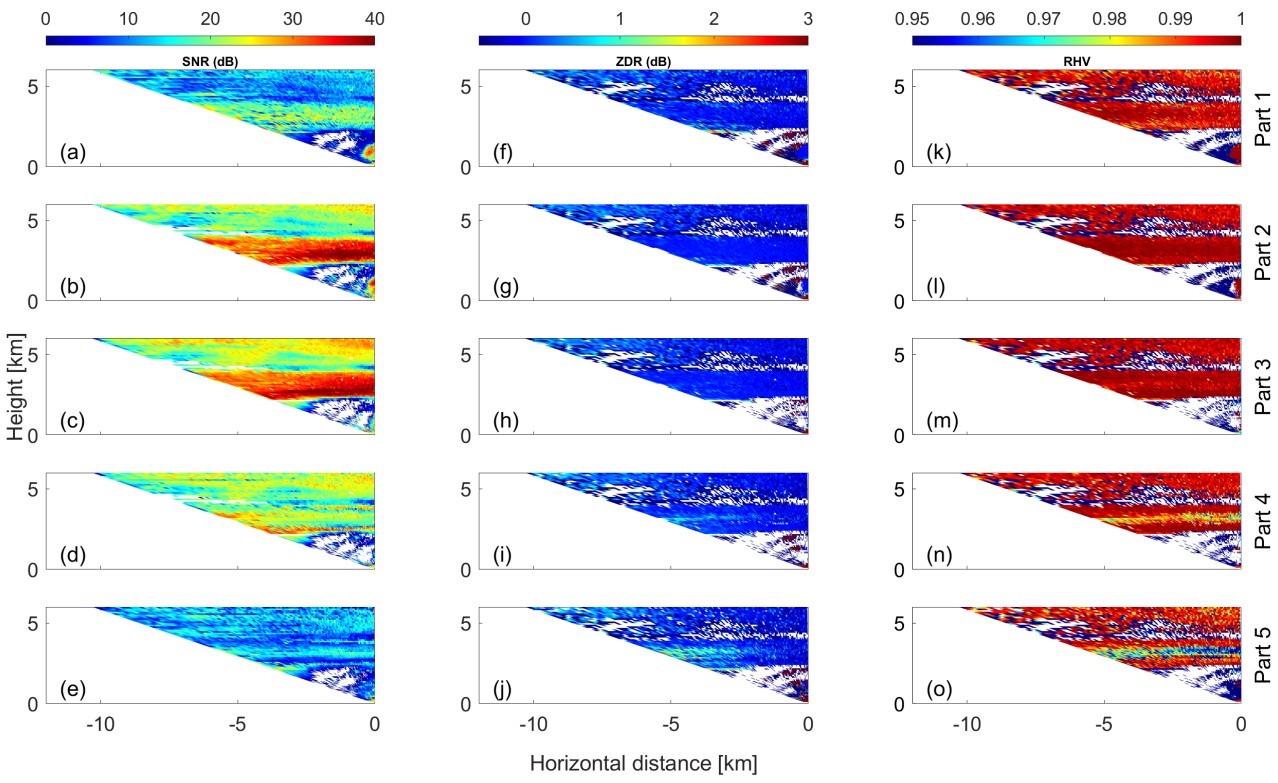

**Figure 7.** Height-vs-distance cross-sections of: (a) SNR, (f) ZDR, and (k) RHV in part 1, (b) SNR, (g) ZDR, and (l) RHV in part 2, (c) SNR, (h) ZDR, and (m) RHV in part 3, (d) SNR, (i) ZDR, and (n) RHV in part 4, (e) SNR, (j) ZDR, and (o) RHV in part 5 of the Doppler spectrum as derived from a RHI scan of Mira-35 MBR4 performed at Cabauw, NL, on November 7, 2014, 09:16–09:18 UTC.

the low SNR in part 1 (representing the fastest falling particles) of the Doppler spectra prevents the complete representation of ZDR and RHV profiles. In Fig. 7 f, between heights of $5.2 - 6.1$ km, the ZDR profile around the vertical angle hovers near $0$ dB. With a decrease in elevation angle, ZDR slightly increases to approximately $0.5$ dB, resembling the main peak of ZDR. Correspondingly, RHV at these heights consistently remains close to 1. These ZDR and RHV signatures suggest the presence of oblate-shaped particles in these altitudes. At heights ranging from the melting layer ($2.3$ km) to $4$ km, the SNR stabilizes at approximately $25$ dB. ZDR variations with decreasing elevation are less pronounced at these heights, while RHV remains close to 1. These characteristics suggest the presence of spherical-shaped particles within this height range. Figure 7 b displays relatively high SNR for part 2 of the analyzed Doppler spectra, implying less noise contamination compared to part 1. However, at certain altitudes between $3.9 - 5.6$ km, low SNR levels hinder discernible signatures in ZDR and RHV profiles, making their retrieval challenging. Conversely, at heights where SNR is high within this range, RHV remains close to 1, indicating a lack of significant variations. In Figure 7 c, higher SNR values provide clearer ZDR and RHV signatures for part 3, as shown in Fig. 7 h and m. Across both higher and lower cloud layers, ZDR remains nearly 0 at all elevation angles, and RHV stays consistently close to 1, indicating spherical-shaped particles in this section. Figures 7 d, i, and n, representing part 4, exhibit signatures consistent with oblate-shaped particles observed at various heights. However, in Figures 7 j and o (part 5), insufficient valid data above $4$ km is evident, and different particle signatures are observable below $4$ km. Below $4$ km, RHV at the zenith angle registers at its lowest value around $0.96$, increasing with decreasing elevation angle. ZDR also exhibits an increase with decreasing elevation angle at these heights, suggesting the presence of prolate particles in part 5 which is representing the slowest-falling particles.

At around $3$ km height, particles are spherical-shaped in parts 1–3, oblate-shaped in part 4, and prolate-shaped in part 5; therefore, this height is chosen for a detailed depiction of the data analysis procedure. Figures 8 a and g showcase complete Doppler spectra of ZDR and RHV at the height of $3.1$ km. Figures 8 -b–f and h–l show the elevation dependency of ZDR and RHV for each of the 5 spectral parts, respectively. In general, homogeneity in ZDR and RHV as a function of elevation angle is given for parts 1 (fastest falling part) to 3, but notable differences are indicated for parts 4 and 5, corresponding to the slow falling particles. These discrepancies are more discernible at higher off-zenith angles, emphasizing the elevation-dependent nature of ZDR and RHV. Figure 8 demonstrates well that splitting the spectra into 5 parts allows for separate analysis of different particles, as will be demonstrated in more detail in the following.

In Figs. 8 -b to d, the ZDR values remain homogeneous across all elevation angles, centered around $0$ dB. Similarly, in Fig 8 h to j, the RHV values exhibit uniformity across elevation angles, consistently equal to 1. These ZDR and RHV signatures collectively suggest the presence of spherical-shaped particles in parts 1 to 3. In Fig. 8 -e, and f the ZDR values at low elevation angles surpass those at the zenith angle. Additionally, in Fig. 8 -k, and l RHV at zenith pointing direction is at its minimum and shows a slight increase toward lower elevation angles. These signatures suggest the potential presence of prolate-shaped particles.

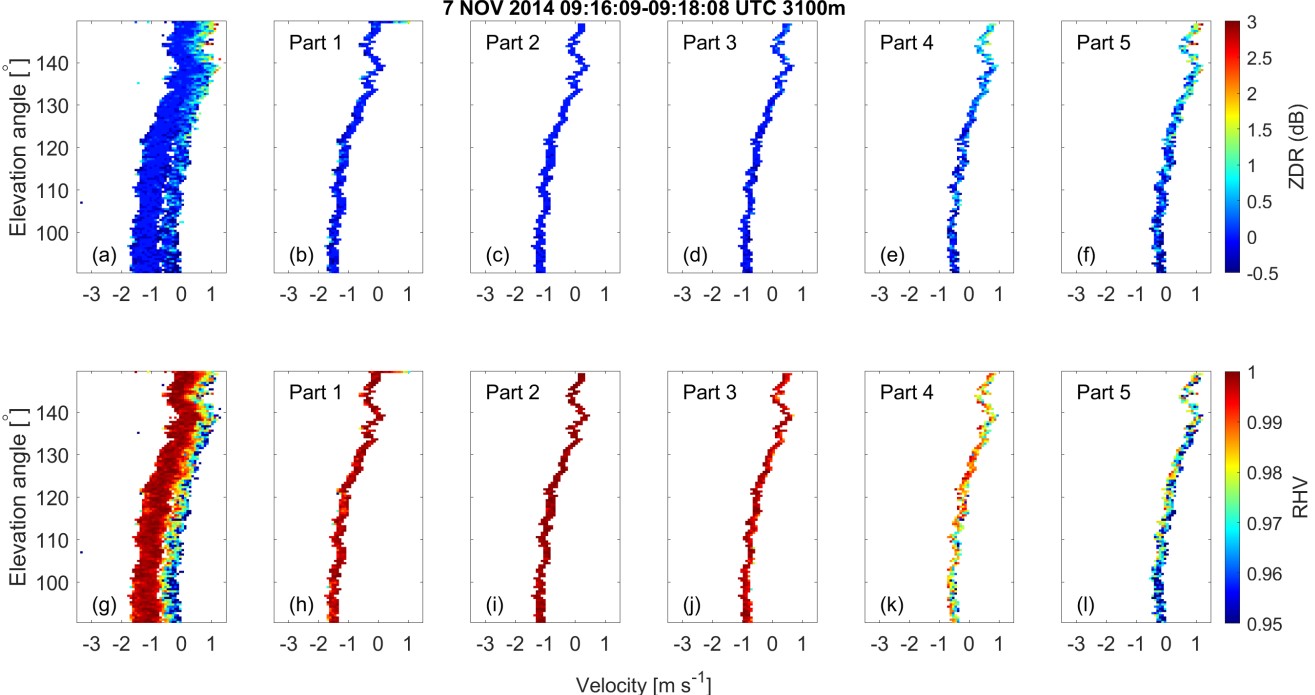

**Figure 8.** Doppler spectra of ZDR and RHV as a function of elevation angle and for each part of the spectrally resolved spectra, shape and orientation retrieval as obtained for 3.1 km height on November 7, 2014, at the time period 09:16:09 – 09:18:08 UTC. (a) ZDR spectrum before segmentation, (b) to (f): ZDR spectrum in parts 1 to 5 after splitting, (g) RHV spectrum before splitting, (h) to (l): RHV spectrum in parts 1 to 5 after splitting.

Figure 9 illustrates the retrieved profiles of polarizability ratio and degree of orientation obtained through the spectrally resolved approach. For the sake of readability, error bars are omitted in this case, as they are in the same order of magnitude as the ones shown in Fig. 6 for the main-peak approach. At an altitude of approximately 6 km, the retrieved polarizability ratio for parts 1 and 2 is 0.75 and 1.25, respectively. This indicates that spherical particles dominate from the cloud top down to an estimated height of 5.5 km. However, there's a data gap between 5.5 and about 4 km height, caused by the too-low SNR values and general lack of data points in this height range. Between a height close to 4 km and down to 2.5 km, the analysis identifies the coexistence of two distinct hydrometeor types. Spherical particles are retrieved from parts 1 (fastest falling part) to 4, indicating their prevalence, while part 5 (slowest falling part) reveals the existence of prolate particles. The prolate shape becomes more evident as height decreases from around 4 km, with the most pronounced prolate shape observed at approximately 3 km, boasting the highest polarizability ratio of around 1.7. As the height decreases further towards the melting layer (height of 2.5 km), the prolate particles transition into a more spherical shape. At an altitude of 3 km, the degree of orientation is at its lowest (around -0.75), suggesting that prolate-shaped particles are more horizontally aligned compared to

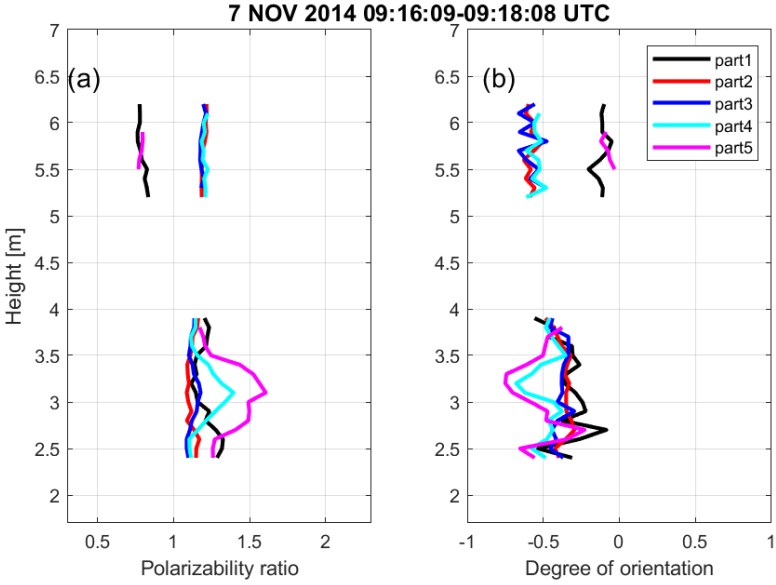

**Figure 9.** (a) Vertical profiles of polarizability ratio and (b) degree of orientation retrieved by the spectrally resolved approach for a RHI scan of Mira-35 MBR4 performed at Cabauw, NL, on November 7, 2014, 09:16–09:18 UTC.

other altitudes. Below 3 km, as the polarizability ratio decreases, the degree of orientation approaches 0, indicating a transition

to randomly oriented prolate-shaped particles.

Based on the presented analysis and the detected co-location of rather spherical particles and prolate particles in the lower part of the cloud system between 09:15 and 09:30 UTC (Fig. 4), relevant conclusions can be drawn. The detected prolate particles were most-likely formed in-situ in the lower layer. Seeding by the upper layer does not seem to have any effects, since prolate ice crystals are also present when no seeding occurred from above (after 09:30 UTC in Fig. 4). This case thus highlights

that the simultaneous presence of particles from a seeder cloud and prolate particles does not necessarily mean that secondary ice formation was triggered by the seeding of particles from higher altitudes into the lower-level cloud layer. This conclusion represents a new aspect for studies of secondary ice formation. In general, it is assumed that the sudden occurrence of prolate particle populations in a seeder-feeder constellation is caused by secondary ice formation processes such as the Hallett-Mossop effect (Billault-Roux et al., 2023). The case study presented in here demonstrates that prolate particles can form independently

of any seeder cloud, likely via heterogeneous freezing, as it is possible even at rather low supercooling temperatures when appropriate ice nucleating particles are present (Hoose and Möhler, 2012).

### 4.2 Second case study 03 Nov 2014, 20:30 – 20:45: Secondary ice formation

The objective of this section is to illustrate how the introduced spectrally resolved approach, in combination with Mira-35 NMRA, can unveil secondary ice production. Based on a secondary ice retrieval from zenith-pointing LDR-mode cloud radar

observations, Li and Moisseev (2020) demonstrated a unique phenomenon: the emergence of a two-layer LDR signature within a singular melting layer. Through thorough analysis of Doppler spectra observations, it was revealed that the upper LDR layer results from the melting of ice needles, possibly formed through the rime-splintering process, whereas the lower layer primarily arises from the melting of background ice particles originating at the cloud top. Additionally, above the two-layer LDR region, an area characterized by relatively high LDR values was observed, suggesting that such elevated LDR values in ice may be ascribed to prolate needle-like particles. Prolate-shaped particles situated above the melting layer, being smaller than other hydrometeor types, descend at a slower rate compared to their larger counterparts, covering a shorter distance within a given timeframe. Consequently, this phenomenon was identified to give rise to the observation of two distinct layers of LDR. Using 2 years of vertical-stare polarimetric cloud radar observations, Li et al. (2021) derived a statistics of the occurrence of columnar ice crystals in co-location to the presence of other cloud hydrometeor types, such as aggregates, graupel or pristine crystals of shapes different to the columnar one. They used these statistics to estimate the intensity of secondary ice formation and eventually derived an ice multiplication factor by comparing the derived secondary ice crystal concentrations to estimates of the concentration of ice nucleating particles. While the studies of Li and Moisseev (2020) and Li et al. (2021) highlight the applicability of vertical-stare cloud radar measurements for identification of columnar ice (presumably secondary ice) within a population of otherwise non-depolarizing ice crystals, their method is not able to infer detailed information about the background population of ice crystals from which the secondary ice formed. Reason is, that oblate and isometric/spherical particles feature similar LDR signatures when observed with a zenith-pointing cloud radar. In the following within this section, the applicability of the spectrally resolved shape and orientation retrieval to identify both, the shape and orientation of the background and of the secondary ice populations, will be presented.

The case study of secondary ice formation in a deep cloud system over Cabauw was conducted for the time interval from 20:37 to 20:47 UTC of 3 November 2014 (Hajipour et al., 2024). In Figure 10, panels a, b, c, and d, showcase time-height cross-sections of radar reflectivity factor, LDR, Doppler velocity, and spectral width measured by the zenith-pointing Mira-35 NMRA. Owing to poor data quality during rainfall, the attenuated backscatter coefficient and volume depolarization ratio retrieved from the lidar measurements are not shown.

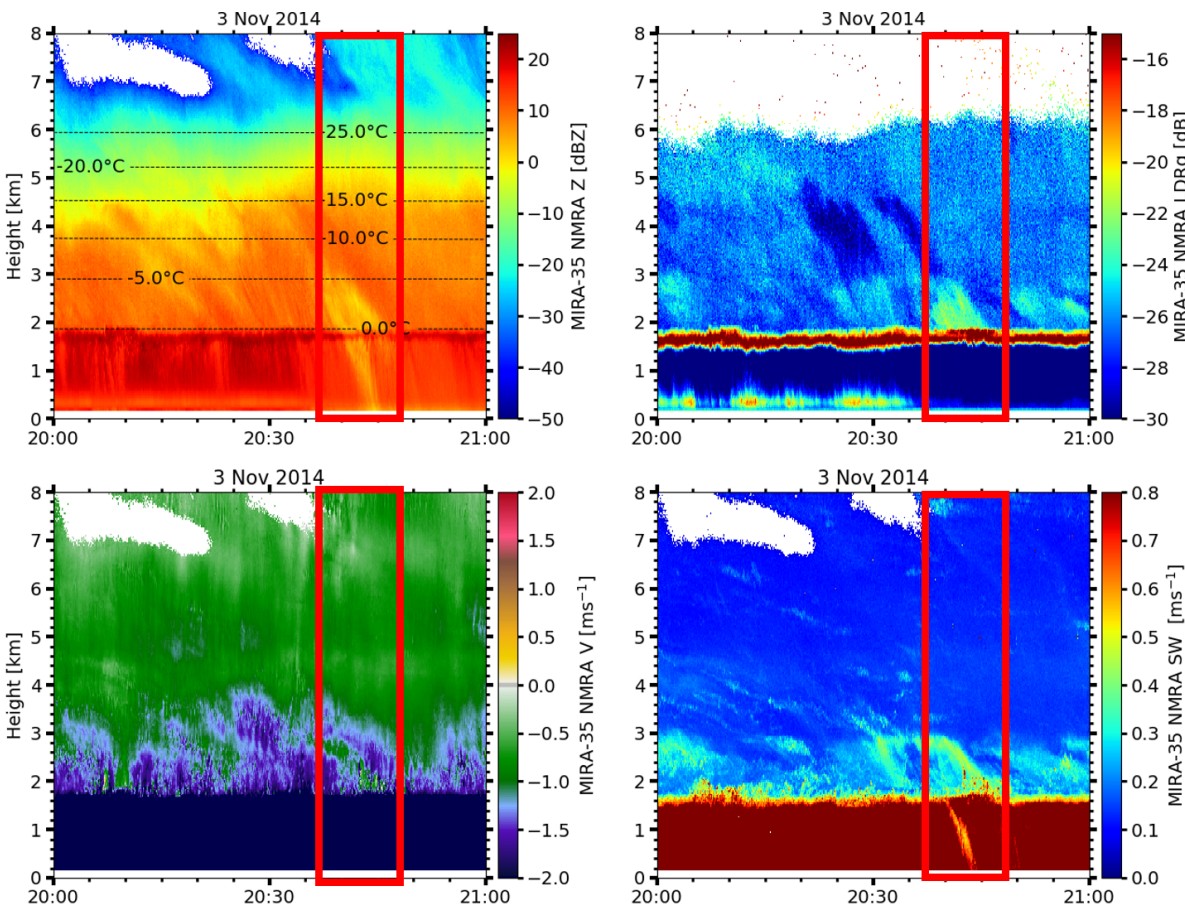

**Figure 10.** (a) Radar reflectivity, (b) LDR, (c) Doppler velocity, (d) spectral width measured by the Mira-35 NMRA. The highlighted period (20:37–20:47 UTC) and altitude range (0–8000 m) signify the focus of the case study where shape retrievals are applied.

Figure 10 provides an overview of the precipitating cloud system observed from 20:00 to 21:00 UTC. Radar reflectivity (panel a) shows strong backscatter values, especially below 4 km, indicating precipitation reaching the surface. The LDR (panel b) increases near the melting layer (2 km), suggesting the presence of melting particles and possibly needle-like ice just above it. Additionally, LDR values around $-25$ dB at the cloud top as observed by vertical-stare Mira-35 NMRA indicate the presence of slightly non-isometric-shaped particles. The Doppler velocity (panel c) reveals strong downward motion, with values exceeding $-1.5\,\mathrm{m\,s^{-1}}$ below 3 km, characteristic of precipitating particles. The spectral width (panel d) is elevated in the lower levels, indicating enhanced turbulence and possibly a mixture of hydrometeor types during active precipitation.

Figure 11 portrays height-vs-distance cross-sections of the SNR (panel a), ZDR (panel b), and RHV (panel c) extracted from the main Doppler spectrum peak during the period 20:41:27–20:43:24 UTC. Above a height level of 6 km, no ZDR or RHV signatures could be inferred because of insufficient SNR. In the uppermost two kilometers of the characterizable cloud layer from 4–6 km height, ZDR shows an increase between zenith direction and increasing off-zenith angles. In middle

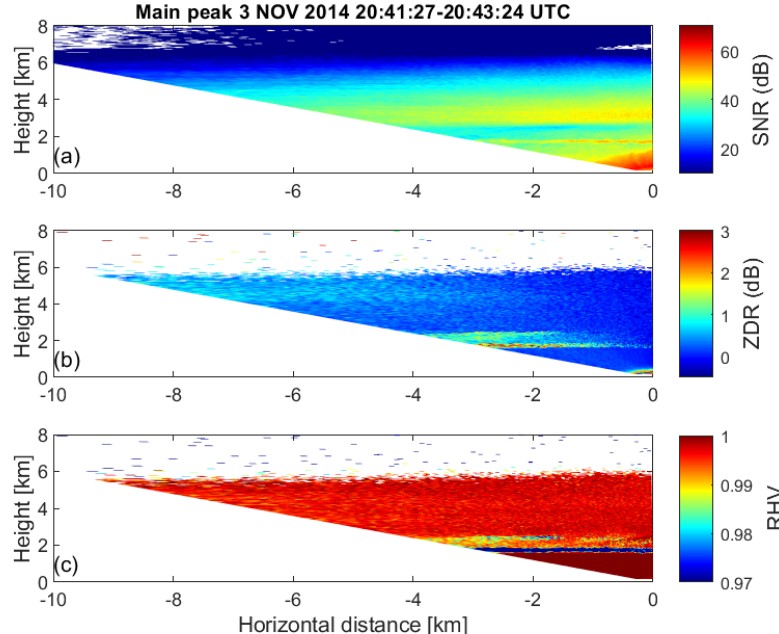

**Figure 11.** Height-vs-distance cross-sections of (a) SNR, (b) ZDR, and (c) RHV of the main Doppler spectrum peak as derived from a RHI scan of STSR-mode Mira-35 performed at Cabauw, NL, on 03 Nov 2014, 20:41–20:43 UTC.

altitudes (3-4 km), ZDR remains consistent across all elevation angles. In lower altitudes (2-3 km), ZDR shows higher values for low elevation angles compared to vertical-stare angles. Throughout the cloud's height, RHV maintains uniformity across all elevation angles at a value of 1 except at lower altitudes. This trend indicates an elevation dependency for both ZDR and RHV at the cloud top and at low altitudes, signifying the presence of oblate-shaped particles. Conversely, the ZDR and RHV signatures at mid-altitude clouds levels represent spherical-shaped (isometric) particles.

Figure 12 a, and b present the profiles of polarizability ratio and degree of orientation, respectively, as obtained using the main-peak approach across four consecutive RHI scans of Mira-35 MBR4 that were performed between 20:30 and 20:45 UTC. In all RHI scans, the polarizability ratio at the cloud top registers at $0.8$, indicating the prevalence of predominantly isometric or slightly oblate particles. Progressing from the cloud top to the height of $4.5$ km, a slight increase in the polarizability ratio is observed in the second RHI scan, reaching $0.9$, indicative of particles with an isometric shape.

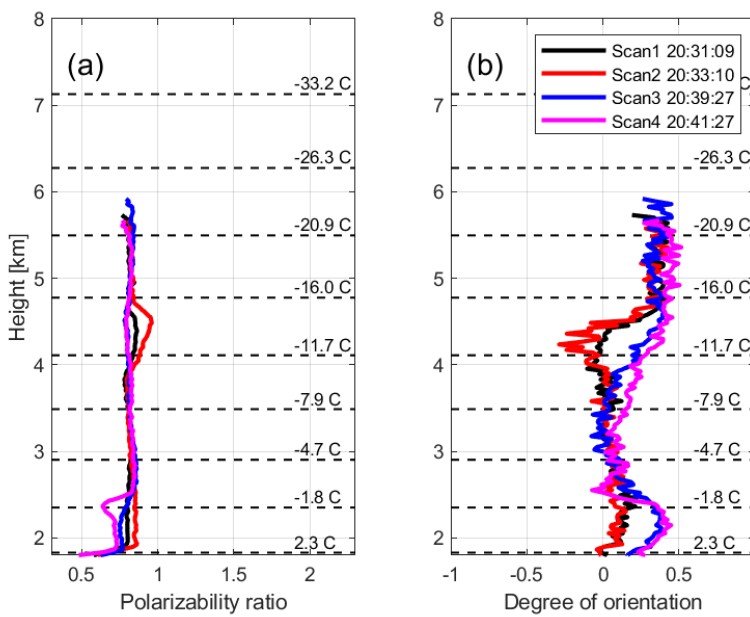

**Figure 12.** (a) Polarizability ratio and (b) degree of orientation values retrieved by the main-peak approach from 4 RHI scans of Mira-35 MBR4 performed at Cabauw, NL, on November 3, 2014, 20:31:09-20:41:27 UTC. Dashed horizontal lines display the temperature levels from the GDAS1 model output that is closest to the observational time (21 UTC on 3 November 2014).

Figure 13 provides a closer examination of the LDR profiles observed between 20:30 and 21:00 UTC, revealing the presence of a secondary layer of elevated LDR values at altitudes between the melting layer and 2.75 km height during the period from 20:40 to 20:47 UTC. As Li et al. (2021) demonstrated, secondary ice production can be identified by analyzing LDR spectra obtained from a vertically pointed cloud radar, such as Mira-35 NMRA. This approach will be evaluated in the following. Figures 14a and b, illustrate profiles of radar reflectivity and LDR spectra acquired from Mira-35 NMRA for the 10-s time

period ending at 20:41:37 UTC. Elevated radar reflectivity levels at 2.9 km and within the lower velocity range of the Doppler spectrum suggest the presence of different hydrometeor types. Additionally, a significant increase in LDR values, reaching about $-17$ dB, was observed from above the melting layer to 2.9 km in the slow-falling velocity range, indicating the dominance of prolate-shaped ice particles in this part of the spectrum. To delve deeper into the analysis and retrieve hydrometeor types also from the fast-falling mode with more detail, the spectrally resolved shape and orientation approach was applied specifically to

the RHI scan recorded during the period from 20:41:27 to 20:43:24 UTC.

    Figure 15 presents height-vs-distance cross-sections displaying SNR, ZDR, and RHV for parts 1 (fastest falling) to 5 (slowest falling) extracted from the selected RHI scan. For parts 1 to 4, the signatures of ZDR are rather similar at all heights above the melting layer (i.e., above 2 km height). Only in the cloud-top region between 4 and 5 km height, the gradient of ZDR vs. elevation is increasing from parts 1 to 4. RHV shows enhanced fluctuations between the different spectral parts 1 to 4. Part

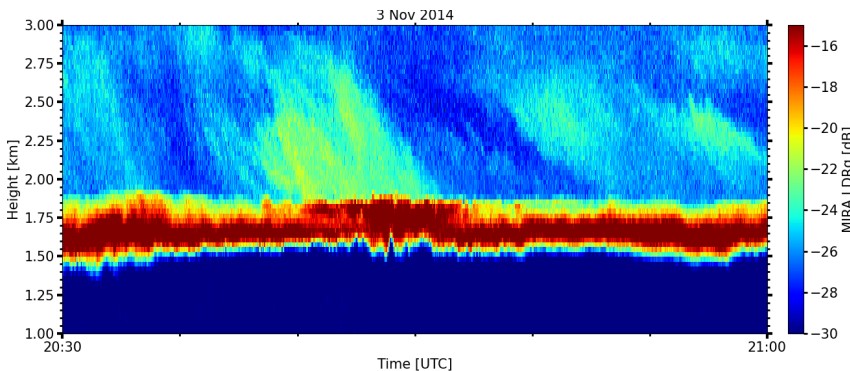

**Figure 13.** Time-height cross section of LDR observed by Mira-35 NMRA during the time interval from 20:30 to 21:00 UTC on 3 November, 2014, in Cabauw, the Netherlands. Elevated LDR signatures were detected at altitudes from above the melting layer and 2.75 km height during the time period from 20:40-20:47 UTC.

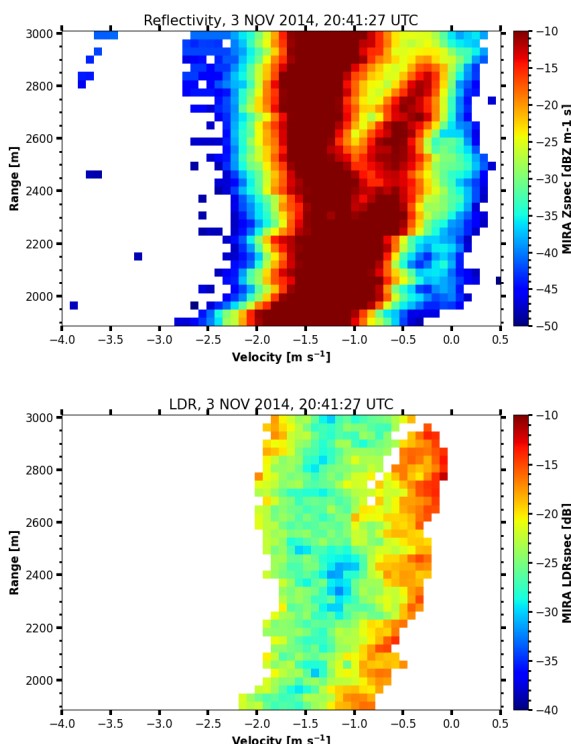

**Figure 14.** Profiles of Doppler spectra of (a) radar reflectivity and (b) LDR at 20:41:27 UTC, on 3 November 2014 measured by Mira-35 NMRA in Cabauw, the Netherlands.

1 features rather noisy RHV signatures at all heights and elevation angles, which are on average slightly below 1. For parts 2 to 4, RHV is approximately constant at all elevation angles at heights above 3 km. At heights below 3 km and above the melting layer, RHV decreases with increasing part number (i.e., with slower-falling Doppler velocities). Both, ZDR and RHV of part 5 show some deviations from the general behavior of the other parts. First, at all heights, ZDR increases with increasing off-zenith angle. Second, RHV from between the melting layer and about 4 km height is constantly low for all elevation angles.

When interpreting the reported signatures of ZDR and RHV, SNR should be considered, as well. In regions of the cloud layer where SNR was below 20 dB, noise artifacts could lead to a reduction of the average values and an increase of variability of RHV. This can potentially be an issue for heights above 3 km of part 1 or for some height regions around 4 km of part 5.

What remains despite the low-SNR regions that occurred at some heights and spectral parts of the case study, is the general feature in ZDR and RHV within the first km above the melting layer. Exemplary for this region, the velocity-vs-elevation

displays for the polarimetric parameters for the height level of 2200 m are shown in Fig. 16.

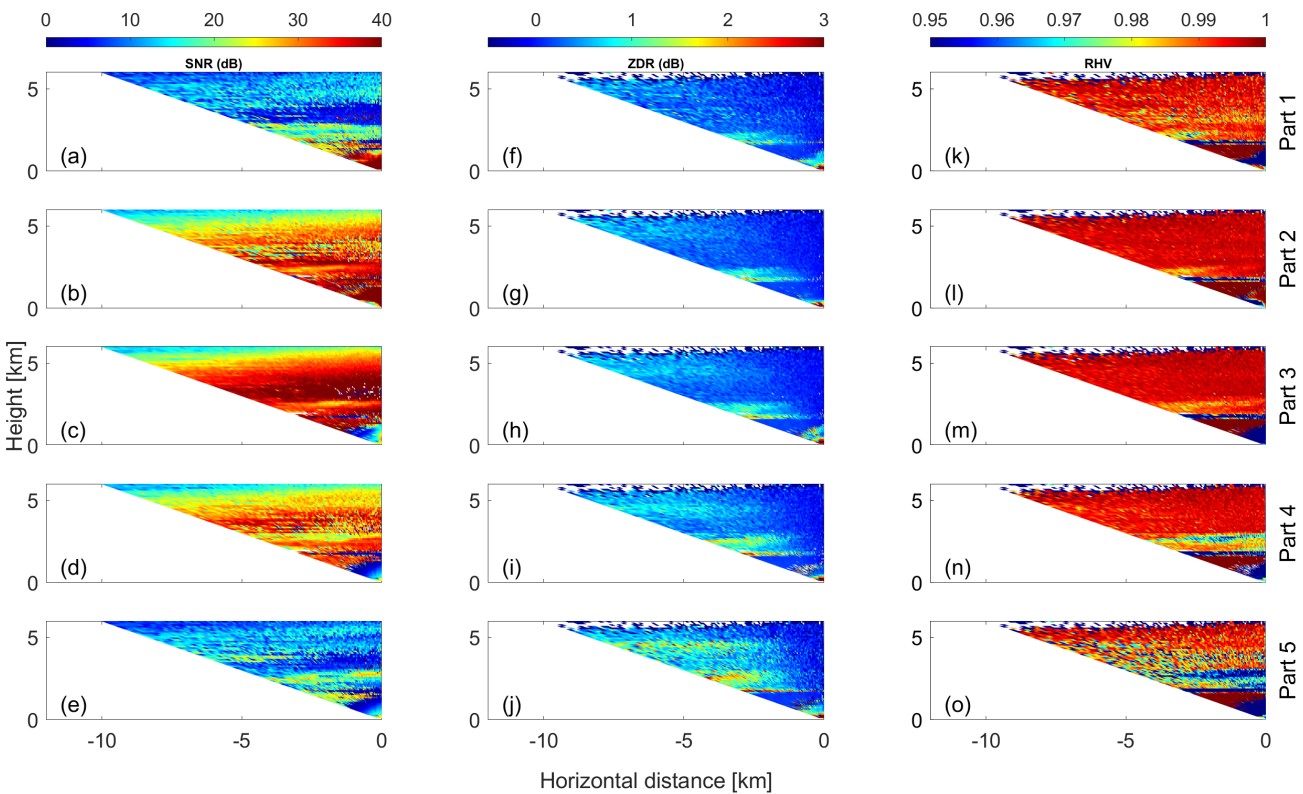

**Figure 15.** Height-vs-distance cross-sections of: (a) SNR, (f) ZDR, and (k) RHV in part 1, (b) SNR, (g) ZDR, and (l) RHV in part 2, (c) SNR, (h) ZDR, and (m) RHV in part 3, (d) SNR, (i) ZDR, and (n) RHV in part 4, (e) SNR, (j) ZDR, and (o) RHV in part 5 of the Doppler spectrum as derived from a RHI scan of Mira-35 MBR4 performed at Cabauw, NL, on November 3, 2014, 20:41–20:43 UTC.

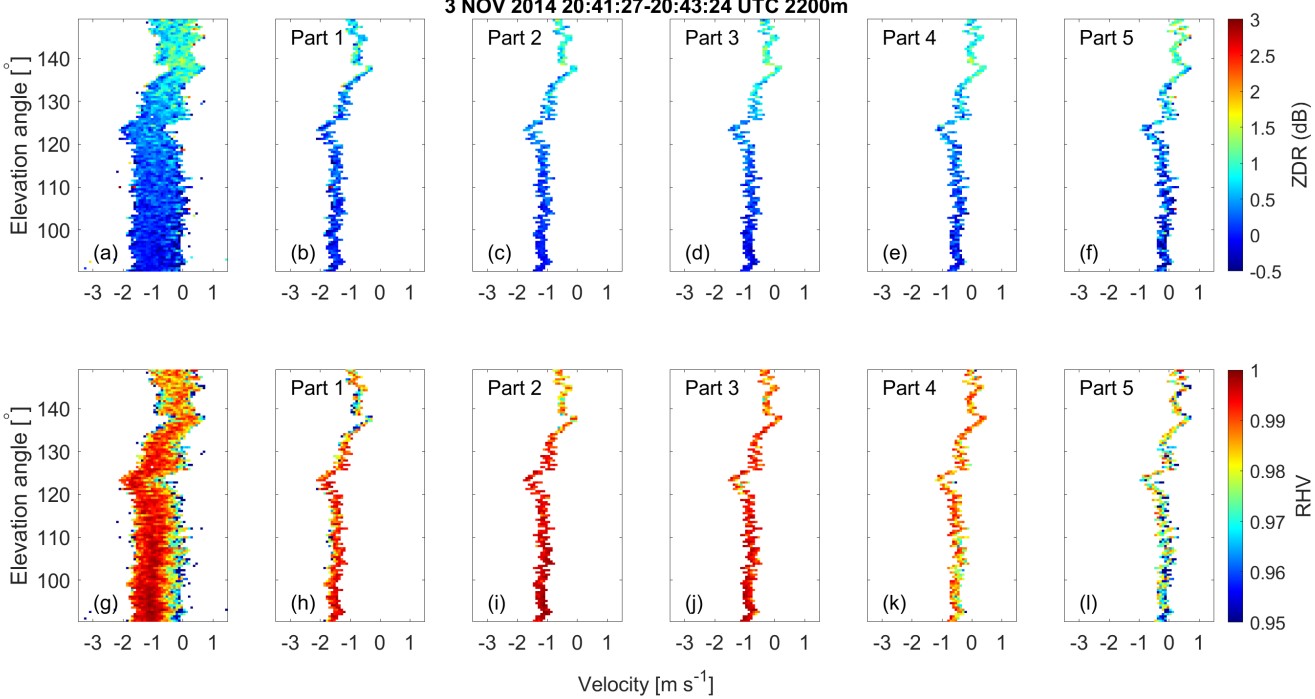

**Figure 16.** Doppler spectra of ZDR and RHV as a function of elevation angle and for each part of the spectrally resolved spectra, shape and orientation retrieval as obtained at 2.2 km height. (a) ZDR spectrum before splitting, (b) to (f): ZDR spectrum in parts 1 to 5 after splitting, (g) RHV spectrum before splitting, (h) to (l): RHV spectrum in parts 1 to 5 after splitting.

Representative for the height levels above 2.5 km, Fig. 17 provides similar illustrations for the height level of 3200 m. By comparing the layers of 2200 and 3200 m, the increased elevation dependency of ZDR and RHV at 2200 m height becomes visible. At 2200 m height, most parts feature oblate-like signatures which are stronger than the ones at 3200 m height. In addition, due to the reverted elevation dependency of RHV in combination with the general elevation dependency of ZDR, part 5 at 2200 m height is indicative of prolate particles.

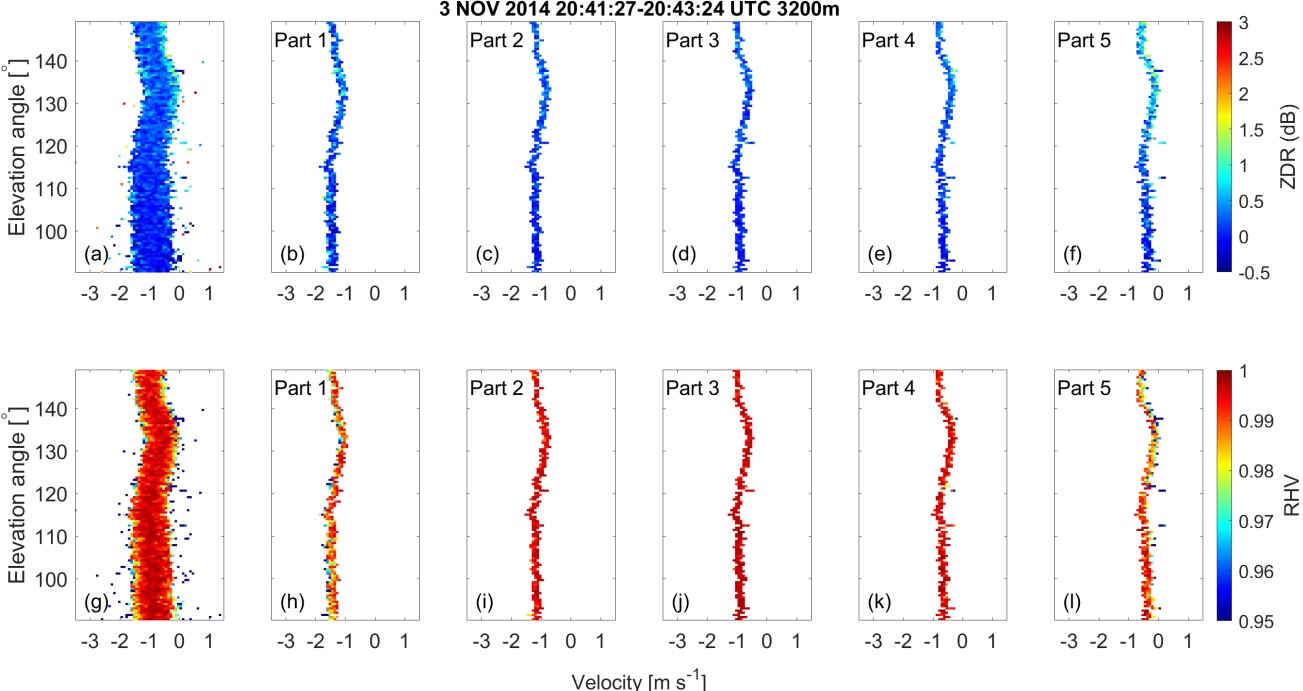

**Figure 17.** Doppler spectra of ZDR and RHV as a function of elevation angle and for each part of the spectrally resolved spectra, shape and orientation retrieval as obtained at 3.2 km height. (a) ZDR spectrum before splitting, (b) to (f): ZDR spectrum in parts 1 to 5 after splitting, (g) RHV spectrum before splitting, (h) to (l): RHV spectrum in parts 1 to 5 after splitting.

Quantitatively, this contrast in particle shape structure is illustrated in Fig. 18, which portrays the derived profiles of the polarizability ratio and degree of orientation obtained using the spectrally resolved approach. Parts 2, 3, and 4 reveal the presence of slightly oblate-shaped particles, with part 4 indicating a more pronounced oblate shape at heights below 3 km and the melting layer, compared to parts 2 and 3. In part 1, the retrieval process tends toward the identification of ice particles with shapes closely resembling spheres, which tend to exhibit oblate characteristics. In part 5, the retrieval method distinguishes prolate-shaped particles above the melting layer and below an altitude of 3 km. Simultaneously, it identifies oblate-shaped particles at altitudes exceeding 3 km.

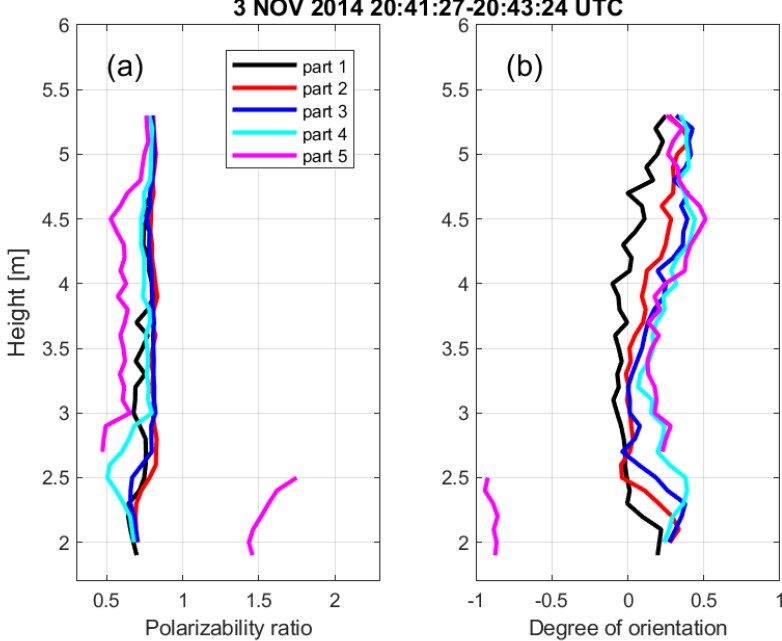

**Figure 18.** (a) Polarizability ratio and (b) degree of orientation values retrieved by spectrally resolved approach from a RHI scan of Mira-35 MBR4 at 20:41–20:43 UTC performed at Cabauw, NL, on November 3, 2014.

In conclusion, the spectrally resolved approach demonstrates the coexistence of various particle types, possibly different in size, within this case study. The main particle type is of oblate shape, possibly representative for dendrites or plates. This crystal type is usually forming at temperatures between $-15$ and $-20\,°C$ (Bailey and Hallett, 2009). Additionally, it suggests the possibility of secondary ice formation that takes place at a height of around 3 km, as indicated by the sudden appearance of prolate-indicating values of the polarizability ratio greater than 1.5 of the slowest-falling Doppler part (part 5, see Fig. 18). In contrast, the original main-peak approach, focusing on the strongest SNR, suggests the presence of only isometric particles. In contrast to the previous case study (Sec. 4.1), where primary ice formation was identified, the occurrence of secondary ice formation in this case is a more-likely explanation for the identified co-location of oblate and columnar ice. Reason is that the time-height cross-sections show only patchy occurrences of LDR structures as well as alternating mean Doppler velocities at heights below approximately 3 km. This suggests a relationship of the hydrometeor formation below this height with the impact of fall streaks (virgae) from higher cloud levels. Primary ice formation would be expected to take place more homogeneously, as was observed in case study 1. Nevertheless, occurrence of primary ice formation similar to what was observed in the first case study cannot fully be excluded, since it was not possible to investigate any potentially existing supercooled liquid layer independently without presence of seeding from higher levels. Based on the presented case study of secondary ice formation it is nevertheless demonstrated that the spectrally resolved shape retrieval provides information about the particle shape distribution that is superior to the information content that can be obtained from zenith-pointing LDR-

mode cloud radar observations. Vertical-stare observations are well suited to identify secondary, prolate ice particles based on their high-LDR signatures. However, for low-LDR values, the vertical-stare observations cannot be used to distinguish between isometric/spherical shapes and oblate shapes. Using the spectrally resolved approach, it was possible to identify that the prolate, secondary ice particles were formed from slightly to moderately oblate ice crystals. Thus, in the presented case, the reason for the production of the secondary ice does not necessarily need to be rime splintering. Indications are given that the branches of oblate ice crystals, such as dendrites fell off, in addition, which then provided the nuclei for the growth of the secondary columnar ice crystals.

## 5  Summary & Conclusions

The significance of mixed-phase clouds in mid-latitude precipitation underscores the need for comprehensive study due to their complex microphysical structure. This study was dedicated to investigating the shape of ice particles within such clouds. We presented an algorithm, the spectrally resolved shape and orientation retrieval, which builds upon the existing main-peak approach to retrieve multiple hydrometer types in the same volume of air. As the main-peak approach was demonstrated to effectively identify the shape and orientation of primary hydrometer types in stratiform clouds, its limitations are evident when dealing with the diverse range of hydrometeor types characterized by variations in size and Doppler velocity. In the spectrally resolved approach, the Doppler spectrum is divided into five parts, and for each part, the differential reflectivity and correlation coefficient are averaged and compared with modeled values. The result yields five pairs of values of polarizability ratio, indicating shape, and degree of orientation, indicating the orientation, of ice particles.

Instead of interpreting the Doppler spectrum segmentation as a strict classification of hydrometeor types, we emphasize that our approach provides a decomposition of the spectrum into distinct Doppler regimes. These regimes reflect variations in particle fall velocities and, by extension, microphysical characteristics such as size, shape, or phase. This composite view allows us to capture the internal variability of the cloud without assigning specific hydrometeor classes to each spectral part. Such a representation aligns with previous studies (e.g., Shupe et al., 2004; Kollias et al., 2007), which observed that Doppler spectra often exhibit multiple peaks or broadened components due to the coexistence of diverse hydrometeor populations. By dividing the spectrum into five parts, respectively regimes, we aim to resolve this complexity in a practical and interpretable way, facilitating further analysis of cloud microphysics and dynamics.

Building upon the main-peak approach, the new technique also integrates the effects of Rayleigh scattering and assumes horizontal homogeneity. This is important for accurately modeling the scattering behavior in scenarios where the signal's wavelength is large relative to the particles being analyzed, as it provides a consistent framework for approximating scattering properties, which is important for interpreting radar measurements. Care must be taken when the observed particles are large and reach sizes which are in the range of the radar wavelength. Then Mie-scattering effects might introduce biases to the polarimetric radar observables. The actual impact of non-Rayleigh scattering effects on bulk polarimetric properties is however under debate at the moment. Matrosov (2021) reports only a modest influence of non-Rayleigh scattering on polarimetric variables. The author notes that the possible reason for the smaller influence of non-Rayleigh scattering on polarimetric variables is

that they are differential (rather than absolute) quantities representing differences/ratios of radar parameters at two orthogonal polarizations. Similar arguments were elaborated on by Teisseire et al. (2024). The separation of the retrieval into different parts of the Doppler spectrum also enables one to exclude the fast-falling parts from an analysis, as the fastest falling particles might be the largest ones, most prone to non-Rayleigh scattering. This approach was not applied yet in the presented study but shall be deployed in follow-up studies. A recent study by (Mak and Unal, 2025) suggests to use spectral differential phase for identification of conditions of non-Rayleigh scattering conditions. As the cloud radar system under study in STSR mode can in principle also provide differential phase, this approach might be evaluated further in follow-up studies.

Besides possible non-Rayleigh scattering effects, the spectrally-resolved shape and orientation retrieval technique assumes horizontal homogeneity, which means it considers the environment to be spatially uniform in the horizontal direction. This assumption simplifies the model by reducing the complexity associated with variations in the medium along the horizontal plane, allowing for more efficient calculations without sacrificing significant accuracy. In the presented study, the homogeneity was evaluated based on (1) inspection of the required monotonic relationships between polarimetric parameters and elevation angle, and (2) the appropriateness of the horizontal wind correction. Additionally, variations in Doppler spectrum width were recognized as a potential indicator of inhomogeneity, though they were not the primary focus in the current evaluation.

One of the aims of the ACCEPT campaign, conducted in Cabauw, the Netherlands in 2014, was to assess the capabilities of both the main peak approach and the spectrally resolved approach. Ensuring the reliability of Doppler spectra for analysis is crucial, necessitating the resolution of issues arising from horizontal wind, particularly the folding problem and the horizontal shift of the Doppler spectrum. Our study, through a first case study, demonstrates how the spectrally resolved approach successfully retrieves multiple hydrometeor types, encompassing both oblate and prolate shapes. Furthermore, our research highlights the new approach's effectiveness in identifying secondary ice production in a second case study. Nevertheless, this study provides also evidence that secondary ice formation or ice multiplication processes do not necessarily need to be active when multiple hydrometeor shapes are present within one cloud volume. As the application of the spectrally-resolved shape retrieval demonstrated, prolate particles can also form independent of any seeding by particles from higher/colder levels.

The extended shape and orientation retrieval has the potential to provide information for higher-level retrievals of cloud microphysical properties and for empirical studies of mixed-phase clouds. State-of-the-art cloud microphysical retrievals, such as the one of Bühl et al. (2019) for ice crystal number size distribution rely on the provision of accurate information about the shape of the investigated hydrometeors. Other retrievals, such as DARDAR (Delanoë and Hogan, 2010) or CAPTIVATE (Mason et al., 2023) rely on the presence of only one hydrometeor type in order to provide plausible results. Shape discrimination techniques which can especially also discriminate between isometric and oblate particles, which is hard to be achieved based on vertical-stare observations, are key to constrain these retrievals. In future, the retrievals might be refined in order to obtain precise microphysical properties (number, size, water content) separately for each identified shape class. An additional valuable future extension would be to implement techniques for tracking of the detected hydrometeor types over a series of range gates, i.e., as a function of height. The more insights about the evolution of the hydrometeors and their interaction can be inferred, similar to the approach of Teisseire et al. (2024). Last but not least, the availability of polarizability ratio and degree of orientation as quantitative cloud parameters is helpful, since the same parameters can also be obtained from numerical model

simulations of clouds, as was previously demonstrated by Welss et al. (2024). In future, simulations of the distribution of the polarizability ratio in a cloud-resolved model will become comparable against their observed counterparts.

*Data availability.* The vertical-stare observations from the remote sensing instruments are available via https://cloudnet.fmi.fi/site/cabauw.
All data which were used for processing and interpretation in the framework of this study can be found at https://doi.org/10.5281/zenodo.14178517 (Hajipour et al., 2024)

.

*Author contributions.* MH developed the spectrally-resolved retrieval and authored the majority of the manuscript. PS supervised the retrieval development and contributed to the manuscript writing. HG, KO, AT, MR contributed technical components and datasets to the retrieval and
added passages to the manuscript.

*Competing interests.* The contact author has declared that none of the authors has any competing interests.

*Acknowledgements.* We are thankful to Alexander Myagkov for the provision of the original main-peak retrieval and to the team of the CESAR observatory in Cabauw and Delft University of Technology, NL, for their support during the ACCEPT field campaign. Acknowledgements are also dedicated to Metek GmbH, Elmshorn, for supporting the preparation and implementation of the Mira-35 MBR4 cloud
radar system.

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
