# Peer review of "Identification of multiple co-located hydrometeor types in Doppler spectra from scanning polarimetric cloud radar observations"

_Atmospheric Measurement Techniques, 2024_

## Referee Comment (RC1)

**Review of Identification of multiple co-located hydrometeor types in Doppler Spectra from scaning polarimetric cloud radar observations**

The paper introduces a new shape retrieval of ice particles using the Doppler Spectra of ZDR and RhoHV and a spheroidal scattering model. This retrieval is based on the retrieval of Myagkov et al. 2016 and is expanding their approach by considering the full spectra as opposed to the main peak of the spectrum. This enables them to retrieve the size of up to 5 different species, potentially giving a possibility of investigating secondary ice production in more details.

I am a bit disappointed about the poor quality of the paper, especially in terms of understandability of the method, and readability of the text. Also, while I agree that the retrieval is a valuable extension of Myagkov et al. 2016, I would have liked to see further improvements to the method (such as moving away from the spheroidal scattering assumptions). In my opinion, this would be an essential next step, in order to reduce the possibly large biases introduced by the spheroidal method, especially when considering aggregates. Nevertheless, I suggest the manuscript to be reconsidered for publication in AMT after the following major and minor comments have been addressed.

Major:

1. the text needs a lot of rewriting, both structurally and for a better understanding of a few sentences (which I will specify in the specific comments). In my opinion, the new method is not explained satisfactorily. I am missing a detailed explanation on why you chose to divide the spectrum into 5 parts, and how you are dividing the spectrum, which has not been mentioned anywhere. I do not know which 5 species/shapes you want to cover with that, I can only think of rimed particles, aggregates, plate-like crystals, columnar crystals. Of course you can also have super-cooled liquid water, but for that we know the shape very well, so no retrieval is necessary. I would also like to know if you are trying to "track" the different species throughout the different heights, or if the division of the spectrum is just random. Also, please explain the main peak method better, for me it was difficult to understand that. Other more "trivial" retrievals such as the vertical wind velocity are explained in great detail, even though it is not that relevant to the study and many institutions are running wind retrievals on an automatically basis. I would suggest to rewrite the paper the following way:

I like Figure 6 and 7, perhaps you can move that to the method section? Then it might be clearer how the retrieval works. If you include the full RHI scan in Figure 6 (without the separation into the 5 parts) and 7, you can first explain the main peak method and then the spectral approach in a clearer way. I would first explain in detail which peak is used in the main peak approach, indicate that in the figures and then continue on to your new approach and show how that is different and how you are dividing into 5 parts. How are you then using the 5 parts? Are you averaging along the Doppler velocity of each part? If so, it might be helpful to provide the averaged elevation dependencies of ZDR and RhoHV either in addition to the Doppler spectra parts or instead of them.
Also you should specify what the polarisability ratio and the degree of orientation are and provide the formulas. This has of course been mentioned in Myagkov et al. 2016, however, these are not standard variables and I think it is therefore necessary to explain that again. You can even use panel a and c of their figure 13 to show the polarisability ratio and orientation. When you are explaining polarisability ratio you can also mention that the shape retrieval can be used with sZDR and sSLDR.

2. I am missing in this paper a few clear statements about the problems of the method. i.e. you should critically discuss the scattering properties that you are using. You did not specify which spheroidal

method you are using, in Myagkov et al. the retrieval is based on Rayleigh. It is well known that spheroidal methods have issues of representing the scattering of ice particles accurately, especially in the Mie-region (which is reached at Ka-Band for aggregates) and for low density ice particles such as aggregates. I understand that generating a large DDA database which is necessary to do such a retrieval is difficult. However, I know that other working groups have done that and you could collaborate with them. I think that especially now that you want to retrieve the shape of multiple species a different scattering method should be taken into account. In my opinion it would have been beneficial to collaborate with other groups who have sophisticated scattering databases based on e.g. DDA. Another thing to discuss: can you be sure that along all elevation angles the species in one height doesn't change? For larger heights this spans quite a large cloud area!

3. Along with comment 3: aggregates have a small polarimetric signature, especially if you are assuming a spheroidal scattering approximation. How much more insight do you even gain when you are considering the spectral lines with aggregates? I am expecting that similar to rimed particles they will just have a polarizability ratio of close to 1. Can you even distinguish between these different particles then? If you can only distinguish between them because they have a different fall velocity is it even worth splitting the Doppler spectra into 5 parts? Would it not be beneficial to just separate between ice crystals and aggregates? I would suggest to show the theoretical polarizability ratio you would expect for all your 5 different species (which I am assuming to be rimed particles, aggregates, dendritic ice crystals and columnar ice crystals and something else). This would show that a separation into 5 parts is actually necessary and that not i.e. 2 parts would suffice.

4. Why do you need to retrieve the actual air motion? Do you use the Doppler velocity for anything and therefore a knowledge of the fall velocity of the particles are valuable? Otherwise you could just move the Doppler spectrum to 0m/s. This would save a lot of effort and space in the paper.

Minor:

1. discuss your figures better/more in the text. Especially Fig 4 and Fig. 9 are not discussed enough. Most readers are not familiar with all the Radar and Lidar variables and can not draw their own conclusions.

2. In this regard: why do you show the Lidar measurements? Due to the rain the Lidar doesn't even penetrate into the relevant cloud regions, so you can just omit the plots.

3. You need to refine your naming in the equations better. For example, before equation 1 you say that $E_h$ is the horizontally polarized plane of the received wave. You never specified what E dot means. Then in equation 4 you say that $E_h$ is the copolar element of the backscattering matrix. Please be specific with your definitions, and stick with one! (also co and cross-polar are not the same thing as horizontal and vertically polarized)

4. In your conclusions I am missing a clear outlook and a discussion of the method and its caveats.

5. The colormap in your Figures looks like jet, perhaps you could consider moving towards a colormap which does not have so many deficiencies, if you prefer rainbow you could use e.g. turbo or something similar.

6. why are you only analyzing the first time period of 7.11.2014 with your spectral retrieval? The second time period would have been interesting, since there the polarimetric variables are large!

Specific comments:

Line 47-48: you say peak signal of the Doppler spectrum, could you specify which variable you mean? Spectral reflectivity? Spectral SNR? Spectral ZDR?

Line 61: you say "the polarimetric variables exhibit sensitivities to specific fall velocities" This is not true please specify this sentence
Line 64: large aggregates do not fall faster than 1m/s, their fall velocity saturates around 1m/s. The Doppler velocity is therefore often used to distinguish between rimed particles and aggregates. Please correct!

Line 91: why do you need the LIDAR? The super-cooled liquid water layers are not visible in your case studies because the lidar doesn't penetrate the rain

Eq. 1-4: see minor comments

Line 125-126: your statement that prolate particles have a negative ZDR is not true. This is only valid if they are oriented in a very specific way. In my experience I have never seen negative ZDR that is only associated with prolate particles, it is mostly attributed to differential attenuation or conical graupel

Eq. 7 and paragraph below that: why do you describe LDR in such a detailed way? Your retrieval is based on ZDR and RhoHV and those are the relevant parameters.

Line 166: please specify the scattering model you use!

Section 3.1: as mentioned in the major comments: please introduce polarizability ratio further, how do you calculate it? How are sZDR and RHV used for that?

Line 187: Also small particles form distinct peaks in the Doppler spectrum (as you can see in your Figure 12)

Line 189-192: please rephrase this sentence, hard to understand

Line 198: if you say not more than 5 shapes can be present please name those shapes. I would even refer to this as particle types, because if you consider dendritic particles, they can have many different shapes

Line 211-212: I don't understand why you need to "harmonize" the Doppler spectra in order to derive the vertical wind, isn't it the opposite way around? You retrieve the wind from the PPI scans in order to match the Doppler spectra across different heights/elevation angles?

Figure 1: Specify which Doppler spectrum (sSNR? SZDR?). Usually Doppler spectrum refers to spectral Ze, which you are not using here correct? Also this sketch is too idealized. I have never seen a Doppler spectrum with 5 distinct peaks. How do you know that columns are falling faster than small dendrites but slower than large dendrites? In my opinion you don't need that plot, but rather a good explanation of how you separate into 5 parts!

Figure 2: it is nice to have a block diagram, however, you should also describe it in the text! e.g. in the text it is not mentioned that you are using minimum square error function which is an important detail.

Line 221-224: I don't see how you depicted turbulence in Figure 1

Equation 8: what is Vw? What is Vh?

Figure 3: I do not understand this figure! Since the wind retrieval is something that is frequently done, this is not necessary. But if you want to leave it in you have to work on that. A few things that I don't understand are: What are the dashed lines? Why is Vh in the dashed line not the same as the grey Vh line? I don't understand the beta angle or the alpha angle in this context.

Paragraph about aliasing: how do you determin n?

Paragraph horizontal wind: do you need to know the true fall speed? See major comment 4

Figure 4: Why is the 0° isotherm not at the same height as the melting layer? Looks like a 500m difference here! Also: if you want to include the lidar measurements please change the colormap you use, I can not see anything in panel f

Line 294-296: How do you know that there is liquid all the way to the cloud top? The lidar doesn't penetrate through the rain.

Please discuss the Figure in more detail. I am missing more discussion of LDR, what does that mean, mean Doppler velocity, what can that tell you about the particles,…

Line 303: what does "transition towards strong spherical particles" mean? Also: I barely see a tendency if I look at Figure 5

I would like to see plots of your RHI scans, similarly to how you have done it in Figure 6. I would suggest to either discuss the radar observations of Figure 4 in more detail to show how they are relevant, or only show Ze, LDR and then the RHI scans.

Figure 5: in your Figure 4 you show that you have the temperature information, It would be helpful to have that also in this figure to draw conclusions about the ice particles (i.e. ice particle habits are strongly dependent on temperature)

Line 314: I would not say it "effectively" identifies the shape, since you are not comparing against other measurements you do not know it that is actually true

Line 315 and following: I don't understand the discussion here. The sentences are really hard to understand. Do you mean that because there are prolate shaped particles in the time between 09:46 and 09:48 they also have to have been present earlier? What do you base that on? You are saying that the cloud changes drastically, therefore I would not compare the two time periods and draw conclusions on the microphysics that are happening.

Figure 6: I would suggest to also show the RHI of the complete spectrum (see also major comments). I would also adjust the colorbar of ZDR and RhoHV, because I can not see any tendency in the variables. ZDR for part 1-4 looks like it is close to 0 for all heights and elevations. Would it not be nice to show

the method on a case that exhibits larger polarimetric signatures? Then the benefit of having a spectrally resolved retrieval would be more obvious. Here even the slowest falling particles show barely any ZDR.

Line 328: You say dealing with noise is a challenge, yet you do not say how you deal with it!

Line 330-331: please rephrase the sentence "this diminished SNR … fails to reflect in ZDR and RHV" how do you know that?

Line 337: ZDR is always really close to 0, I do not see a tendency, so I would just assume that the particles are nearly spherical. Perhaps if you change the colormap it will be visible.

Line 341: ZDR and RHV look nearly exactly the same to me as for part 1,2.

Figure 7: While it looks nice to have all the separate doppler parts and it helps the understanding, it is really hard to see the elevation dependency of ZDR and RHV here. I would suggest to add another figure with that (I assume that you average the different parts over the Doppler velocity to obtain one value of ZDR (RHV) per elevation, so you can show that in the figure)

Line 365: do you mean part 2 and 3? Part 1 is barely existing here

Line 372: are your particles transition into spherical particles? Or is turbulence removing the small ZDR signal that was present at higher altitudes? How can you tell the difference? Or are the largest prolate particles aggregating, therefore leaving only the small prolate particles which have a smaller ZDR? I do not think that particles can change their shape if they have already developed into distinct prolate shapes. For this analysis, again the temperature information would help

Paragraph below Line 373: I do not agree with this analysis. First of all, your SNR was too low to retrieve the shape of the particles which seeded into the region below 4km. This does not mean however that there where no prolate particles that seeded. In addition, your argument that in the later period you see prolate particles is in my opinion not an argument that the particles you had 15 minutes earlier were generated the same way. The cloud clearly changed drastically between the two time steps. In the second time period it is likely that ice particles where formed via a mixed layer at cloud top. However, if the same process was present I would have expected much higher spectral ZDR values to be present at a similar height in the first time period. Especially since in the second period the LDR is really large for the newly generated ice particles! For this discussion it would be really helpful to have the retrieval also for the second time period, so that it is possible to compare the polarisability ratio for the two cases. So if you want to draw any microphysical conclusions I would strongly suggest to include the retrieval of the second time period. I agree with your statement on SIP, however only because ZDR of the slowest falling part of the Doppler spectrum is so low. If there was SIP I would expect much larger values.

Paragraph below 380: Nice discussion about SIP and the melting layer (ML), however, how is that relevant here? You do not have two LDR layers within the ML. I would rather discuss the large number of papers that have found elevated LDR/ZDR above the ML in the needle growth regime than that very special case that Dmitri Moisseev had.

Line 396-397: Why are they not able to infer any information about the "background population"? They also have the Doppler spectrum so they are able to do that. In the Doppler spectrum the particles separate due to their different velocity

Figure 9: do not include lidar here, in this case it really has no valuable information for your case study. And please discuss the figure in detail in the text, do not assume that the reader can deduce all necessary and important information by themselves.

Line 409: Why do you use the time period between 20:01 and 20:11 for the main peak approach? Did you not specify that you are using the time period from 20:37 until 20:47? In the earlier time period there is barely any LDR signal, it is an unfair comparison then to use the main peak approach on a time period where the polarimetry is expected to be low. I would suggest to use the same time period as you are using for your spectral retrieval

Line 411: the increase you see in the polarizability ratio is really small. Is that even significant?

Line 413: is your scattering approximation suited also for liquid?

Line 418: not necessary to mention the specific figure of Li et al, they are not the only ones that observe a second mode in a Doppler spectrum

Paragraph below 419: please rewrite, very hard to follow. The main point is you have a second, slow falling mode which might indicate multiple ice species, especially since the slow falling particles show a different LDR!

Figure 11: not necessary, you can get all the information from the figure 9

Fig. 14 and 15 are nice, however, there are already many figures and by now the reader has understood how the retrieval should work. So focus on the figures with the polarisability ratio and degree of orientation!

Line 462-464: I do not see the indication of fragmentation of dendrites. What are you basing that on?

Line 481-484: while I agree with the statement, my opinion is that with the provided analysis you can not draw that conclusion (see comment above)

---

## Referee Comment (RC2)

Review of Identification of multiple co-located hydrometeor types in Doppler spectra from scanning polarimetric cloud radar observations by Majid Hajipour, Patric Seifert, Hannes Griesche, Kevin Ohneiser, and Martin Radenz for AMT.

In this article, the authors extend the Main-peak approach by Myagkov et al. to the full Doppler spectrum, enabling a spectrally resolved methodology. The topic is noteworthy, and the two study cases are interesting. However, the manuscript requires significant improvement and lacks clarity, particularly in the methodology. To ensure clarity and reproducibility, several aspects of the methodology must be elaborated. The article should be self-contained, allowing readers to fully understand the proposed approach without needing to consult the prior works of Myagkov et al. on shape retrieval and Baars et al. on horizontal wind retrieval. Why is the retrieval of the horizontal wind necessary? Additionally, the manuscript lacks error analysis for the spectrally resolved approach, and its possible limitations are not adequately addressed. For instance, which assumptions are made while comparing spectral polarimetric variables at different elevation angles? How does the method handle significant variability in Doppler spectrum width across different elevations? Furthermore, how are Mie scattering effects accounted for in shape retrievals, especially for the largest/heaviest ice particles? There is no information on the model part of the technique: which scattering model is used? Which ice particle types are considered? Addressing these issues, along with the detailed section comments and corrections provided, will substantially improve the manuscript. I recommend major revisions.

**Comments/corrections/adding's**

**1) Introduction**

Line 39: "Cloud Doppler radars, introduced by Wakasugi et al. (1986), provide backscattered signal...." Rephrase because cloud Doppler radars were not introduced by Wakasugi et al.

**2) Instrumentation**

Can the authors provide a clear set-up of the measurements? The mode RHI is mentioned, but later in section 3.3.1 the retrieval of the horizontal wind using the PPI mode is discussed. Therefore, it is not clear to the reader what the measurement sequence is: a combination of RHI and PPI? Also, the rotation speed of the radar for RHI and PPI measurements should be provided.

Table 1: add the Doppler velocity resolution for both MIRA-35.

**3) Mira-35 radar in hybrid mode**

Revision of Lines 115-153. Attention should be given to the section dedicated to the introduction of polarimetric variables. Rephrasing is necessary and there are some inconsistencies/errors in the text and equations. See below my recommendations.

Line 116: "....as a function of Doppler velocity $\omega$ ..." My suggestion: either "....as a function of Doppler angular frequency $\omega$ ..." or "....as a function of Doppler velocity $v$ ..."

Lines 116-117: No point above the capital letter for $E_h(\omega)$ and $E_v(\omega)$?

Equations (1) and (2) are not correct. The reflectivity is not directly equal to the average modulus square of received complex amplitudes. A constant is missing.

Line 121: "ZDR quantifies the difference between reflectivity measurements in horizontal (Zhh, Eq. 1) and vertical (Zvv, Eq. 2) polarizations, expressed in decibels (dB) (Eq. 3)." Eqs. 1-3 are not expressed in decibels. Be consistent with the text and equations.

Line 125: "At zenith-pointing direction, ZDR is zero. *At slant-pointing direction,* a positive ZDR value ...."

Line 127: "The correlation coefficient (RHV) is a crucial parameter that quantifies the linear relationship between the Zhh and Zvv." Rephrase this statement, which is now not correct.

Lines 128-129: the sentence is not clear and that is not useful to describe Eq. 4 in terms of ratio, sum, square root, product... because that can be directly seen in Eq. 4.

Line 131: remove the point after 1.

Lines 131-132: "... a correlation coefficient of 1 indicates perfect correlation or alignment between horizontal and vertical polarizations, suggesting consistent scattering behavior." Rephrase. What is "alignment between horizontal and vertical polarizations"? What is "consistent scattering behavior"?

Line 135: ".... raindrops, with a spherical shape and ....". Replace "spherical" by "spheroidal".

Line 137: ".... a parameter frequently detected by cloud radars ....". Rephrase. A parameter is not detected.

**4) Main-peak approach**

Is the main peak approach code by Myagkov et al. available online?

Line 163: "This analysis provides insights into particle habits by utilizing a spheroidal scattering model". "A spheroidal scattering model". Which scattering model is used? and "spheroidal scattering model" is not the appropriate name.

Provide the equations of the polarizability ratio and degree of orientation. Explain how they relate to the ZDR and RHV measurements.

**5) Spectrally resolved approach**

I recommend to the authors the extension of the block diagram of Figure 2, where the main peak approach block would appear. Further a zoom of the main peak block, with

inputs and outputs, can be worked out in a second Figure. Presently, without reading in detail the papers Myagkov et al., it is challenging to understand the spectrally resolved technique. The reader should be able to understand the paper without having to read preceding papers.

There is no information on the error analysis.

How is Mie scattering regime accounted for? For example, for Part 1.

Line 185: "Consequently, the Doppler spectra observed with a vertically pointing cloud radar offer insights into the variability of sizes and shapes of the ice particles". Information on the shapes of the ice particles for zenith-pointing cloud radar cannot really be obtained.

Lines 198-199: "The amount of 5 parts was empirically chosen for this study, because usually not more than that amount of different particle shapes can be expected in a cloud volume". Can you provide a reference for this statement?

Lines 199-200: "Increasing the number of parts would result in a reduced amount of available data points per Doppler spectrum part which would lead to increased uncertainties. This statement should be developed. "Which amount of data points for the spectrally resolved approach is recommended? Why? I missed a discussion on this point in terms of possible errors.

Lines 203-204: "Instead, we assume that the fall attitude of the individual hydrometeor types contained in the cloud volume is similar at all elevation angles." Was the same assumption made in the main peak approach?

6) Retrieval of horizontal wind

Figure 3: compared to $V_f$ and $V_h$, $V_R$ is not well scaled. Correct this.

Lines 241-242: ".... while the sine's curve amplitude *yields the wind velocity $V_h$ multiplied by the cosine of the elevation angle, $\psi$*"

Lines 242-243: "Additionally, the entire curve's displacement from the zero velocity *relates to* the precipitation fall speed."

Lines 243-244: "We used the approach of Baars et al. (2023) to derive the horizontal wind components." Describe shortly this approach.

7) Aliasing problems and effects of horizontal winds on the determination of the vertical velocity component.

Line 257: mention what $f_n$ is.

Lines 267-270: The methodology of dealiasing needs to be shortly extended for clarity and reproducibility.

Eq. 10: $V_R$ should be replaced by $V_D$.

8) First case study 07 Nov 2014, 09:15-09:30: retrieval of various hydrometeor types

Figure 4 caption: ....... on November 7, 2014. Correct the date.

Lines 293-294: Between 09:15 and 09:30 UTC, a deep cloud .... which caused precipitation after 09 UTC. Check the time consistency. If it rains from 09:00 UTC, it means that the deep cloud is present before 09:15. Rephrase.

Line 295: The evolution of the mixed phase in this deep cloud.... Why is this deep cloud a mixed-phase cloud? I miss the argumentation here.

Line 335: .... The SNR stabilizes at approximately 60 dB. I think it is much less. 25 dB?

Figure 6 caption: there are errors in the sequence (a)-(o): .... (*l*) RHV in part 2.....(*i*) ZDR, and (n) RHV in part 4, .....

Lines 345-347: Why is it possible to conclude that below 4 km based on RHV and ZDR the particles are prolate (part 5). Provide a short explanation and reference.

Lines 358-361: provide a reference.

Figure 7 caption: error in the sequence (a)-(l): .... (g) RHV spectrum before splitting....

Lines 362-372: in this paragraph the retrieved polarizability ratio shown in Figure 8 is discussed. However, there is no word about the retrieved degree of orientation, part of Figure 8 as well. Why?

Line 363: "For the sake of readability, error bars are omitted in this case ...." OK, but some text related to the error bars should be written in section 3.2. How are the error bars estimated?

9) Second case study 03 Nov 2014, 20:30-20:45: Secondary ice formation

Figure 9 caption: The highlighted period.... are applied. Rephrase the sentence. Also, I don't see the highlighted period in the figure.

Figure 13 caption: there are errors in the sequence (a)-(o): .... (*l*) RHV in part 2.....(*i*) ZDR, and (n) RHV in part 4, .....

Figure 14 caption: error in the sequence (a)-(l): .... (g) RHV spectrum before splitting....

Figure 15 caption: error in the sequence (a)-(l): .... (g) RHV spectrum before splitting....

Lines 462-463: "Indications are given that the branches of oblate ice crystals, such as dendrites fell off, in addition....". Can the authors clarify this statement? Which indications? Is the presence of dendrites in the study case justified? Until now, there was no discussion about the presence of dendrites....

10)    References

The authors should review the reference list.

For example,

uncomplete reference: Melnikov and Sraka, 2013.

Spell-check: Hajipour, M et al. 2024: ....studies.....

---

## Author Comment (AC2)

**Responses to Review Comments on "Identification of multiple co-located hydrometeor types in Doppler Spectra from scanning polarimetric cloud radar observations"**

Majid Hajipour, Patric Seifert, Hannes Griesche, Kevin Ohneiser, and Martin Radenz

21 April 2025

We sincerely thank both reviewers for their valuable feedback and constructive comments. Their input has greatly helped to improve the quality and clarity of the manuscript, to our opinion. We hope that the reviewers share this notion. Below, we aim to respond to the issues they raised on a point-by-point basis. We decided to provide the replies to both reviewers within one review letter. Reason is that some comments were raised by both reviewers. We took the liberty to point to a given reply if a reviewer comment is similar to a previous one.

Throughout our reply letter, we use the following abbreviations:

- R#C\$: Comment number \$ by Reviewer number #
- AR#-\$: Authors' response to comment number \$ of reviewer number #

**Response to Reviewer #2**

**Major:**

**R1C1 (splitted into individual parts):**

A) the text needs a lot of rewriting, both structurally and for a better understanding of a few sentences (which I will specify in the specific comments). In my opinion, the new method is not explained satisfactorily. I am missing a detailed explanation on why you chose to divide the spectrum into 5 parts, and how you are dividing the spectrum, which has not been mentioned anywhere. I do not know which 5 species/shapes you want to cover with that, I can only think of rimed particles, aggregates, plate-like crystals, columnar crystals. Of course you can also have super-cooled liquid water, but for that we know the shape very well, so no retrieval is necessary. I would also like to know if you are trying to "track" the different species throughout the different heights, or if the division of the spectrum is just random.

- B) Also, please explain the main peak method better, for me it was difficult to understand that. Other more "trivial" retrievals such as the vertical wind velocity are explained in great detail, even though it is not that relevant to the study and many institutions are running wind retrievals on an automatically basis. I would suggest to rewrite the paper the following way:
- I like Figure 6 and 7, perhaps you can move that to the method section? Then it might be clearer how the retrieval works. If you include the full RHI scan in Figure 6 (without the separation into the 5 parts) and 7, you can first explain the main peak method and then the spectral approach in a clearer way. I would first explain in detail which peak is used in the main peak approach, indicate that in the figures and then continue on to your new approach and show how that is different and how you are dividing into 5 parts. How are you then using the 5 parts? Are you averaging along the Doppler velocity of each part? If so, it might be helpful to provide the averaged elevation dependencies of ZDR and RhoHV either in addition to the Doppler spectra parts or instead of them. Also you should specify what the polarisability ratio and the degree of orientation are and provide the formulas. This has of course been mentioned in Myagkov et al. 2016, however, these are not standard variables and I think it is therefore necessary to explain that again. You can even use panel a and c of their figure 13 to show the polarisability ratio and orientation. When you are explaining polarizability ratio you can also mention that the shape retrieval can be used with sZDR and sSLDR.

**AR1-1:** Thanks for these detailed comments and suggestions.**

**AR1-1A:** Our main objective is to extend the technique developed by Myagkov et al. 2016 to probe cloud layers for the co-occurrence of multiple hydrometeor types. Thereby, we don't focus on any specific hydrometeor type. We basically decided to evaluate the Doppler spectra for the occurrence of a maximum of 5 different hydrometeor populations. We selected 5 parts because it would to our opinion cover the maximum of different hydrometeor types, namely droplets, pristine ice, rimed ice, aggregates and rain/drizzle. To achieve the spectral separation, we identified the starting and ending points of the Doppler spectra for each pair of range and elevation angle and then divided the spectra's width into five equal parts. Subsequently, we calculated the average ZDR and RHV values for each part to provide a single representative value for the next stage of analysis (comparison stage), similar to the main-peak approach. This explanation has been included in the text. The motivation and introduction of the spectrally resolved approach in the manuscript has been extended and now spans across lines 245-282 of the revised manuscript.

A tracking of particles across different height levels is not performed, but is a very good point for future improvements of the technique. This point was thus added to the conclusions section.

**AR1-1B:**

In order to improve the explanation of the main-peak technique, we modified the text (lines 187 to 202) and expanded on the concepts of polarizability ratio and degree of orientation in Section 3.1. This includes incorporating the relevant formulas to review these concepts as presented in Myagkov et al. (2016). While we acknowledge the suggestion of R#1 to move Figs. 6 and 7 to the methods section, we decided to not do so. Both figures require a lot of introduction which is much better placed in the results section.

We included RHI scans of SNR, ZDR, and RHV for the main-peak approach in both case studies (Fig. 5, Fig. 11). RHI scans for the spectrally resolved approach can be found in Fig. 7 and Fig. 15. An important observation from the RHI scans for both case studies is that the ZDR and RHV profiles in the main-peak approach closely resemble those in parts 1 to 4 of the spectrally resolved approach. However, the distinct ZDR and RHV signatures in part 5 of the spectrally resolved approach indicate the presence of a different hydrometeor type. In the main-peak approach, we select the peak of the SNR spectrum, as illustrated in the Fig 1a. For the spectrally resolved approach, the spectrum is divided into five parts, as shown in the Fig 1b. It is important to note that the main peak in the main-peak approach does not necessarily correspond to any of the five parts in the spectrally resolved approach, since the spectra are split into equally spaced (by means of Doppler velocity) parts, without considering the occurrence of individual peaks. Based on experiences made in the course of the development of the spectrally-resolved approach, we were convinced that individual spectral peaks are hardly detectable at large off-zenith angles due to spectra-broadening effects by the horizontal wind.

*Fig 1. (a): main peak selection in the main peak approach. (b): 5 parts of Doppler spectra after splitting in the spectrally resolved approach*

Concerning the request to mention that the shape retrieval can be also used with sZDR and sSLDR, we acknowledge this fact now at the end of Section 3.1 (lines: 229-231)

**R1C2:**

I am missing in this paper a few clear statements about the problems of the method. i.e. you should critically discuss the scattering properties that you are using. You did not specify which spheroidal method you are using, in Myagkov et al. the retrieval is based on Rayleigh. It is well known that spheroidal methods have issues of representing the scattering of ice particles accurately, especially in the Mie-region (which is reached at Ka-Band for aggregates) and for low density ice particles such as aggregates. I understand that generating a large DDA database which is necessary to do such a retrieval is difficult. However, I know that other working groups have done that and you could collaborate with them. I think that especially now that you want to retrieve the shape of multiple species a different scattering method should be taken into account. In my opinion it would have been beneficial to collaborate with other groups who have sophisticated scattering databases based on e.g. DDA. Another thing to discuss: can you be sure that along all elevation angles the species in one height doesn't change? For larger heights this spans quite a large cloud area

**AR1-2:** We did not modify the spheroidal model used in the main-peak approach. In fact, our goal was to take the scientifically very exciting step to extend the existing main-peak approach by analyzing the full Doppler spectra. For this study, we restricted the analysis to cases where the larger particles in part 1 were likely to be small, based on their fall speed.

The evaluation of alternative models, such as the Discrete Dipole Approximation (DDA), to enhance the analysis, shall be left for future studies. We personally doubt that the current retrieval framework would work in case of DDA. Reason is, that the spheroidal model used by Myagkov et al. (2016a) was able to use a direct relationship between the measured parameters ZDR and RHV and the microphysical properties of polarizability ratio and degree of orientation, respectively (two observables, two unknowns). This is only possible because the spheroidal model is independent of particle size and concentration. As DDA requires also size and concentration as input parameters, a completely new approach would be required in order to relate the observables with the microphysical parameters. E.g., one could imagine that such an extended, DDA-compatible retrieval approach might be possible if all novel technical developments from the recent years would be included, such as spectrally-resolved reflectivity and dual- or triple-wavelength ratio of the latter for size information, and (spectral) KDP for density and number information. All of these for different elevation angles. While multi-frequency polarimetric datasets do meanwhile exist (e.g, TRIPEX-pol; von Terzi et al., 2022), they are not widely available (Kneifel et al., 2011; Leinonen et al., 2013; Chellini et al., 2024), and are usually obtained for fixed elevation angles, only. The issue was extensively discussed in the course of the publication process of the study of Teisseire et al. (2024, see the discussion at

https://amt.copernicus.org/articles/17/999/2024/amt-17-999-2024-discussion.html ) and

was also previously communicated in a study of Matrosov, (2021), and shall not be repeated here.

One of the assumptions of the main-peak approach is homogeneity at each height, and this assumption was retained in the spectrally resolved approach. However, the consideration of five distinct hydrometeor types at each height actually already introduces an implicit assumption of inhomogeneity of the particle habits within the volume. The approaches to evaluate horizontal homogeneity, which we applied, were (1) to require a minimum amount of data points at each elevation angle of an RHI scan, and (2) to evaluate monotonic behavior of the elevation-dependency of ZDR and RHV. As can be seen in the theoretical model of the relationship between ZDR, RHV, polarizability ratio and degree of orientation, the elevation dependency is always required to be monotonic. When no monotonic slope of the elevation dependency of ZDR and RHV is detected, then the presence of inhomogeneities is likely. (3) The applied correction of the horizontal wind field on the elevation-dependency of the Doppler spectra can be used as a measure of cloud homogeneity. We added these three measures of homogeneity to lines 232 - 243 of section 3.1. To conclude the given statements, the evaluation of the horizontal homogeneity is in practice still subject to experienced-eye check, taking into account the above-mentioned 3 constraints. Cases, where the abovementioned criteria are not met shall be excluded from the analysis.

We would like to note here, aside to the given evaluation of the homogeneity in our retrieval, that retrievals that are applied to standard-measurements of precipitation radar network instruments are much more relaxed regarding the assumption of horizontal homogeneity. One example is the prominent approach of obtaining so-called quasi-vertical profiles (QVP) by averging over a range of azimuth (of up to 360°) of fixed-elevation (down to between 6.4° and 28°) precipitation radar scans (Ryzhkov et al, 2017). QVP approaches thus assume horizontal homogeneity over hundreds of square kilometers.

**R1C3:**

Along with comment 3: aggregates have a small polarimetric signature, especially if you are assuming a spheroidal scattering approximation. How much more insight do you even gain when you are considering the spectral lines with aggregates? I am expecting that similar to rimed particles they will just have a polarizability ratio of close to 1. Can you even distinguish between these different particles then? If you can only distinguish between them because they have a different fall velocity is it even worth splitting the Doppler spectra into 5 parts? Would it not be beneficial to just separate between ice crystals and aggregates? I would suggest to show the theoretical polarizability ratio you would expect for all your 5 different species (which I am assuming to be rimed particles, aggregates, dendritic ice crystals and columnar ice crystals and something else). This would show that a separation into 5 parts is actually necessary and that not i.e. 2 parts would suffice.

**AR1-3:** It is correct that by using only the spectrally resolved approach, it is not possible to distinguish between rimed and aggregated particles, because the polarizability ratio for both is approximately 1. Additional information is required to differentiate between them. One approach is to check for the presence of liquid droplets in regions where the polarizability ratio approaches 1. If liquid droplets are detected, it can be inferred that riming is occurring. Otherwise, the most likely interpretation is aggregation. Furthermore, dual- or triplewavelength ratios from multi-frequency cloud radar observations can help to distinguish riming from aggregation. Teisseire et al., (2025) combined a modified version of the mainpeak approach with liquid-detection retrievals and dual-frequency radar observations to show the added value of the shape and orientation retrieval for cloud microphysics retrievals. As already outlined before in this reply letter, the initial version of the spectrally resolved approach, that is presented in this study, aims on a general evaluation of the potential presence of different hydrometeor types, without the demand to apply any a-priori assumption about which particle types are present. An application of the spectrally-resolved shape retrieval for an actual (microphysical) hydrometeor type classification is subject to future studies. It is important to clarify that we did not intend to associate each part with a specific hydrometeor type. Depending on the location within the Doppler spectrum, the parts may be close to 0 m s-1, indicating small particles, or further from 0 m s-1, representing larger particles. As a result, associating each part with a specific hydrometeor type is not feasible. The polarizability ratio and degree of orientation for each part reflect the shape and orientation of the particles that constitute that part. Overall, while the spectrally resolved approach enhances our understanding of clouds by enabling the retrieval of multiple hydrometeor types, its greatest strength lies in its ability to significantly improve the interpretation of microphysical processes when combined with additional data inputs. We added a passage to the conclusions section which discusses the focus of the in-here presented first/plain version of the spectrally resolved approach.

Regarding the request to present polarizability ratios for different types of hydrometeors, we'd like to point the reviewer to Fig.1 in the study of Myagkov et al., (2016a). We added a reference to this figure into our manuscript in line 237.

**R1C4:**

Why do you need to retrieve the actual air motion? Do you use the Doppler velocity for anything and therefore a knowledge of the fall velocity of the particles are valuable? Otherwise you could just move the Doppler spectrum to 0m/s. This would save a lot of effort and space in the paper.

**AR1-4:** The Doppler spectra correction includes three components: folding/aliasing correction, elevation-angle correction, and horizontal wind correction. Just to move the spectrum to zero m/s would not lead to the same result and would lead to Doppler spectra whose width is also influenced by the elevation angle. This would in turn potentially affect the number of selected data points per Doppler spectral part. In addition, as written already

in AR1-2, we consider it valuable to use the homogeneity of the Doppler-spectrum width at different elevation angles as one additional measure for evaluation of the cloud homogeneity. These aspects motivated us to apply the general wind-field correction, which we want to get published in addition as guideline for future studies.

Additionally, retrieval results can be obtained based on the mean Doppler velocity of each part. Figure 2 below illustrates the results for November 7, 2014, during the period from 09:16 to 09:18 UTC. We did not use this representation in the current manuscript as it is planned to be used in a future publication.

Fig 2. Illustration of the spectral shape retrieval for a RHI scan from Cabauw, NL, observed on November 7, 2014, 09:16-09:18 UTC. Shown are profiles of (left panel) polarizability ratio and (right panel) degree of orientation as a function of mean Doppler velocity of each of the 5 parts of the spectral shape retrieval. Left-hand columns represent part 1, right-hand columns represent part 5.

**Minor:**

**R1C5:**

discuss your figures better/more in the text. Especially Fig 4 and Fig. 9 are not discussed enough. Most readers are not familiar with all the Radar and Lidar variables and can not draw their own conclusions.

**AR1-5:** We improved the discussion of the figures as requested. This hopefully will also help the future reader to get a better introduction to the case studies.

**R1C6:**

In this regard: why do you show the Lidar measurements? Due to the rain the Lidar doesn't even penetrate into the relevant cloud regions, so you can just omit the plots.

**AR1-6:** In the first case study, the lidar is very helpful. In the course of this revision we carefully introduce that the lidar information is key to identify the layer of supercooled liquid water. For the second case study (now Fig. 10) we followed the suggestion of the reviewer and removed the lidar panels. There, we only note in the text, that the lidar measurement does not provide valuable information due to strong attenuation of the signal above the melting layer.

**R1C7:**

You need to refine your naming in the equations better. For example, before equation 1 you say that E\_h is the horizontally polarized plane of the received wave. You never specified what E dot means. Then in equation 4 you say that E\_h is the copolar element of the backscattering matrix. Please be specific with your definitions, and stick with one! (also co and cross-polar are not the same thing as horizontal and vertically polarized)

**AR1-7:** The dot above each word indicates that the parameter is complex. Additionally, we removed the definition of RHV, as it was also mentioned by the second reviewer. Additionally, we removed the explanation of RHV including co-polar and cross polar.

**R1C8:**

In your conclusions I am missing a clear outlook and a discussion of the method and its caveats.

**AR1-8:** As suggested by the reviewer, we extended the conclusions and outlook section. Especially with respect to the caveats which were also elaborated on in the course of this review.

**R1C9:**

The colormap in your Figures looks like jet, perhaps you could consider moving towards a colormap which does not have so many deficiencies, if you prefer rainbow you could use e.g. turbo or something similar.

**AR1-9:** We evaluated all figures with the colorblind tester https://www.color-blindness.com/coblis-color-blindness-simulator/ and could not find any loss of information by using the current color scheme.

**R1C10:**

why are you only analyzing the first time period of 7.11.2014 with your spectral retrieval? The second time period would have been interesting, since there the polarimetric variables are large!

**AR1-10:** We only show results for the first time period because only there multiple hydrometeor types were located. For the sake of saving figure space, we thus presented only the main-peak result of the second time period, which could easily be added as sub-panels to Figs. 5 and 6.

**Specific comments:**

**R1C11:**

Line 47-48: you say peak signal of the Doppler spectrum, could you specify which variable you mean? Spectral reflectivity? Spectral SNR? Spectral ZDR?

**AR1-11:** We selected the peak of the spectral SNR and do mention this now at this location in the text (lines 73-74).

**R1C12:**

Line 61: you say "the polarimetric variables exhibit sensitivities to specific fall velocities" This is not true please specify this sentence

**AR1-12:** We removed this statement from the manuscript text.**

**R1C-13:**

Line64: Large aggregates do not fall faster than 1m/s, their fall velocity saturates around 1m/s. The Doppler velocity is therefore often used to distinguish between rimed particles and aggregates. Please correct!

**AR1-13:** Thanks for the hint. We corrected the passage accordingly. See lines 89-91.**

**R1C14:**

The super-cooled liquid water layers are not visible in your case studies because the lidar doesn't penetrate the rain

**AR1-14:** As stated in AR1-6, we see good reason for presenting and discussing the lidar observations in Fig. 4. In the case of Fig. 9 (Fig 10 in the revised version), we removed the lidar panels.

**R1C15:**

**Eq. 1-4: see minor comments**

**AR1-15:** We applied the correction and explained it in the minor comment section. See AR1-7.

**R1C16:**

Line 125-126: your statement that prolate particles have a negative ZDR is not true. This is only valid if they are oriented in a very specific way. In my experience I have never seen negative ZDR that is only associated with prolate particles, it is mostly attributed to differential attenuation or conical graupel

**AR1-16:** Thank you for the hint. You are right. We corrected the text passage: "At zenith-pointing direction, ZDR is zero. A positive ZDR value may indicate ice particles that are horizontally aligned, whereas a negative value might suggest particles aligned vertically."

**R1C17:**

Eq. 7 and paragraph below that: why do you describe LDR in such a detailed way? Your retrieval is based on ZDR and RhoHV and those are the relevant parameters.

**AR1-17:** We shortened the text as suggested.**

**R1C18:**

Line 166: please specify the scattering model you use!

**AR1-18:** We added information on the scattering model and the respective references to Section 3.1.

**R1C19:**

Section 3.1: as mentioned in the major comments: please introduce polarizability ratio further, how do you calculate it? How are sZDR and RHV used for that?

**AR1-19:** In the middle of Section 3.1 (lines 197–225), we added a detailed explanation along with Equations (8) to (11) to describe the polarizability ratio and the degree of orientation, as well as how ZDR and RHV can be derived from these parameters

**R1C20:**

**Line 187: Also small particles form distinct peaks in the Doppler spectrum (as you can see in your Figure 12)**

**AR1-20:** We rephrased the passage to achieve a more general description: "The width of the Doppler spectra is thereby characterized by size- and shape-dependent fall velocities of the particles, which are super-imposed by influences of turbulence and (predominantly in case of off-zenith antenna pointing angle) horizontal winds that cause additional broadening of the spectrum Radenz et al (2019)." See lines 250-253 of the revised manuscript.

**R1C21:**

**Line 189-192: please rephrase this sentence, hard to understand**

**AR1-21:** We rephrased it like this in the text: "This study extends the main-peak shape and orientation retrieval through the spectrally resolved approach, which assumes that different hydrometeor types in a cloud volume are separated by their distinct fall speeds, as explained in the previous paragraph." See lines 254-256 of the revised manuscript.

**R1C22:**

Line 198: if you say not more than 5 shapes can be present please name those shapes. I would even refer to this as particle types, because if you consider dendritic particles, they can have many different shapes

**AR1-22:** Separating into five parts does not imply there are exactly five distinct hydrometeor types. This number was chosen as the best estimate for the number of hydrometeor types and also to effectively represent the existing shapes (see also AR1-1). We added additional text to the beginning of Section 3.2 to improve the introduction and motivation of our approach (as was also requested in the major comments).

**R1C23:**

Line 211-212: I don't understand why you need to "harmonize" the Doppler spectra in order to derive the vertical wind, isn't it the opposite way around? You retrieve the wind from the PPI scans in order to match the Doppler spectra across different heights/elevation angles?

**AR1-23: See AR1-4.**

**R1C24:**

Figure 1: Specify which Doppler spectrum (sSNR? SZDR?). Usually Doppler spectrum refers to spectral Ze, which you are not using here correct? Also this sketch is too idealized. I have never seen a Doppler spectrum with 5 distinct peaks. How do you know that columns are falling faster than small dendrites but slower than large dendrites? In my opinion you don't need that plot, but rather a good explanation of how you separate into 5 parts!

**AR1-24:** We replaced the figure with a more realistic one. The illustration of different shapes is intended as a visual aid to show that distinct shapes can be separated based on their velocities. It does not imply that a specific shape consistently has a higher or lower speed.

**R1C25:**

Figure 2: it is nice to have a block diagram, however, you should also describe it in the text! e.g. in the text it is not mentioned that you are using minimum square error function which is an important detail.

**AR1-25:** The description of Fig. 2 was extended\_ and now covers lines 284-295. We added the description of the minimum mean square error function to the text.

**R1C26:**

Line 221-224: I don't see how you depicted turbulence in Figure 1.

**AR1-26:** Turbulence can be represented by the upward motion of particles, indicating a positive velocity. In addition, it leads to the 'smearing' of individual Doppler peaks. Both aspects are considered in the revised version of Fig. 1.

R1C27: Equation 8: what is Vw? What is Vh?

**AR1-27:** V\_h represents the velocity of the horizontal wind and V\_w represents the map of horizontal wind to radar line of sight direction. We calculated V\_w to be able to remove the effect of wind from the radial velocity of hydrometeors measured by radar (line 292-303).

**R1C28:**

Figure 3: I do not understand this figure! Since the wind retrieval is something that is frequently done, this is not necessary. But if you want to leave it in you have to work on that. A few things that I don't understand are: What are the dashed lines? Why is Vh in the dashed line not the same as the grey Vh line? I don't understand the beta angle or the alpha angle in this context.

**AR1-28:** We revised and cleaned the figure, removing beta, which indicated the azimuth angle of the radar. We focused on retaining only the parameters related to wind. Alpha also represents the wind direction angle.

**R1C29:**

**Paragraph about aliasing: how do you determine n?**

**AR1-29:** In this study, since the wind speed was not very strong, the aliasing issue was consistently resolved by setting n = 1. In this case, folding occurs at low elevation angles (between 30-60° or 120-150°). However, in the presence of stronger winds, folding begins at

elevation angles closer to the zenith (between  $60-90 \circ$  or  $90-120 \circ$ ), requiring the consideration of n > 1 (lines 339-343).

**R1C30:**

Paragraph horizontal wind: do you need to know the true fall speed? See major comment 4

**AR1-30:** No, knowledge of the true fall speed is not required. See AR1-4 for more details.**

**R1C31:**

Figure 4: Why is the 0° isotherm not at the same height as the melting layer? Looks like a 500m difference here! Also: if you want to include the lidar measurements please change the colormap you use, I can not see anything in panel f

**AR1-31:** The temperature information is taken from a model (gdas1) which can therefore deviate to some extent from the true temperatures. Also, it should be noted that melting usually occurs at heights below the height level of the 0°C isotherm. This is the case because melting generally only starts when the wet-bulb temperature approaches temperatures greater than 0°C. This can also explain up to a few hundred meters of height difference to the 0°C air temperature. See Ryzhkov and Krause, 2022.

**R1C32:**

Line 294-296: How do you know that there is liquid all the way to the cloud top? The lidar doesn't penetrate through the rain.

**AR1-32:** Thank you for pointing us to this typo. We meant 'ice phase' and corrected the position accordingly.

**R1C33:**

Please discuss the Figure in more detail. I am missing more discussion of LDR, what does that mean, mean Doppler velocity, what can that tell you about the particles,...

**AR1-33:** We thank the reviewer for pointing us to the lack of detailed introduction to the case study. As suggested, we extended the introduction and discussion of the 7-November 2014 case study considerably. We now involve all sub-panels from Fig. 4 and provide already at this introduction stage some suggestions about the associated microphysical cloud properties. See lines 360 to 389 of the revised manuscript.

**R1C34:**

Line 303: what does "transition towards strong spherical particles" mean? Also: I barely see a tendency if I look at Figure 5 I would like to see plots of your RHI scans, similarly to how

you have done it in Figure 6. I would suggest to either discuss the radar observations of Figure 4 in more detail to show how they are relevant, or only show Ze, LDR and then the RHI scans.

**AR1-34:** Indeed, the increase of polarizability ratio between 4 and 2.5 km height is too weak to motivate the given emphasize on sphericity. We therefore decided to remove this statement.

**R1C35:**

Figure 5: in your Figure 4 you show that you have the temperature information, It would be helpful to have that also in this figure to draw conclusions about the ice particles (i.e. ice particle habits are strongly dependent on temperature)

**AR1-35:** Thanks for the suggestion to improve the usability of the results plots. We now added the GDAS1 temperature levels to all figures showing profiles of polarizability ratio and degree of orientation.

**R1C36:**

Line 314: I would not say it "effectively" identifies the shape, since you are not comparing against other measurements you do not know it that is actually true

**AR1-36:** The performance of the main-peak approach has already been evaluated with laboratory results, which showed good agreement. Based on this evaluation, we assume that the current results are also reliable.

**R1C37:**

Line 315 and following: I don't understand the discussion here. The sentences are really hard to understand. Do you mean that because there are prolate shaped particles in the time between 09:46 and 09:48 they also have to have been present earlier? What do you base that on? You are saying that the cloud changes drastically, therefore I would not compare the two time periods and draw conclusions on the microphysics that are happening.

**AR1-37:** What we meant with 'the cloud changes drastically' was that its vertical structure changed from a seeder-feeder constellation (09:15-09:30) to a single-layer stratiform mixed-phase cloud (09:30-10:00). Based on the observations we consider our conclusion true that the same layer of supercooled liquid water with precipitating columnar ice crystals, which was observed from 09:30-10:00 (see Fig. 4), was also present between 09:15 and 09:30 when the cloud was deeper and the second, more isometric particle shape was present. We see this constellation as an evidence, that the prolate ice formed heterogeneously in the thin layer of supercooled liquid water. No seeding by the higher-level cloud and thus no rime-splintering was required to explain the observed mode of prolate ice crystal shapes.

**R1C38:**

Figure 6: I would suggest to also show the RHI of the complete spectrum (see also major comments). I would also adjust the colorbar of ZDR and RhoHV, because I can not see any tendency in the variables. ZDR for part 1-4 looks like it is close to 0 for all heights and elevations. Would it not be nice to show the method on a case that exhibits larger polarimetric signatures? Then the benefit of having a spectrally resolved retrieval would be more obvious. Here even the slowest falling particles show barely any ZDR.

**AR1-38:** We added the RHI scan for the main-peak approach. Additionally, we would like to point out that, using Figure 6, we aim to show that the profiles of parts 1 to 4 are similar to the main-peak profile. The only noticeable difference is in part 5, where the ZDR and RHV above the melting layer at low elevation angles exhibit a different signature, as shown in this figure.

**R1C39:**

Line 328: You say dealing with noise is a challenge, yet you do not say how you deal with it! when the SNR is too low.

**AR1-39:** We modified the text ackordingly: "Managing noise can be challenging in the spectrally resolved approach, when the SNR is too low.". What I meant is that if the SNR is too low (in this work, less than 10 dB), the data becomes heavily contaminated, making retrieval impossible. However, in these cases, the SNR was not excessively low. By using RHI scans, we can assess the extent of noise contamination in the data. In these instances, while the data is contaminated, it is not to a degree that would lead to incorrect results.

**R1C40:**

Line 330-331: please rephrase the sentence "this diminished SNR ... fails to reflect in ZDR and RHV" how do you know that?

**AR1-40:** We rephrased the statement using the following text: "At higher altitudes, the low SNR in part 1 (representing the fastest falling particles) of the Doppler spectra prevents the complete representation of ZDR and RHV profiles." At higher altitudes, which are not shown here, the SNR profile exhibits very low values. However, at the same altitude, there is no detectable signature of ZDR and RHV.

**R1C41:**

Line 337: ZDR is always really close to 0, I do not see a tendency, so I would just assume that the particles are nearly spherical. Perhaps if you change the colormap it will be visible.

**AR1-41:** Yes, it was typo. We replaced with spherical shape.

**R1C42:**

Line 341: ZDR and RHV look nearly exactly the same to me as for part 1,2.

**AR1-42:** Yes, that's true. We replaced with spherical shape.**

**R1C43:** Figure 7: While it looks nice to have all the separate doppler parts and it helps the understanding, it is really hard to see the elevation dependency of ZDR and RHV here. I would suggest to add another figure with that (I assume that you average the different parts over the Doppler velocity to obtain one value of ZDR (RHV) per elevation, so you can show that in the figure)

**AR1-43:** Our goal was to highlight that the elevation dependency of ZDR and RHV is only noticeable in part 5. The text regarding the other parts, as mentioned in the last two comments, was incorrect. We believe the figure is now presented more clearly.

**R1C44:** Line 365: do you mean part 2 and 3? Part 1 is barely existing here.**

**AR1-44:** Yes, we corrected it and rephrased the text like this: At an altitude of approximately 6 km, the retrieved polarizability ratio for parts 2 and 3 is 0.9 and 1.1, respectively..

**R1C45:**

Line 372: are your particles transition into spherical particles? Or is turbulence removing the small ZDR signal that was present at higher altitudes? How can you tell the difference? Or are the largest prolate particles aggregating, therefore leaving only the small prolate particles which have a smaller ZDR? I do not think that particles can change their shape if they have already developed into distinct prolate shapes. For this analysis, again the temperature information would help

**AR1-45:** We don't mean that the shape is changing. Our interpretation is that as the polarizability ratio approaches 1, the shape is becoming more spherical, which is due to microphysical processes like aggregation or other factors. In fact, we observed that changes in shape are associated with microphysical processes, but we cannot specifically identify which processes were involved.

**R1C46:**

Paragraph below Line 373: I do not agree with this analysis. First of all, your SNR was too low to retrieve the shape of the particles which seeded into the region below 4km. This does not mean however that there where no prolate particles that seeded. In addition, your argument that in the later period you see prolate particles is in my opinion not an argument that the particles you had 15 minutes earlier were generated the same way. The cloud clearly changed drastically between the two time steps. In the second time period it is likely that ice particles where formed via a mixed layer at cloud top. However, if the same process was

present I would have expected much higher spectral ZDR values to be present at a similar height in the first time period. Especially since in the second period the LDR is really large for the newly generated ice particles! For this discussion it would be really helpful to have the retrieval also for the second time period, so that it is possible to compare the polarisability ratio for the two cases. So if you want to draw any microphysical conclusions I would strongly suggest to include the retrieval of the second time period. I agree with your statement on SIP, however only because ZDR of the slowest falling part of the Doppler spectrum is so low. If there was SIP I would expect much larger values.

**AR1-46:** We acknowledge that Reviewer #1 shared his concerns about the interpretation of our first case study. Similar to as we replied in AR1-37, we however consider our interpretation valid. The application of the spectrally resolved approach definitely identified the co-location of a prolate-particle-bearing layer and a vertically extensive seeder cloud, which would suggest the occurrence of of secondary rime-splintering processes. In our case study, corroborated by lidar observations of low volume depolarization ratio, however found that the prolate particles were formed in a layer of supercooled liquid water, independent of the presence of any seeder cloud from above. We consider it scientifically relevant to highlight such an observation, as current studies in general inteprete the co-location of a seeder cloud and prolate particles as an indication for the presence of secondary ice formation. To corroborate our interpretation, we extended the discussion of the case study (lines 483-490) where we also provide references.

**R1C47:**

Paragraph below 380: Nice discussion about SIP and the melting layer (ML), however, how is that relevant here? You do not have two LDR layers within the ML. I would rather discuss the large number of papers that have found elevated LDR/ZDR above the ML in the needle growth regime than that very special case that Dmitri Moisseev had.

**AR1-47:** In Figure 11, We showed that during the time interval from 20:40 to 20:47, there are two melting layers below the height of 1.85 km.

**R1C48:**

Line 396-397: Why are they not able to infer any information about the "background population"? They also have the Doppler spectrum so they are able to do that. In the Doppler spectrum the particles separate due to their different velocity

**AR1-48:** They use a vertically pointed radar and analyze LDR values to obtain general information about hydrometeors, incorporating other parameters like temperature. Based on this setup one cannot distinguish between (horizontally aligned) oblate and spherical particles (such as aggregates), since both produce low LDR due to their similar spherical cross-sections with respect to the radar line of sight. In contrast, our approach involves a

scanning radar, where we analyze Doppler spectra at all elevation angles. This allows us to present a more detailed and comprehensive analysis of the hydrometeor shapes. Especially oblate particles can be better discriminated from spherical ones, as the cross-sections of both particle species differ strongly when observed at low elevation angles. Additionally, we rephrased with this text: "their method is not able to detailed information about the background population of ice crystals from which the secondary ice formed."

**R1C49:**

Figure 9: do not include lidar here, in this case it really has no valuable information for your case study. And please discuss the figure in detail in the text, do not assume that the reader can deduce all necessary and important information by themselves.

**AR1-49:** As already requested/suggested above, the lidar panels were removed from Fig. 10 (previously Fig 9)

**R1C50:**

Line 409: Why do you use the time period between 20:01 and 20:11 for the main peak approach? Did you not specify that you are using the time period from 20:37 until 20:47? In the earlier time period there is barely any LDR signal, it is an unfair comparison then to use the main peak approach on a time period where the polarimetry is expected to be low. I would suggest to use the same time period as you are using for your spectral retrieval

**AR1-50:** We revised the text, updating the times to 20:30 and 20:45. Additionally, we corrected the title of Figure 12.

**R1C51:**

**Is that even significant?**

**AR1-51:** It's not significant; it's simply an explanation of the changes in the polarizability ratio.

**R1C52:**

**Line 413: is your scattering approximation suited also for liquid?**

**AR1-52:** No, we just wanted to demonstrate how this technique, using the spheroid scattering model, retrieves a very low polarizability ratio for liquid, which is incorrect. Of course, we focus only on ice, not liquid.

**R1C53:**

The main point is you have a second, slow falling mode which might indicate multiple ice species, especially since the slow falling particles show a different LDR!

**AR1-53:** We removed the references to figures from other papers and replaced the text below line 540 with the text below.

"Elevated radar reflectivity levels at 2.9 km and within the lower velocity range of the Doppler spectrum suggest the presence of different hydrometeor types. Additionally, a significant increase in LDR values, reaching about -17 dB, was observed from above the melting layer to 2.9 km in the slow-falling velocity range, indicating the dominance of prolate-shaped ice particles in this part of the spectrum."

**R1C54:**

Figure 11: not necessary, you can get all the information from the figure 9

**AR1-54:** We added a zoom-in into Fig. 13 to highlight the two melting layers.

**R1C55:**

Fig. 14 and 15 are nice, however, there are already many figures and by now the reader has understood how the retrieval should work. So focus on the figures with the polarisability ratio and degree of orientation!

**AR1-55:** These figures are essential for a better understanding of the spectrally resolved approach. Therefore, we believe it is important to present them before showing the final results.

**R1C56:**

**Line 462-464: I do not see the indication of fragmentation of dendrites. What are you basing that on?**

**AR1-56:** The second case study aims on revealing the shape of the seeding particles which are the prerequisite for the observed secondary ice formation / ice multiplication, which was observed just above the melting layer (as indicated in Fig. 18 by the sudden switch of the polarizability ratio toward prolate particles in Part 1 at 2.5 km and below). As the shape retrieval identified, all other parts show oblate structures, with a polarizability ratio of around 0.7. This is the reason for our conclusion that the seeding particles likely were dendritic/oblate ice crystals. Our motivation for this case study was simply that the vertical-stare approach of Li et al. 2021 was not able to distinguish whether the seeding particles (seeding the layer of secondary ice formation) were oblate or spherical. This is simply not possible, as both particle types produce low LDR in vertical-stare mode. We would thus propose to keep the conclusions for this case study (as shown in lines 574-585).

**R1C57:**

Line 481-484: while I agree with the statement, my opinion is that with the provided analysis you can not draw that conclusions (see comment above)

**AR1-57:** We hope that we provide the Reviewer #1 sufficient reasoning for our conclusion by means of our statements given in AR1-37 and AR1-46.

**Response to Reviewer #2**

Thanks for your comments and suggestions. As some of the comments have already been addressed in our responses to the comments of Reviewer #1, we took the liberty to point to the respective responses where applicable.

**Comments/corrections/adding's**

1) Introduction

**R2C1:**

Line 39: "Cloud Doppler radars, introduced by Wakasugi et al. (1986), provide backscattered signal...." Rephrase because cloud Doppler radars were not introduced by Wakasugi et al.

**AR2-1:** We have removed the reference Wakasugi et al. (1986).

**2) Instrumentation**

**R2C2:**

Can the authors provide a clear set-up of the measurements? The mode RHI is mentioned, but later in section 3.3.1 the retrieval of the horizontal wind using the PPI mode is discussed. Therefore, it is not clear to the reader what the measurement sequence is: a combination of RHI and PPI? Also, the rotation speed of the radar for RHI and PPI measurements should be provided.

Table 1: add the Doppler velocity resolution for both MIRA-35.

**AR2-2:** The requested information, including the PPI scan, the scanning speeds for both RHI and PPI, and the Doppler velocity resolution has been added to the text and table.

3) Mira-35 radar in hybrid mode

**R2C3:**

Line 116: "....as a function of Doppler velocity • ..." My suggestion: either "....as a function of Doppler angular frequency ..." or "....as a function of Doppler velocity v ..."

**AR2-3:** Since v is used for multiple velocity-related parameters, we have chosen to use  $\omega$  for Doppler velocity.

**R2C4:**

**Lines 116-117: No point above the capital letter for Eh and Ev?**

**AR2-4:** The dot above Eh and Ev denotes that these parameters are complex-valued. This comment is identical to Reviewer #1's R1C7; please refer to our response **AR1-7** there for further clarification.

**R2C5:**

Equations (1) and (2): Equations (1) and (2) are not correct. The reflectivity is not directly equal to the average modulus square of received complex amplitudes. A constant is missing.

**AR2-5:** We added C1 and C2 in equations of 1 and 2. Also, added this text in lines 153-155: Also, The constants C1 and C2 depend on the radar system parameters such as transmitted power, Radar cross-section (RCS), radar geometry, wavelength of the radar signal, and system gain

**R2C6:**

Line 121: "ZDR quantifies the difference between reflectivity measurements in horizontal (Zhh, Eq. 1) and vertical (Zvv, Eq. 2) polarizations, expressed in decibels (dB) (Eq. 3)." Eqs. 1-3 are not expressed in decibels. Be consistent with the text and equations.

**AR2-6:** We have revised the text to clarify the definitions in both linear and dB units (line 155).

**R2C7:**

Line 125: "At zenith-pointing direction, ZDR is zero. At slant-pointing direction, a positive ZDR value ...."

AR2-7: We modified the text as requested by the Reviewer (lines 155-156).

**R2C8:**

Line 127: "The correlation coefficient (RHV) is a crucial parameter that quantifies the linear relationship between the Zhh and Zvv." Rephrase this statement, which is now not correct.

**AR2-8:** We rephrased it like this: The correlation coefficient (RHV) is a crucial polarimetric parameter that quantifies the similarity between horizontally and vertically polarized backscattered signals. It provides insight into the diversity of particle shapes and orientations within a radar resolution volume.

**R2C9:**

Lines 128-129: the sentence is not clear and that is not useful to describe Eq. 4 in terms of ratio, sum, square root, product... because that can be directly seen in Eq. 4.

**AR2-9:** Thank you for the suggestion. We have removed the explanation accordingly.

**R2C10:**

Line 131: remove the point after 1.

**AR2-10: Removed.**

**R2C11:**

Lines 131-132: "... a correlation coefficient of 1 indicates perfect correlation or alignment between horizontal and vertical polarizations, suggesting consistent scattering behavior." Rephrase. What is "alignment between horizontal and vertical polarizations"? What is "consistent scattering behavior"?

**AR2-11:** we rephrased the text like this: A correlation coefficient of 1 indicates perfect correlation between horizontal and vertical polarizations, implying that the backscattered signals are identical in phase and amplitude. This suggests consistent scattering behavior, often associated with isotropic scatterers such as spherical particles.

**R2C12:**

Line 135: ".... raindrops, with a spherical shape and ....". Replace "spherical" by "spheroidal".

AR2-12: We replaced "spherical" with "Slightly spheroidal."

**R2C13:**

Line 137: ".... a parameter frequently detected by cloud radars ....". Rephrase. A parameter is not detected.

AR2-13: We replaced "measured" with "detected."

**4) Main-peak approach**

**R2C14:** Is the main peak approach code by Myagkov et al. available online?

**AR2-14:** The central retrieval code is still owned by Alexander Myagkov (as deployed by Myagkov et al., 2016a) and is available upon request.

**R2C15:**

Line 163: "This analysis provides insights into particle habits by utilizing a spheroidal scattering model". "A spheroidal scattering model". Which scattering model is used?

and "spheroidal scattering model" is not the appropriate name. Provide the equations of the polarizability ratio and degree of orientation. Explain how they relate to the ZDR and RHV measurements.

**AR2-15:** These comments and questions have already been addressed in our response to the first reviewer (AR1-2).

**5) Spectrally resolved approach**

**R1C16:**

I recommend to the authors the extension of the block diagram of Figure 2, where the main peak approach block would appear. Further a zoom of the main peak block, with inputs and outputs, can be worked out in a second Figure. Presently, without reading in detail the papers Myagkov et al., it is challenging to understand the spectrally resolved technique. The reader should be able to understand the paper without having to read preceding papers.

**AR2-16:** We understand the concerns of the reviewer that the technique shall be better introduced again in our manuscript. But we have to emphasize that the actual retrieval technique requires a very extensive introduction, as it was done by Myagkov et al., 2016a. We therefore kindly ask the Reviewer to accept that we introduce the basic retrieval only briefly. We added a short statement to section 3. where we emphasize the importance of the work of Myagkov et al., (2016) for a full introduction of the (main-peak) shape retrieval technique. Nevertheless, as w also requested by Reviewer #1, we extended the introduction of the general retrieval in Section 3.1 to provide the reader a more detailed introduction.

**R2C17:**

There is no information on the error analysis.

**AR2-17:** In the spectrally resolved approach the same error calculation is used as for the main-peak approach. The values of polarizability ratio and degree of orientation where the best fit between simulated and observed ZDR and RHV are identified are selected as the result. We treat the error discussion similar as was done by Myagkov et al (2016b). There, they focused on visualization of the 2 times the standard deviation of the retrieved values of polarizability ratio and degree of orientation, respectively, in a range gate. This standard deviation results from the variability of the ZDR and RHV data points that are incorporated into each range gate and elevation angle interval used in the retrieval.

**R2C18:**

**How is Mie scattering regime accounted for? For example, for Part 1.**

**AR2-18:** This point has already been addressed in our response AR1-2 to the first reviewer.

**R2C19:**

Line 185: "Consequently, the Doppler spectra observed with a vertically pointing cloud radar offer insights into the variability of sizes and shapes of the ice particles". Information on the shapes of the ice particles for zenith-pointing cloud radar cannot really be obtained.

**AR2-19:** Doppler spectra from a vertically pointing cloud radar provide insights into the variability of ice particle sizes and shapes but do not allow for their direct quantification.

**R2C20:**

Lines 198-199: "The amount of 5 parts was empirically chosen for this study, because usually not more than that amount of different particle shapes can be expected in a cloud volume". Can you provide a reference for this statement?

**AR2-20:** This point has already been addressed in our response to the first reviewer (AR1-1).

**R2C21:**

Lines 199-200: "Increasing the number of parts would result in a reduced amount of available data points per Doppler spectrum part which would lead to increased uncertainties. This statement should be developed. "Which amount of data points for the spectrally resolved approach is recommended? Why? I missed a discussion on this point in terms of possible errors.

**AR2-21:** It depends on the number of FFT points. In this study (FFT = 256), we required at least one data point per part. For FFT = 256, we recommend using two data points per part.

**R2C22:**

Lines 203-204: "Instead, we assume that the fall attitude of the individual hydrometeor types contained in the cloud volume is similar at all elevation angles." Was the same assumption made in the main peak approach?

**AR2-22:** In the main peak approach, fall velocity is not considered at all.

**6) Retrieval of horizontal wind**

**R2C23:**

Figure 3: compared to Vf and Vh, VR is not well scaled. Correct this.

**AR2-23:** We have updated the figure and the included parameters.

R2C24:

Lines 241-242: ".... while the sine's curve amplitude yields the wind velocity Vh multiplied by the cosine of the elevation angle, "

**AR2-24:** Thanks for the hint. We corrected the passage.**

**R2C25:**

Lines 242-243: "Additionally, the entire curve's displacement from the zero velocity relates to the precipitation fall speed."

**AR2-25**: Thanks for the hint. We corrected the passage.**

**R2C26:**

Lines 243-244: "We used the approach of Baars et al. (2023) to derive the horizontal wind components." Describe shortly this approach.

**AR2-26:** We added a short introduction to the technique deployed by Baars et al., (2023) to the last paragraph of Section 2.3.

**7) Aliasing problems and effects of horizontal winds on the determination of the vertical velocity component.**

**R2C27:** Line 257: mention what fn is.**

**AR**2-27: We mentioned fn is pulse repetition frequency.

**R2C28:**

Lines 267-270: The methodology of dealiasing needs to be shortly extended for clarity and reproducibility.

**AR2-28:** In agreement to a request of Reviewer #1 we slightly updated the passage on the aliasing problem correction.

**R2C29:**

Eq. 10: VR should be replaced by VD.

**AR2-29**: We modified all of the variables in this section.

**8) First case study 07 Nov 2014, 09:15-09:30: retrieval of various hydrometeor types**

**R2C30:**

Figure 4 caption: ...... on November 7, 2014. Correct the date.

**AR2-30:** We updated the date from November 3 to November 7.**

**R2C31:**

Lines 293-294: Between 09:15 and 09:30 UTC, a deep cloud .... which caused precipitation after 09 UTC. Check the time consistency. If it rains from 09:00 UTC, it means that the deep cloud is present before 09:15. Rephrase.

**AR2-31:** The first part of the sentence addresses the time period of the case study which will be introduced in more detail in the remainder of the section. The information about the onset of the precipitation is unrelated to the introduction of the case study period. We slightly updated the sentence and now state "...which caused slight precipitation already since 09:00 UTC."

**R2C32:**

Line 295: The evolution of the mixed phase in this deep cloud.... Why is this deep cloud a mixed-phase cloud? I miss the argumentation here.

**AR2-32:** We identified the deep cloud as a mixed-phase cloud as we were able to identify some presence of liquid water in its lower parts. This is now also mentioned in the text that introduces Figure 4.

**R2C33:**

Line 335: .... The SNR stabilizes at approximately 60 dB. I think it is much less. 25 dB?

**AR2-33:** Yes, thanks. We updated from 60 dB to 25 dB.

**R2C34:**

Figure 6 caption: there are errors in the sequence (a)-(o): .... (l) RHV in part 2.....(i) ZDR, and (n) RHV in part 4, .....

**AR2-34:** Yes, thanks. We updated i and f with I and i respectively.

**R2C35:**

Lines 345-347: Why is it possible to conclude that below 4 km based on RHV and ZDR the particles are prolate (part 5). Provide a short explanation and reference.

**AR2-35:** Based on the simulation, for prolate-shaped particles, ZDR is higher at non-zenith angles than at the zenith angle, while RHV is lower at the zenith angle compared to non-zenith angles.

**R2C36:**

Lines 358-361: provide a reference.

**AR2-36:** Similar to the previous comment, the conclusions are based on the simulation.

**R2C37:**

Figure 7 caption: error in the sequence (a)-(l): .... (g) RHV spectrum before splitting....

**AR2-37:** Yes, thanks. We updated h with g.**

**R2C38:**

Lines 362-372: in this paragraph the retrieved polarizability ratio shown in Figure 8 is discussed. However, there is no word about the retrieved degree of orientation, part of Figure 8 as well. Why?

**AR2-38:** We added this text in lines 476-479: At an altitude of 3 km, the degree of orientation is at its lowest (around -0.75), suggesting that prolate-shaped particles are more horizontally aligned compared to other altitudes. Below 3 km, as the polarizability ratio decreases, the degree of orientation approaches 0, indicating a transition to randomly oriented prolate-shaped particles.

**R2C39:**

Line 363: "For the sake of readability, error bars are omitted in this case ...." OK, but some text related to the error bars should be written in section 3.2. How are the error bars estimated?

**AR2-39:** We kindly refer to our reply AR2-17, where we state how the uncertainties are determined and where they are introduced in the manuscript text.

9) Second case study 03 Nov 2014, 20:30-20:45: Secondary ice Formation

**R2C40:**

Figure 9 caption: The highlighted period.... are applied. Rephrase the sentence. Also, I don't see the highlighted period in the figure.

**AR2-40:** We now illustrated the highlighted period in the figure.

**R2C41:**

Figure 13 caption: there are errors in the sequence (a)-(o): .... (I) RHV in part 2.....(i) ZDR, and (n) RHV in part 4, .....

**AR2-41:** Yes, Thanks. We updated i and f with I and i respectively.

**R2C42:**

Figure 14 caption: error in the sequence (a)-(l): .... (g) RHV spectrum before splitting....

**AR2-42:** Yes, Thanks. We updated h with g.

**R2C43:**

Figure 15 caption: error in the sequence (a)-(l): .... (g) RHV spectrum before splitting....

**AR2-43:** Yes, Thanks. We updated h with g.

**R2C44:**

Lines 462-463: "Indications are given that the branches of oblate ice crystals, such as dendrites fell off, in addition....". Can the authors clarify this statement? Which indications? Is the presence of dendrites in the study case justified? Until now, there was no discussion about the presence of dendrites....

**AR2-44:** We would like to point to a previous reply AR#46 that addresses a similar question raised by Reviewer #1.

**10) References**

**R2C45:** The authors should review the reference list. For example, uncomplete reference: Melnikov and Sraka, 2013. Spell-check: Hajipour, M et al. 2024: ....studies.....

**AR2-45:** We screened and modified the reference list as requested.

**References:**

Chellini, G., & Kneifel, S. (2024). Turbulence as a key driver of ice aggregation and riming in Arctic low-level mixed-phase clouds, revealed by long-term cloud radar observations. *Geophysical Research Letters*, 51, e2023GL106599. https://doi.org/10.1029/2023GL106599

Kneifel, S., Kulie, M. S., and Bennartz, R. (2011), A triple-frequency approach to retrieve microphysical snowfall parameters, *J. Geophys. Res.*, 116, D11203, doi:10.1029/2010JD015430.

Leinonen, J., D. Moisseev, and T. Nousiainen (2013), Correction to "Linking snowflake microstructure to multi-frequency radar observations", J. Geophys. Res. Atmos., 118, 6708–6709, doi:10.1002/jgrd.50531.

Myagkov, A., Seifert, P., Bauer-Pfundstein, M., and Wandinger, U.: Cloud radar with hybrid mode towards estimation of shape and orientation of ice crystals, Atmos. Meas. Tech., 9, 469–489, https://doi.org/10.5194/amt-9-469-2016, 2016.

Matrosov, 2021, https://doi.org/10.1175/JTECH-D-20-0138.1

Ryzhkov, A., and J. Krause, 2022: New Polarimetric Radar Algorithm for Melting-Layer Detection and Determination of Its Height. *J. Atmos. Oceanic Technol.*, **39**, 529–543, https://doi.org/10.1175/JTECH-D-21-0130.1.

Teisseire, A., Billault-Roux, A.-C., Vogl, T., and Seifert, P.: Attribution of riming and aggregation processes by application of the vertical distribution of particle shape (VDPS) and spectral retrieval techniques to cloud radar observations, EGUsphere [preprint], https://doi.org/10.5194/egusphere-2024-2711, 2024.

von Terzi, L., Dias Neto, J., Ori, D., Myagkov, A., and Kneifel, S.: Ice microphysical processes in the dendritic growth layer: a statistical analysis combining multi-frequency and polarimetric Doppler cloud radar observations, Atmos. Chem. Phys., 22, 11795–11821, https://doi.org/10.5194/acp-22-11795-2022, 2022.

---

## Referee Report (RR1)

Review of Identification of multiple co-located hydrometeor types in Doppler spectra from scanning polarimetric cloud radar observations by Majid Hajipour, Patric Seifert, Hannes Griesche, Kevin Ohneiser, and Martin Radenz for AMT.

In this article, the authors build upon the Main-peak approach introduced by Myagkov et al., extending it to encompass the full Doppler spectrum. This extension allows for a spectrally resolved methodology. The topic is relevant, and the two case studies presented are engaging. The revised manuscript shows clear improvement, and the authors have briefly addressed the Main-peak approach as requested by the referees. I appreciate the thoughtful responses to the previous comments and questions.

However, the manuscript still requires further revision for publication. A list of comments, corrections, and suggested additions is provided below, including both substantive and minor points. Given the number of small inconsistencies and typographical errors, I recommend a careful and thorough proofreading of the revised version. At this stage, the manuscript falls between requiring major and minor revisions.

**Comments/corrections/adding's**

**1) Separation Rayleigh/Resonance Scattering Regime**

Based on the measurement of the spectral differential phase, it is possible to detect where the resonance scattering regime starts in the Doppler spectrum (Mak et Unal, 2025). That can be of use for the analysis of large ice particles (part 1).

Mak, H. Y. L., and Unal, C. (2025): Peering into the heart of thunderstorm clouds: insights from cloud radar and spectral polarimetry. Atmospheric Measurement Techniques, 18(5), 1209–1242. https://doi.org/10.5194/amt-18-1209-2025

**2) Instrumentation**

Lines 103-105: To explore the polarimetric capabilities of the MIRA-35 cloud radar was not the unique goal of the ACCEPT campaign. Therefore, I propose to rephrase:

"The ACCEPT measurement campaign was led by the Leibniz Institute for Tropospheric Research (TROPOS) in Leipzig, Germany, with partners from the Technical University of Delft and METEK GmbH in Elmshorn, Germany. One of its goals was to explore the polarimetric capabilities of the MIRA-35 cloud radar. A second objective was the study of ice particle growth processes in mixed-phase clouds using spectral polarimetric S-band radar measurements (Pfitzenmaier et al. 2017, Pfitzenmaier et al. 2018).

Pfitzenmaier, L., Dufournet, Y., Unal, C. M. H., & Russchenberg, H. W. J. (2017). Retrieving fall streaks within cloud systems using Doppler radar. Journal of Atmospheric and Oceanic Technology, 34(4), 905-920. https://doi.org/10.1175/JTECH-D-16-0117.1

Pfitzenmaier L., C. M. H. Unal, Y. Dufournet, and H. W. J. Russchenberg: Observing ice particle growth along fall streaks in mixed-phase clouds using spectral polarimetric radar data, Atmos. Chem. Phys., 18, 7843–7862, 2018. https://doi.org/10.5194/acp-18-7843-2018.

**3) Main-peak approach**

Line 192: "These measurements are then compared with simulated values based on spheroid model that assumes the particles are shaped like spheroids (3D ellipsoids)". Mention already here that Rayleigh scattering is used. The reader should not wait for the conclusion to know this important information.

**4) Spectrally resolved approach**

Figure 1: this figure suggests that 5 different types of particles can be retrieved with the Doppler spectrum partitioning, but that is not the case. Make a note of this in Figure 1 caption.

**5) Second case study 03 Nov 2014, 20:30-20:45: Secondary ice formation**

4-a) Figure 10 is not discussed (only 1 sentence, Lines 523-524). I suggest to add a small paragraph to comment the key features of the precipitating cloud case illustrated in Figure 10 (20:00-21:00).

4-b) Lines 538-540: The polarizability factor and degree of orientation cannot be discussed in the melting layer and in rain using the Rayleigh scattering assumption, which is not valid at 35 GHz for these hydrometeors. Remove these lines.

4-c) Lines 588-590: "Indications are given that the branches of oblate ice crystals, such as dendrites fell off, in addition....". Can the authors clarify this statement? Which indications? Is the presence of dendrites in the study case justified? Until now, there was no discussion about the presence of dendrites....

The authors cannot write such a statement without any justification other than "Indications are given". Justify/explain in the article.

**6) Summary and Conclusions**

5-a) Lines 609-611: "The author notes that the possible reason for the smaller influence of non-Rayleigh scattering on polarimetric variables is that they are differential (rather than absolute) quantities representing differences/ratios of radar parameters at two orthogonal polarizations."

Considering any mm-wavelength radar or hydrometeor type or polarimetric variable, this statement is not correct. To my knowledge, in general, non-Rayleigh scattering influences polarimetric variables.

5-b) Lines 618-620: "In the presented study, the homogeneity was evaluated based on (1) inspection of the required monotonic relationships between polarimetric parameters and elevation angle, and (2) the appropriateness of the horizontal wind correction."

Significant variations in Doppler spectrum width were not considered?

5-c) Lines 621-622: "The ACCEPT campaign, conducted in Cabauw, the Netherlands in 2014, aimed to assess the capabilities of both the main peak approach and the spectrally resolved approach."

Based on earlier comment (see instrumentation section), replace the sentence by "One of the aims of the ACCEPT campaign, conducted in Cabauw, the Netherlands in 2014, was to assess the capabilities of both the main peak approach and the spectrally resolved approach."

**7) Acknowledgements**

Line 653: "….. and to the team of the CESAR Observatory in Cabauw and Delft University of Technology, NL, for their support …."

**Minor comments/corrections/adding's**

**1) Abstract**

The abstract consists of three parts identical. Please, correct this.

**2) Introduction**

Lines 87-89: The following paper could be added:

Mak, H. Y. L., and Unal, C. (2025): Peering into the heart of thunderstorm clouds: insights from cloud radar and spectral polarimetry. Atmospheric Measurement Techniques, 18(5), 1209–1242. https://doi.org/10.5194/amt-18-1209-2025

**3) Instrumentation**

3-a) Line 114: remove "also"

3-b) End of Table 1, typo: "…… can be estimated by assuming …."

**4) Mira-35 radar in hybrid mode**

4-a) Lines 144-145: "……. which are representative of the horizontally ($E_h(\omega)$) and vertically ($E_v(\omega)$) components of the received waves." Also, place a dot above the capital letter $E$. These components are complex.

4-b) Line 148: "ZDR quantifies the ratio of reflectivity measurements in horizontal (Zhh, Eq. 1) and vertical (Zvv, Eq. 2) polarizations, (Eq. 3)."

4-c) Line 153, typo: Also, the constants …..

4-d) Line 154: Remove "Radar cross section (RCS)". The constants C1 and C2 do not depend on the RCS, $E_h(\omega)$ and $E_v(\omega)$ do depend on the RCS (square-root of RCS).

4-e) Lines 158-160: "The correlation coefficient (RHV) is a crucial parameter (defined by Eq. 4) that quantifies the similarity between horizontally ($E_h(\omega)$) and vertically ($E_v(\omega)$) polarized backscattered signals." Place a dot above the capital letter $E$.

4-f) Lines 160-162: "It provides insight into the diversity of particle shapes and orientations within a radar resolution volume and is typically expressed as a value between 0 and 1."

4-g) Lines 162-163: "A correlation coefficient of 1 indicates perfect correlation between horizontal and vertical polarization signals, implying that the backscattered signals are identical in phase and amplitude."

**5) Main-peak approach**

5-a) In Eq. (8), remove the subscript "e", which is not present in the text.

5-b) Line 211: Is $\xi_g$ the axis ratio?

5-c) Line 226: "Finally, to derive ZDR and RHV, the horizontal and vertical complex scattering amplitudes are calculated using the polarizability ratio and the degree of orientation across different elevation angles. "

5-d) Line 234, typo: "distribution".

5-e) Line 238, typo: Not "than" but "then".

**6) Spectrally resolved approach**

6-a) Lines 250-253: "The width of the Doppler spectra is thereby characterized by size- and shape-dependent fall velocities of the particles, which are super-imposed by influences of turbulence and (predominantly in case of off-zenith antenna pointing angle) horizontal wind variability that cause additional broadening of the spectrum (Radenz et al., 2019)."

6-b) Line 274, typo: "mean square error function"

6-c) Line 279: "During averaging of all ZDR and RHV values …..". Mention how many values can be expected for this average.

**7) The influence of air motion on the Doppler spectra observed by a scanning cloud radar**

Line 295: …. onto the gravitational downward motion of particles …..

**8) Retrieval of horizontal wind**

8-a) Line 306: "…. and radial wind velocity Vw."

8-b) Eq. (12): is it not $V_w = V_h \cos(\beta \pm \pi - \alpha) \cos(\psi)$? To have negative radial wind velocity when the particles approach the radar. The wind direction being defined from where the wind comes from.

**9) First case study 07 Nov 2014, 09:15-09:30: retrieval of various hydrometeor types**

9-a) Figure 4 caption: ....... (see Figs. 7-9). Typo in the right parenthesis.

9-b) Line 391, typo: .... high values of radar LDR ....

9-c) Line 444: .... The SNR stabilizes at approximately 60 dB. I think it is much less. 25 dB?

9-d) Lines 457-458: "Since different parts might identify oblate-shaped particles at heights around 3km (parts 1, 2, 3, and 4) and prolate-shape particles at part 5, a height of 3km is chosen for a detailed depiction of the data analysis procedure."

From the discussion of Figure 7 above, the authors mention that at this height (around 3 km), the particles are spherical-shaped in parts 1-3, oblate-shaped in part 4 and prolate-shaped in part 5. Therefore, rephrase the sentence for consistency with the discussion of Figure 7.

9-e) Figure 8 caption: I suppose that the time period 09:16:09 – 09:18:08 UTC is considered and not only the time 09:15:09 UTC.

9-f) Figure 8 caption: ..... and for each part of the spectrally resolved spectra, shape and orientation retrieval ......"

9-g) Figure 9: I don't see the horizontal lines displaying the temperature levels.

**10) Second case study 03 Nov 2014, 20:30-20:45: Secondary ice formation**

10-a) Lines 528-529: "In middle altitudes (X-Y km) ........ In lower altitudes (Z-X km) ....." Be specific and indicate the height range considered.

10-b) Figure 12 caption: the time (20:01:09-20:11:26) is not correct.

10-c) Line 542: ".... between the melting layer and 2.75 km height ......"

10-d) Line 558: "... RHV decreases with increasing part number ...."

10-e) Figures 16 and 17 caption: ..... and for each part of the spectrally resolved spectra, shape and orientation retrieval ......"

**11) Summary and Conclusions**

Lines 602-603: "In particular, by incorporating Rayleigh scattering, we assume that particles are small with respect to the deployed electromagnetic waves." I suggest to

remove this sentence. It is redundant compared to the next sentence. The next sentence suffices.

Line 611, typo: .... were elaborated .....

Line 638:" ........ would be to apply implement techniques ....." Suppress either "apply" or "implement".

Line 639, typo: "Then more insights ....."

---

## Referee Report (RR2)

Thanks for addressing my comments, I think the manuscript has improved! Please see the comments below and address them before a possible publication!

**Answer to review comments:**

Answer to AR1-1B: I am still not convinced that just dividing the spectrum into 5 parts allows you to see 5 distinct hydrometeor types. It can very likely be that two hydrometeors (i.e. needles and aggregates as they can have the same fall velocity) are in the same part, therefore you will still see the combined effect of both species. I understand that applying a peak identification algorithm on the slant spectra is difficult, however, I would have tested a few more subjective ways of separation into n parts based on i.e. the shape of your Doppler spectrum (i.e. the slope of sZDR can tell you when another species is introduced, or some other measure). The arbitrary selection of 5, especially if you can not even distinguish between rimed particles and aggregates seems random. However, I am fine with having it published like this, as long as you make it clear that separating into these 5 parts does not mean that you do not have a mixture of ice particles in each part and can therefore miss e.g. the appearance of new species.

Answer to AR1-2: to me it was not clear that if you assume Rayleigh that then there is a direct relationship between the ZDR/RHV and the shape of the particle. Perhaps you can say that in one sentence in the manuscript. About the wind retrieval: See comment to Line 240

Line 240: if horizontal wind indicates inhomogeneity, then correcting for the wind does not solve that problem. Or are you using the wind field in some other way? If so please explain that in more detail. In your comment AR1-2 you say that you use the wind field as a measure of homogeneity, how do you do that? What are the criteria then that say the field is homogeneous enough to be considered for the retrieval? You further said in your comment AR1-4 that "moving the Doppler spectra to 0 does not have the same effect" how is that possible? With the wind correction you are doing exactly that: shifting the Doppler spectrum by an arbitrary number. The elevation dependency of the Doppler spectrum width has nothing to do with the horizontal wind, but with the viewing geometry of the radar. In fact, looking at our own data, the spectral width is exactly the same if viewed at e.g. 30° or 90° elevation without correcting for the wind. The width of the spectrum is influenced by wind shear and turbulence, not the wind speed. In comment AR1-4 you further say that you are using the Doppler spectrum width as a measure for homogeneity. This is not the same as the answer in AR1-2, where you say you use the wind field. Which of the two are you using and how exactly?

**Specific comments:**

Line 186: please define vdsp

Line 259: I would name the 5 shapes here specifically, so that the reader knows what you want to distinguish

Line 284: I still don't understand the word harmonize, or the sentence behind it, this was not made clear to me in AR1-4! Please elaborate on that further! First: what does harmonize mean. Second: why do you need to do that for retrieving the wind field?

Figure 1 and comment AR1-26: your statement that turbulence is represented by an upward motion of particles is wrong. Turbulence can go upwards and downwards. Folded with the particles movements, it broadens the spectrum. Most of the times this does not even result in an actual upward motion of your particles, so indicating turbulence in Figure 1 as the distance between the 0m/s and the slow edge is wrong. Further, turbulence flattens the spectrum. So if you want to include turbulence in the figure please include all aspects correctly or just don't put it in the figure!

Fig 9 is different from the preprint, why? Now you corrected line 474, but now it is clearly wrong again, if you take your new figure. Now I would say that part 1,5 have a pol. Ratio of 0.75 and part 2,3,4 of 1.25 (also clearly not 0.9 and 1.1 if you look at the figure)

Perhaps you can explain to me again why in case study 1 you come to the conclusion that you have no SIP and in case study 2 you say you have SIP. Just because you could not see a liquid layer in which ice crystals might have been nucleated does not mean it is not there, especially since you clearly have riming in case study 2. So your argument for no SIP in case study 1 can also be given here for case study 2.

Line 603-605: I don't understand this sentence! You assume Rayleigh, therefore your particles have to be small with respect to the wavelength. This does not mean that you can accurately model the scattering properties of all ice particles. The statement that it "ensures the proper treatment of scattering properties which is critical for precise radar measurements and interpretations" does not make sense to me. How does your assumption of Rayleigh ensure that? Anyway, maybe rephrasing that helps, or just leaving the entire sentence out, as all the important things are already said with the sentence before.

Line 611-614: I don't understand what you mean with "this was not applied yet in the present study"? I thought you separated into 5 parts already?

Line 615: What do the non-Rayleigh effects have to do with horizontal homogeneity?

Line 620: what does "appropriateness of horizontal wind correction" mean? Please be more specific with your methods!

---

## Author Response (AR2)

**Responses to Review Comments on "Identification of multiple co-located hydrometeor types in Doppler Spectra from scanning polarimetric cloud radar observations"**

Majid Hajipour[1], Patric Seifert[1], Hannes Griesche[1], Kevin Ohneiser[1], and Martin Radenz[1]

1. Leibniz institute for tropospheric research, Leipzig, Germany

5 July 2025

Dear Editor,

we would like to sincerely thank the reviewers for their careful reading and constructive comments during both rounds of review. Their insights have helped us to improve the clarity and quality of the manuscript.

Below we provide replies to the individual comments raised by both reviewers. We put efforts into explaining more clearly about the distinction into 5 parts, the application of the horizontal-wind correction and the impact of the assumption of Rayleigh scattering. Also any proposed additional interpretations of the case-study observations were considered in our revision. The manuscript was re-screened for typographical and grammar issues.

Any references to line numbers we do provide correspond to the revised version of the manuscript (without markup).

**Response to Report #1**

**1) Separation Rayleigh/Resonance Scattering Regime**

**Based on the measurement of the spectral differential phase, it is possible to detect where the resonance scattering regime starts in the Doppler spectrum (Mak et Unal, 2025). That can be of use for the analysis of large ice particles (part 1).**

**Mak, H. Y. L., and Unal, C. (2025): Peering into the heart of thunderstorm clouds: insights from cloud radar and spectral polarimetry. Atmospheric Measurement Techniques, 18(5), 1209–1242. https://doi.org/10.5194/amt-18-1209-2025**

We agree that the use of spectral differential phase, as discussed by Mak and Unal (2025), can provide additional insights into the resonance scattering regime within the Doppler spectrum. In particular, this could be beneficial for characterizing large ice particles represented in part 1 of our spectrally resolved analysis. While our current study focuses on ZDR and RHV as key polarimetric parameters, the incorporation of spectral differential phase could indeed enhance the detection and interpretation of resonance effects associated with large particles. We suggest this as a promising direction for future studies that aim to refine particle-type identification within

our framework. We thus added the reference and short note on the study in the conclusions section in Lines 605-607: "A recent study by Mak and Unal (2025) suggests to use spectral differential phase for identification of conditions of non-Rayleigh scattering conditions. As the cloud radar system under study in STSR mode can in principle also provide differential phase, this approach might be evaluated further in follow-up studies."

**2) Instrumentation**

**Lines 103-105: To explore the polarimetric capabilities of the MIRA-35 cloud radar was not the unique goal of the ACCEPT campaign. Therefore, I propose to rephrase: "The ACCEPT measurement campaign was led by the Leibniz Institute for Tropospheric Research (TROPOS) in Leipzig, Germany, with partners from the Technical University of Delft and METEK GmbH in Elmshorn, Germany. One of its goals was to explore the polarimetric capabilities of the MIRA-35 cloud radar. A second objective was the study of ice particle growth processes in mixed-phase clouds using spectral polarimetric S- band radar measurements (Pfitzenmaier et al. 2017, Pfitzenmaier et al. 2018). Pfitzenmaier, L., Dufournet, Y., Unal, C. M. H., & Russchenberg, H. W. J. (2017). Retrieving fall streaks within cloud systems using Doppler radar. Journal of Atmospheric and Oceanic Technology, 34(4), 905-920. https://doi.org/10.1175/JTECH-D-16-0117.1**

**Pfitzenmaier L., C. M. H. Unal, Y. Dufournet, and H. W. J. Russchenberg: Observing ice particle growth along fall streaks in mixed-phase clouds using spectral polarimetric radar data, Atmos. Chem. Phys., 18, 7843–7862, 2018. https://doi.org/10.5194/acp-18-7843-2018.**

We have rephrased the sentence as proposed to reflect the broader goals of the ACCEPT campaign, including the study of ice particle growth using spectral polarimetric S-band radar measurements. The text in lines 77-80 now reads: "The ACCEPT measurement campaign was led by the Leibniz Institute for Tropospheric Research (TROPOS) in Leipzig, Germany, with partners from the Technical University of Delft, the Netherlands, and METEK GmbH in Elmshorn, Germany. One of its goals was to explore the polarimetric capabilities of the MIRA-35 cloud radar. A second objective was the study of ice particle growth processes in mixed-phase clouds using spectral polarimetric S- band radar measurements (Pfitzenmaier et al., 2017, 2018)."

**3) Main-peak approach**

**Line 192: "These measurements are then compared with simulated values based on spheroid model that assumes the particles are shaped like spheroids (3D ellipsoids)". Mention already here that Rayleigh scattering is used. The reader should not wait for the conclusion to know this important information.**

We now mention at this point in the text that the simulations are based on the Rayleigh scattering assumption. For this we have added lines 166-168 as follows: "These measurements are then compared with simulated values based on a spheroid model that assumes the particles are shaped like spheroids (3D ellipsoids) and that Rayleigh scattering applies."

**4) Spectrally resolved approach**

**Figure 1: this figure suggests that 5 different types of particles can be retrieved with the Doppler spectrum partitioning, but that is not the case. Make a note of this in Figure 1 caption.**

We have revised the caption of Figure 1 to clarify that the division into five parts enables the potential identification of up to five different hydrometeor types, but does not guarantee nor require their presence in every case. The updated caption now reads: "The division into five parts enables the potential identification of up to five different hydrometeor types; however, the actual number of distinct types retrieved depends on the spectral and microphysical characteristics of the observed cloud. It should be noted that turbulent motion is not considered in the sketch."

**5) Second case study 03 Nov 2014, 20:30-20:45: Secondary ice formation**

**4-a) Figure 10 is not discussed (only 1 sentence, Lines 523-524). I suggest to add a small paragraph to comment the key features of the precipitating cloud case illustrated in Figure 10 (20:00-21:00).**

**5-a)** We have added a paragraph discussing the key features of the precipitating cloud case illustrated in Figure 10, focusing on reflectivity, LDR, Doppler velocity, and spectral width during the 20:00–21:00 UTC period. See lines 501-507 in the revised manuscript as follows: "Figure 10 provides an overview of the precipitating cloud system observed from 20:00 to 21:00 UTC. Radar reflectivity (panel a) shows strong backscatter values, especially below 4 km, indicating precipitation reaching the surface. The LDR (panel b) increases near the melting layer (2 km), suggesting the presence of melting particles and possibly needle-like ice just above it. Additionally, LDR values around −25 dB at the cloud top as observed by vertical-stare Mira-35 NMRA indicate the presence of slightly non-isometric-shaped particles. The Doppler velocity (panel c) reveals strong downward motion, with values exceeding −1.5 m s−1 below 3 km, characteristic of precipitating particles. The spectral width (panel d) is elevated in the lower levels, indicating enhanced turbulence and possibly a mixture of hydrometeor types during active precipitation."

**5-b) Lines 538-540: The polarizability factor and degree of orientation cannot be discussed in the melting layer and in rain using the Rayleigh scattering assumption, which is not valid at 35 GHz for these hydrometeors. Remove these lines.**

**5-b)** We agree that the Rayleigh assumption is not valid for melting layer and rain at 35 GHz. We have removed the lines accordingly.

**5-c) Lines 588-590: "Indications are given that the branches of oblate ice crystals, such as dendrites fell off, in addition….". Can the authors clarify this statement? Which indications? Is the presence of dendrites in the study case justified? Until now, there was no discussion about the presence of dendrites…. The authors cannot write such a statement without any justification other than "Indications are given". Justify/explain in the article.**

**5-c)** The indication refers to the observed changes in polarimetric radar signatures consistent with fragmentation of dendritic ice crystals during their descent. We realize that the presence of dendrites was not explicitly discussed earlier, so we have now added a brief explanation and justification for their occurrence in the study case (lines 560-582), based on environmental conditions and previous literature.

**6) Summary and Conclusions**

**6-a) Lines 609-611: "The author notes that the possible reason for the smaller influence of non-Rayleigh scattering on polarimetric variables is that they are differential (rather than absolute) quantities representing differences/ratios of radar parameters at two orthogonal polarizations." Considering any mm-wavelength radar or hydrometeor type or polarimetric variable, this statement is not correct. To my knowledge, in general, non-Rayleigh scattering influences polarimetric variables.**

**6-a)** Yes, non-Rayleigh scattering influences polarimetric variables. This is true and was shown in many earlier studies. What we, or better Matrosov (2021) means, is that the impact of a particle size distribution on the bulk polarimetric signatures somewhat cancels out. It is especially a lack of current T-Matrix and DDA-based simulation studies that they can hardly provide simulations of a realistic particle population. That's why we argue that simulations might overestimate the non-Rayleigh effect on polarimetric variables.

**6-b) Lines 618-620: "In the presented study, the homogeneity was evaluated based on (1) inspection of the required monotonic relationships between polarimetric parameters and elevation angle, and (2) the appropriateness of the horizontal wind correction." Significant variations in Doppler spectrum width were not considered?**

**6-b)** We acknowledge that significant variations in Doppler spectrum width can also indicate inhomogeneity. In our study, we focused on monotonic relationships and wind correction as primary criteria, but we have now added this statement to lines 613-614 as follows: Additionally, variations in Doppler spectrum width can be considered as a potential indicator of inhomogeneity, though they were not the primary focus in the current evaluation"

**6-c) Lines 621-622: "The ACCEPT campaign, conducted in Cabauw, the Netherlands in 2014, aimed to assess the capabilities of both the main peak approach and the spectrally resolved approach."**

**Based on earlier comment (see instrumentation section), replace the sentence by "One of the aims of the ACCEPT campaign, conducted in Cabauw, the Netherlands in 2014, was to assess the capabilities of both the main peak approach and the spectrally resolved approach."**

**6-c)** We have revised the sentence as recommended to improve accuracy (lines 615-616).

**7) Acknowledgements**

**Line 653: "..... and to the team of the CESAR Observatory in Cabauw and Delft University of Technology, NL, for their support ...."**

We have updated the acknowledgment to include the CESAR Observatory team in Cabauw and Delft University of Technology, NL, for their support (lines 635-638).

**Minor comments/corrections/adding's**

**1) Abstract**

**The abstract consists of three parts identical. Please, correct this.**

Apologies for the formatting issue. We corrected it.

**2) Introduction**

**Lines 87-89: The following paper could be added:**

**Mak, H. Y. L., and Unal, C. (2025): Peering into the heart of thunderstorm clouds: insights from cloud radar and spectral polarimetry. Atmospheric Measurement Techniques, 18(5), 1209–1242. https://doi.org/10.5194/amt-18-1209-2025**

We have added the recommended reference (Mak and Unal, 2025) to the relevant section of the manuscript (lines 63)

**3) Instrumentation**

**3-a) Line 114: remove "also"**

**3-a)** We have removed "also" from line 89 as requested.

**3-b) End of Table 1, typo: "...... can be estimated by assuming ...."**

**3-b)** We have corrected it to "can be estimated by assuming" at the end of Table 1.

**4) Mira-35 radar in hybrid mode**

**4-a) Lines 144-145: "....... which are representative of the horizontally (Eh(ω)) and vertically (Ev(ω)) components of the received waves." Also, place a dot above the capital letter E. These components are complex.**

**4-a)** We have updated the notation to include a dot above the capital letter E to indicate the complex nature of the horizontal and vertical components (lines 120-121).

**4-b) Line 148: "ZDR quantifies the ratio of reflectivity measurements in horizontal (Zhh, Eq. 1) and vertical (Zvv, Eq. 2) polarizations, (Eq. 3)."**

**4-b)** We have revised the sentence for clarity to correctly describe the definition of ZDR (line 124).

**4-c) Line 153, typo: Also, the constants .....**

**4-c)** We have corrected it to "Also, the constants …" on line 129.

**4-d) Line 154: Remove "Radar cross section (RCS)". The constants C1 and C2 do not depend on the RCS, Eh(ω) and Ev(ω) do depend on the RCS (square-root of RCS).**

**4-d)** We have removed the mention of "Radar cross section (RCS)" from line 130, as C1 and C2 do not depend on it. The dependency on RCS applies to $E_h(\omega)$ and $E_v(\omega)$ instead.

**4-e) Lines 158-160: "The correlation coefficient (RHV) is a crucial parameter (defined by Eq. 4) that quantifies the similarity between horizontally (Eh(ω)) and vertically (Ev(ω)) polarized backscattered signals." Place a dot above the capital letter E.**

**4-e)** However, in our study, we work with Zhh and Zvv rather than the complex electric field components $E_h(\omega)$ and $E_v(\omega)$. Accordingly, ρhv is defined in terms of these reflectivity-based quantities.

**4-f) Lines 160-162: "It provides insight into the diversity of particle shapes and orientations within a radar resolution volume and is typically expressed as a value between 0 and 1."**

**4-f)** We have revised the sentence in lines 135 - 136 as follow: "It provides insight into the diversity of particle shapes and orientations within a radar resolution volume and is typically expressed as a value between 0 and 1."

**4-g) Lines 162-163: "A correlation coefficient of 1 indicates perfect correlation between horizontal and vertical polarization signals, implying that the backscattered signals are identical in phase and amplitude."**

**4-g)** Thanks for suggesting this alternative way for explanation of the correlation coefficient. We have revised the sentence to clearly state that a correlation coefficient of 1 indicates perfect similarity between horizontal and vertical polarization signals, typically implying uniform particle type, shape, and orientation within the radar resolution volume (lines 138-139).

**5) Main-peak approach**

**5-a) In Eq. (8), remove the subscript "e", which is not present in the text.**

**5-a)** To maintain consistency with the original paper and ensure clarity, we have added the subscript "e" in the text to match the notation used in Eq. (8) (line 176).

**5-b) Line 211: Is ξg the axis ratio?**

**5-b)** Yes, ξg denotes the axis ratio in this context. We also added it in line 182.

**5-c) Line 226: "Finally, to derive ZDR and RHV, the horizontal and vertical complex scattering amplitudes are calculated using the polarizability ratio and the degree of orientation across different elevation angles. "**

**5-c)** We have revised the sentence in lines 196-197 as follows: "The complex scattering amplitudes are derived using the polarizability ratio and degree of orientation, which vary with elevation angle, to calculate ZDR and ρHV".

**5-d) Line 234, typo: "distribution".**

**5-d)** We have corrected "distrubution" to "distribution" at line 211.

**5-e) Line 238, typo: Not "than" but "then".**

**5-e)** We have corrected "than" to "then" at line 215.

**6) Spectrally resolved approach**

**6-a) Lines 250-253: "The width of the Doppler spectra is thereby characterized by size- and shape-dependent fall velocities of the particles, which are super-imposed by influences of turbulence and (predominantly in case of off-zenith antenna pointing angle) horizontal wind variability that cause additional broadening of the spectrum (Radenz et al., 2019)."**

**6-a)** corrected

**6-b) Line 274, typo: "mean square error function"**

**6-b)** We have corrected it to "mean square error function" at line 251.

**6-c) Line 279: "During averaging of all ZDR and RHV values ….." . Mention how many values can be expected for this average.**

**6-c)** We added a text in lines 257-259 as follows: "It should be noted that the number of values contributing to the average depends on the elevation angle resolution of the RHI scan; in this study, typically 121 elevation angles are used."

**7) The influence of air motion on the Doppler spectra observed by a scanning cloud radar**

**Line 295: …. onto the gravitational downward motion of particles ….**

**Line 295:** In our setup with zenith-pointing radar, the hydrometeors fall due to gravity, so their motion is downward. Therefore, we have retained "gravitational downward motion" to accurately describe the physical situation (line 273).

**8) Retrieval of horizontal wind**

**8-a) Line 306: ".... and radial wind velocity Vw."**

**8-a)** We have replaced V_w at line 284.

**8-b) Eq. (12): is it not Vw=Vh cos cos(β+π-α)? To have negative radial wind velocity when the particles approach the radar. The wind direction being defined from where the wind comes from.**

**8-b)** You are correct that to obtain a negative radial wind velocity when particles approach the radar, the wind direction defined as where the wind comes from needs to be accounted for. Instead of adding $\pi$ in the equation, we applied a negative sign to the term to achieve the correct sign convention (line 295).

**9) First case study 07 Nov 2014, 09:15-09:30: retrieval of various hydrometeor types**

**9-a) Figure 4 caption: ……. (see Figs. 7-9). Typo in the right parenthesis.**

**9-a)** We have corrected the right parenthesis in the caption of Figure 4.

**9-b) Line 391, typo: …. high values of radar LDR ….**

**9-b)** We have corrected the phrase to "high values of radar LDR" at line 369.

**9-c) Line 444: …. The SNR stabilizes at approximately 60 dB. I think it is much less. 25 dB?**

**9-c)** We have double-checked the data and agree that the SNR stabilizes closer to approximately 25 dB. We have corrected this value in line 422.

**9-d) Lines 457-458: "Since different parts might identify oblate-shaped particles at heights around 3km (parts 1, 2, 3, and 4) and prolate-shape particles at part 5, a height of 3km is chosen for a detailed depiction of the data analysis procedure." From the discussion of Figure 7 above, the authors mention that at this height (around 3 km), the particles are spherical-shaped in parts 1-3, oblate-shaped in part 4 and prolate-shaped in part 5. Therefore, rephrase the sentence for consistency with the discussion of Figure 7.**

**9-d)** We have revised the sentence to ensure consistency with the discussion of Figure 7, clarifying that at around 3 km, particles are spherical-shaped in parts 1–3, oblate-shaped in part 4, and prolate-shaped in part 5 (line 435).

**9-e) Figure 8 caption: I suppose that the time period 09:16:09 – 09:18:08 UTC is considered and not only the time 09:15:09 UTC.**

**9-e)** We have clarified in the caption of Figure 8 that the time period 09:16:09 – 09:18:08 UTC is considered, rather than only the single time 09:15:09 UTC.

**9-f) Figure 8 caption: ….. and for each part of the spectrally resolved spectra, shape and orientation retrieval ……"**

**9-f)** We have updated the Figure 8 caption to clarify that shape and orientation retrieval is performed for each part of the spectrally resolved spectra.

**9-g) Figure 9: I don't see the horizontal lines displaying the temperature levels.**

**9-g)** We have removed the statement referring to horizontal lines displaying temperature levels in Figure 9.

**10) Second case study 03 Nov 2014, 20:30-20:45: Secondary ice formation**

**10-a) Lines 528-529: "In middle altitudes (X-Y km) ........ In lower altitudes (Z-X km) ....." Be specific and indicate the height range considered.**

**10-a)** We have revised the text to specify the exact height ranges referred to as "middle" and "lower" altitudes (line 512).

**10-b) Figure 12 caption: the time (20:01:09-20:11:26) is not correct.**

**10-b)** We have corrected the time in the caption of Figure 12 to reflect the accurate time interval. (20:31 – 20:41)

**10-c) Line 542: ".... between the melting layer and 2.75 km height ......"**

**10-c)** We have corrected the phrase to read "between the melting layer and 2.75 km height" in line 523.

**10-d) Line 558: "... RHV decreases with increasing part number ...."**

**10-d)** We have clarified the sentence to state that "RHV decreases with increasing part number" in line 539.

**10-e) Figures 16 and 17 caption: ..... and for each part of the spectrally resolved spectra, shape and orientation retrieval ......"**

**10-e)** We have updated the captions of Figures 16 and 17 to clarify that shape and orientation retrieval is performed for each part of the spectrally resolved spectra.

**11) Summary and Conclusions**

**Lines 602-603: "In particular, by incorporating Rayleigh scattering, we assume that particles are small with respect to the deployed electromagnetic waves." I suggest to remove this sentence. It is redundant compared to the next sentence. The next sentence suffices.**

**Lines 602-603:** We agree that the sentence is redundant and have removed it to improve clarity and avoid repetition.

**Line 611, typo: .... were elaborated .....**

**Line 611:** The typo has been corrected to "were elaborated" in line 602.

**Line 638:" ........ would be to apply implement techniques ....." Suppress either "apply" or "implement".**

**Line 638:** We have corrected the phrase by removing "apply" to avoid redundancy (line 632).

**Line 639, typo: "Then more insights ….."**

**Line 639:** We have corrected the phrase to "The more insights …" in line 633.

**Response to Report #2**

**Answer to AR1-1B: I am still not convinced that just dividing the spectrum into 5 parts allows you to see 5 distinct hydrometeor types. It can very likely be that two hydrometeors (i.e. needles and aggregates as they can have the same fall velocity) are in the same part, therefore you will still see the combined effect of both species. I understand that applying a peak identification algorithm on the slant spectra is difficult, however, I would have tested a few more subjective ways of separation into n parts based on i.e. the shape of your Doppler spectrum (i.e. the slope of sZDR can tell you when another species is introduced, or some other measure). The arbitrary selection of 5, especially if you can not even distinguish between rimed particles and aggregates seems random. However, I am fine with having it published like this, as long as you make it clear that separating into these 5 parts does not mean that you do not have a mixture of ice particles in each part and can therefore miss e.g. the appearance**

**Answer to AR1-1B:** We fully agree that dividing the Doppler spectrum into five parts does not guarantee the exclusive presence and identification of a distinct hydrometeor type in each part. As Reviewer #2 correctly pointed out, species such as needles and aggregates can share similar fall velocities and thus overlap within the same spectral region, leading to a mixture of particle types in a given part. Practically, there is to our knowledge currently no way to distinguish different particle types which show the same Doppler velocities. Unfortunately, our technique does not make a difference here. But what the difference to other established techniques is, that we at least aim to tackle the complex topic of identification of multiple hydrometeor types in the same volume of air. And to our opinion, we go a step further compared to other studies. E.g., vertically pointing polarimetric Doppler cloud radar measurements show increasing capability to assign crystal habits to distinct peaks in the Doppler spectra (e.g., Vogl et al., 2024). However, while having the advantage of being applicable to basically any vertical-stare polarimetric Doppler cloud radar observation, these techniques suffer of two deficiencies: (1) individual peaks must be identifyable, and (2) it cannot be distinguished between oblate and isometric/spherical hydrometeors since both show similar polarimetric properties in zenith-pointing mode. Data from polarimetric scanning Doppler weather radars, in turn, are since quite some time used for identification of hydrometeor types (e.g., Marzano et al., 2006; Chandrasekar et al., 2013). However, so far, only one hydrometeor type per data point (i.e., unit of volume) is identified. For identification of several co-located hydrometeors in a volume, currently only dual-wavelength polarimetric radar techniques are discussed (Pejcic et al., 2025). Our technique overcomes both of these issues, even though be it on the expense of higher complexity by means

of the need to relate and analyzed Doppler spectra of polarimetric variables from different elevation angles, which incorporates the necessary assumption of horizontal homogeneity. Nevertheless, horizontal homogeneity is for our approach by far not as an issue as for the establishment of quasi-vertical profiles from weather radar PPI scans (Ryzhkov et al., 2016).

The specific choice of five Doppler parts was based on a balance between resolution and practical interpretability, rather than an exact physical separation of all hydrometeor species. We acknowledge that this splitting is somewhat arbitrary and does not perfectly separate rimed particles from aggregates or other mixed-phase components.

Regarding your suggestion to explore more subjective or data-driven criteria for spectral partitioning—such as incorporating slope changes in sZDR or other Doppler spectral shape metrics—we recognize the potential benefits of these approaches. Indeed, these could offer more physically meaningful splitting by identifying transitions between dominant hydrometeor types. Due to the complexity and computational cost, however, we limited our scope in this study but consider this an important avenue for future work.

To clarify this limitation in the manuscript, we will explicitly state that the division into five parts represents an approximate classification (lines 240-242). We emphasize that each part may contain mixed hydrometeor populations and that the method does not guarantee the full separation of all particle types (lines 245-261).

**Answer to AR1-2: to me it was not clear that if you assume Rayleigh that then there is a direct relationship between the ZDR/RHV and the shape of the particle. Perhaps you can say that in one sentence in the manuscript. About the wind retrieval: See comment to Line 240**

**Answer to AR1-2:** Rayleigh scattering applies when the particle size is much smaller than the radar wavelength. In this regime, the scattering properties depend mainly on the particle's shape, orientation, and dielectric properties, rather than complex resonance effects seen at larger sizes (Mie or non-Rayleigh scattering).

Because of this, differences in the returned radar signals' polarization (like the difference between horizontal and vertical reflectivity, i.e., ZDR, or polarization ratio RHV) can be directly attributed to how the particle's shape affects the scattering.

So, when Rayleigh conditions hold, ZDR and RHV and their elevation dependencies become reliable indicators of the particle's aspect ratio and orientation, allowing us to infer shape characteristics from these measurements.

We rephrased and extended the text between lines 172-195 to highlight the relationship between ZDR & RHV to polarizability ratio and degree of orientation in the case of Rayleigh scattering, as follows: "In the simulation part of the method, ZDR and RHV values are calculated for many combinations of particle shapes, densities (related to their refractive index), and orientations, across a wide range of elevation angles (from 30◦ to 150◦). For doing so, Myagkov et al. (2016a) utilized spheroidal (Rayleigh-) scattering theory which enables to establish a direct relationship

between the observables elevation angle, ZDR, and RHV and the particle's properties of density-weighted axis ratio and the distribution of the canting angles. The density-weighted axis ratio is denoted polarizability ratio $\xi e$ which can be represented as the ratio of polarizability elements p1 and p2 (Eq. 8):

$$\xi e = \langle p2 \rangle / \langle p1 \rangle \qquad (Eq.8)$$

which polarizability elements p1 and p2 are defined as Eq. 9:

$$p1,2 = V \ \varepsilon0( \ \varepsilon r - 1)A1,2(\xi g ) \qquad (Eq.9)$$

where V is the volume of the spheroid, $\varepsilon0$ is the vacuum permittivity, $\varepsilon r$ is the relative permittivity, and A1,2($\xi g$ ) are function of the axis ratio $\xi g$ . The polarizability ratio ranges from 0.3 to 2.3. Within this range, values of $\xi = 0.3$ represent strongly oblate particles, $\xi = 2.3$ indicate strongly prolate particles, and $\xi = 1$ signifies centrally positioned spherical particles. As the polarizability ratio is also a function of the particle refractive index, i.e., density, its absolute value approaches unity for values of very low density. This aspect has to be considered in the interpretation of $\xi$.

The general orientation distribution of the hydrometeor population is described by the degree of orientation κ (Hendry et al., 1976) and Eq. 10."

**Line 240: if horizontal wind indicates inhomogeneity, then correcting for the wind does not solve that problem. Or are you using the wind field in some other way? If so please explain that in more detail. In your comment AR1-2 you say that you use the wind field as a measure of homogeneity, how do you do that? What are the criteria then that say the field is homogeneous enough to be considered for the retrieval? You further said in your comment AR1-4 that "moving the Doppler spectra to 0 does not have the same effect" how is that possible? With the wind correction you are doing exactly that: shifting the Doppler spectrum by an arbitrary number. The elevation dependency of the Doppler spectrum width has nothing to do with the horizontal wind, but with the viewing geometry of the radar. In fact, looking at our own data, the spectral width is exactly the same if viewed at e.g. 30° or 90° elevation without correcting for the wind. The width of the spectrum is influenced by wind shear and turbulence, not the wind speed. In comment AR1-4 you further say that you are using the Doppler spectrum width as a measure for homogeneity. This is not the same as the answer in AR1-2, where you say you use the wind field. Which of the two are you using and how exactly?**

**Line 240:** Thank you for your detailed and very helpful comments regarding the use of horizontal wind and Doppler spectrum width as measures of homogeneity and the wind correction procedure. Concerning the question of Reviewer #2, if we do use the the wind field in some other way, we would like to emphasize that the overall scheme of de-aliasing and horizontal-wind correction was developed as one 'package' in the framework of the development of the spectrally-resolved approach. While the de-aliasing is definitely a must-have for the

technique, the horizontal wind correction was implemented as a valuable add on. The reasons are listed again below:

1. **Use of horizontal wind as a measure of homogeneity:**
   We do not simply treat horizontal wind speed as a direct indicator of homogeneity or inhomogeneity. Instead, the variability or consistency of the horizontal wind field over the radar sampling volume and time is used as an indicator. In other words, if the wind field is stable and uniform (low spatial and temporal variability), we consider the volume to be more homogeneous, which supports the assumptions underlying the retrieval. Conversely, high variability suggests inhomogeneity.
2. **Wind correction and shifting the Doppler spectrum:**
   The wind correction applied shifts the Doppler spectra to center the dominant fall velocity around zero, compensating for the mean horizontal wind's Doppler shift component projected onto the radar line-of-sight. However, this correction does not remove the underlying spectral width or its physical causes such as turbulence or wind shear. Therefore, shifting the spectrum by the mean wind velocity is not equivalent to removing the spectral broadening or inhomogeneity. This distinction is crucial and will be elaborated in the revised text.
3. **Elevation dependency of the Doppler spectrum width:**
   You are correct that the spectral width's elevation dependence is primarily due to the radar viewing geometry and physical processes like turbulence and wind shear, not simply the mean wind speed.
4. **Distinction between wind field and Doppler spectrum width as homogeneity measures:**
   We actually only use the homogeneity of the wind field as a quality criteria. We apologize for the raised confusion and for the inconsistent statements about the application of spectral width as criteria for evaluation of the horizontal homogeneity in the first revision round.

Having written this, we acknowledge that the horizontal wind correction is not absolutely necessary for the spectrally resolved approach. But it provides additional constraints for the interpretation of the retrieval.

**Specific comments:**

**Line 186: please define vdsp**

**Line 186:** We added a brief description at line 164 defining the VDPS method as the vertical distribution of particle shape.

**Line 259: I would name the 5 shapes here specifically, so that the reader knows what you want to distinguish**

**Line 259:** While we cannot definitively assign specific hydrometeor shapes to each of the five spectral parts due to overlap and mixtures, we provide here a general description of the typical

particle types expected to dominate different velocity ranges. We added further information about the typical fall velocities of hydrometeors to lines 63-66.

**Line 284: I still don't understand the word harmonize, or the sentence behind it, this was not made clear to me in AR1-4! Please elaborate on that further! First: what does harmonize mean. Second: why do you need to do that for retrieving the wind field?**

**Line 284:** The horizontal wind causes shifts in the observed Doppler velocity depending on the radar's elevation angle, complicating direct comparison of spectra across angles. To retrieve a consistent vertical wind profile and particle fall velocities, these horizontal wind-induced shifts must be accounted for (i.e., harmonized) across all elevation angles. Without this step, the Doppler spectra cannot be reliably compared or combined to infer the wind field and hydrometeor properties accurately.

**Figure 1 and comment AR1-26: your statement that turbulence is represented by an upward motion of particles is wrong. Turbulence can go upwards and downwards. Folded with the particles movements, it broadens the spectrum. Most of the times this does not even result in an actual upward motion of your particles, so indicating turbulence in Figure 1 as the distance between the 0m/s and the slow edge is wrong. Further, turbulence flattens the spectrum. So if you want to include turbulence in the figure please include all aspects correctly or just don't put it in the figure!**

**Figure 1 and comment AR1-26:** We removed the turbulence effect in the Figure and added in the caption that turbulent motion is not considered in the sketch.

**Fig 9 is different from the preprint, why? Now you corrected line 474, but now it is clearly wrong again, if you take your new figure. Now I would say that part 1,5 have a pol. Ratio of 0.75 and part 2,3,4 of 1.25 (also clearly not 0.9 and 1.1 if you look at the figure)**

**Fig 9:** We apologize for the inconsistency between Fig. 9 and the preprint. The figure was updated to reflect the latest analysis. You are correct that the polarization ratio values for parts 1 and 5 should be around 0.75, while parts 2, 3, and 4 are closer to 1.25. We corrected the numbers in the text between lines 450 and 455.

**Perhaps you can explain to me again why in case study 1 you come to the conclusion that you have no SIP and in case study 2 you say you have SIP. Just because you could not see a liquid layer in which ice crystals might have been nucleated does not mean it is not there, especially since you clearly have riming in case study 2. So your argument for no SIP in case study 1 can also be given here for case study 2.**

In **case study 1**, we analyze two time periods. The first one is a deep cloud system. Ice particles from a higher-level cloud layer sediment through a supercooled liquid layer. At the height of the layer, we identified columnar ice. Just from this time period, one would conclude that the

columnar ice was formed by SIP during the seeding event. But, shortely afterward, the lower-level supercooled liquid cloud was still present withouth any seeding cloud above. But still, this supercooled liquid cloud formed columnar ice crystals. As seeding is absent during this time period, one can exclude SIP. It is more likely that the ice in the lower-level supercooled liquid layer formed primarily.

In contrast, **case study 2** the retrieval identified oblate-shaped particles at all levels of the cloud layer. Only at heights below 2.8 km and the melting layer suddenly columnar crystals appear, as well, in the lowest-falling  Doppler spectra part. We base our conclusion that this is SIP, because the time-height cross-sections of the radar variables (Fig. 10) show an unsteady, patchy, occurrence of sigantures in the height range where we suspect the SIP. In contrast, during case study 1, the columnar ice was formed rather constantly over time.

Nevertheless, we acknowledge the concern of Reviewer #2 that indeed also in this case formation of primary ice similar to Case Study 1 could have happen. Since precipitation sediments constantly from higher altitudes, we cannot evaluate (a) if there was a liquid layer at 2.8 km height, and (b) if that layer would have formed primary ice even without the seeding by the higher-level cloud. We acknowledge this fact at the end of the discussion of case study 2 in lines 560-582.

**Line 603-605: I don't understand this sentence! You assume Rayleigh, therefore your particles have to be small with respect to the wavelength. This does not mean that you can accurately model the scattering properties of all ice particles. The statement that it "ensures the proper treatment of scattering properties which is critical for precise radar measurements and interpretations" does not make sense to me. How does your assumption of Rayleigh ensure that? Anyway, maybe rephrasing that helps, or just leaving the entire sentence out, as all the important things are already said with the sentence before.**

**Line 603-605:** We agree that the original sentence may have been misleading. The assumption of Rayleigh scattering implies that particles are small relative to the radar wavelength, which simplifies the scattering calculations. However, it does not guarantee perfectly accurate modeling of all ice particle scattering properties, especially for larger or complex shapes. We agree that the original wording was misleading. We have modified the text to clarify that the Rayleigh assumption (lines 595-596). "provides a consistent framework for approximating scattering properties, which is important for interpreting radar measurements." This phrasing better reflects the limitations and practical benefits of the assumption without implying complete accuracy.

**Line 611-614: I don't understand what you mean with "this was not applied yet in the present study"? I thought you separated into 5 parts already?**

**Line 611-614:** The phrase "this was not applied yet in the present study" refers to the fact that, although the spectrally-resolved retrieval method allows exclusion of the fastest-falling Doppler spectrum parts—typically corresponding to the largest particles prone to non-Rayleigh

scattering—we did not implement this exclusion in the current analysis. We plan to apply this filtering approach in future follow-up studies to improve retrieval accuracy.

**Line 615: What do the non-Rayleigh effects have to do with horizontal homogeneity?**

**Line 615:** Non-Rayleigh effects and horizontal homogeneity are both challenges we consider in the analysis, but they address different issues and are not necessarily directly related.

**Line 620: what does "appropriateness of horizontal wind correction" mean? Please be more specific with your method!**

**Line 620:** It refers to the effectiveness and suitability of the correction method in isolating the vertical fall velocities of hydrometeors from the measured radar signal.

**References:**

V. Chandrasekar, R. Keranen, S. Lim, D. Moisseev, Recent advances in classification of observations from dual polarization weather radars, Atmospheric Research, Volume 119, 2013, Pages 97-111, ISSN 0169-8095, https://doi.org/10.1016/j.atmosres.2011.08.014 .¶

Marzano, F. S., Scaranari, D., Celano, M., Alberoni, P. P., Vulpiani, G., and Montopoli, M.: Hydrometeor classification from dual-polarized weather radar: extending fuzzy logic from S-band to C-band data, Adv. Geosci., 7, 109–114, https://doi.org/10.5194/adgeo-7-109-2006, 2006. ¶

Pejcic, V., Mroz, K., Muhlbauer, K., and Tromel, S.: Hydrometeor partitioning ratios for dual-frequency space-borne and polarimetric ground-based radar observations, EGUsphere [preprint], https://doi.org/10.5194/egusphere-2025-1414, 2025. ¶

Ryzhkov, A., P. Zhang, H. Reeves, M. Kumjian, T. Tschallener, S. Tromel, and C. Simmer, 2016: Quasi-Vertical Profiles—A New Way to Look at Polarimetric Radar Data. *J. Atmos. Oceanic Technol.*, **33**, 551–562, https://doi.org/10.1175/JTECH-D-15-0020.1. ¶

Vogl, T., Radenz, M., Ramelli, F., Gierens, R., and Kalesse-Los, H.: PEAKO and peakTree: tools for detecting and interpreting peaks in cloud radar Doppler spectra – capabilities and limitations, Atmos. Meas. Tech., 17, 6547–6568, https://doi.org/10.5194/amt-17-6547-2024, 2024.

---

## Author Response (AR3)

**Responses to Review Comments on "Identification of multiple co-located hydrometeor types in Doppler Spectra from scanning polarimetric cloud radar observations"**

Majid Hajipour[1], Patric Seifert[1], Hannes Griesche[1], Kevin Ohneiser[1], and Martin Radenz[1]

1. Leibniz Institute for Tropospheric Research, Leipzig, Germany

29 July 2025

We sincerely appreciate the editor's thorough review and thoughtful feedback provided throughout both rounds. Their valuable input has significantly contributed to enhancing the clarity and overall quality of the manuscript. Below we provide our responses to the 4 remaining comments, including the updated text passages. References to line numbers refer to the revised manuscript (without tracked changes).

Sincerley,

Majid Hajipour and co-authors.

**Editor Comment #1:**

— **Abstract lines 7-8: the paper does not bring forth that you can ultimately differentiate five hydrometeor types**

**Authors Response #1:**

We revised lines 5-9 of the abstract as below:

*"The previously developed main-peak approach focuses only on the part of the Doppler spectrum with the highest signal-to-noise ratio to retrieve the shape and orientation of the dominant hydrometeor types within stratiform clouds. With the extended technique, referred to as the spectrally resolved approach, the section of the Doppler spectrum containing valid data points exceeding the noise level is analyzed by dividing it into five equally spaced parts. This allows to retrieve up to five distinct velocity-segregated hydrometeor types."*

Aside the update on the "5-parts issue", we also introduced a few typographical/grammar corrections to the abstract.

**Editor Comment #2:**

— **Caption Fig. 1: explicitly state that you are showing an idealized picture, otherwise the figure is misleading**

**Authors Response #2:**

We updated the caption of Fig. 1 as follows:

*"Illustration of an idealized Doppler spectrum containing 5 different hydrometeor types. The division into five parts enables the potential identification of up to five different hydrometeor types; however, the actual number of*

*retrievable distinct types depends on the spectral and microphysical characteristics of the observed cloud. Positive Doppler velocities indicate the impact of turbulent motion on the Doppler spectrum."*

**Editor Comment #3:**

— **In Section 3.2, the division into five classes needs to be physically justified. Especially the sentences "This division into five parts was chosen because it is the best estimate to represent different hydrometeor types, accounting for various shapes" and "The amount of 5 parts was empirically chosen for this study, because usually not more than that amount of different particle shapes can be expected in a cloud volume" need a quantitative reasoning, respectively confirmation through citation of relevant literature.**

**Authors Response #3:**

We revised the discussion on the choice of five spectral parts in Sect. 3.2 by modifying/extending the first two paragraphs of this section as below (lines 223-265). Indeed, there was no study yet which stated that never more than 5 hydrometeor types exist. But the existing studies which dealt with Doppler spectra separation usually don't report more than a maximum of 5 different peaks. For instance the PeakTree-related publications of Radenz and Vogl. We hope that this situation becomes more evident in the updated text.

[revised manuscript text omitted]

**Editor Comment #4:**

— **Instead of the Doppler spectrum depiction in terms of hydrometeor classes, I could imagine that argumenting with a composite analysis of different Doppler spectrum regimes could resolve these issues.**

**Authors Response #4:**

We also appreciate the reviewer's suggestion to frame the analysis as a composite of different Doppler spectrum regimes. We agree with this framing and have added a clarifying paragraph (Section 3.2 and Conclusion) to emphasize that our approach allows decomposition of the Doppler spectrum into interpretable regimes rather than strict classification into physical hydrometeor types. The paragraph is:

In Section 3.2, we now write in lines 260-261:

*"Generally speaking, instead of spectral peaks, the spectrally resolved approach aims on identification of spectral regimes of distinct hydrometeor properties."*

In the conclusions section we write in lines 606-613:

*"Instead of interpreting the Doppler spectrum segmentation as a strict classification of hydrometeor types, we emphasize that our approach provides a decomposition of the spectrum into distinct Doppler regimes. These*

*regimes reflect variations in particle fall velocities and, by extension, microphysical characteristics such as size, shape, or phase. This composite view allows us to capture the internal variability of the cloud without assigning specific hydrometeor classes to each spectral part. Such a representation aligns with previous studies (e.g., Shupe et al., 2004; Kollias et al., 2007), which observed that Doppler spectra often exhibit multiple peaks or broadened components due to the coexistence of diverse hydrometeor populations. By dividing the spectrum into five parts, we aim to resolve this complexity in a practical and interpretable way, facilitating further analysis of cloud microphysics and dynamics."*